# Scenario set-up and forcing data for impact model evaluation and impact attribution within the third round of the Inter-Sectoral Model Intercomparison Project (ISIMIP3a)

Katja Frieler[1], Jan Volkholz[1], Stefan Lange[1], Jacob Schewe[1], Matthias Mengel[1], María del Rocío Rivas López[1], Christian Otto[1], Christopher P.O. Reyer[1], Dirk Nikolaus Karger[2], Johanna T. Malle[2], Simon Treu[1], Christoph Menz[1], Julia L. Blanchard[3], Cheryl S. Harrison[4], Colleen M. Petrik[5], Tyler D. Eddy[6], Kelly Ortega-Cisneros[7], Camilla Novaglio[3], Yannick Rousseau[3], Reg A. Watson[3], Charles Stock[8], Xiao Liu[9], Ryan Heneghan[10], Derek Tittensor[11], Olivier Maury[12], Matthias Büchner[1], Thomas Vogt[1], Tingting Wang[13], Fubao Sun[13], Inga J. Sauer[1,14], Johannes Koch[1], Inne Vanderkelen[15,16,17], Jonas Jägermeyr[1,18,19], Christoph Müller[1], Sam Rabin[20], Jochen Klar[1], Iliusi D. Vega del Valle[1], Gitta Lasslop[21], Sarah Chadburn[22], Eleanor Burke[23], Angela Gallego-Sala[24], Noah Smith[22], Jinfeng Chang[25], Stijn Hantson[26], Chantelle Burton[23], Anne Gädeke[1], Fang Li[27], Simon N. Gosling[28], Hannes Müller Schmied[21,29], Fred Hattermann[1], Jida Wang[30], Fangfang Yao[31], Thomas Hickler[21], Rafael Marcé[32,33], Don Pierson[34], Wim Thiery[15], Daniel Mercado-Bettín[32], Robert Ladwig[35,] Ana I. Ayala[34], Matthew Forrest[21], Michel Bechtold[36]

Affiliations:
[1]Potsdam Institute for Climate Impact Research, 14473 Potsdam, Germany
[2]Swiss Federal Research Institute WSL, Zürcherstrasse 111, 8903 Birmensdorf, Switzerland
[3]Institute for Marine and Antarctic Studies, University of Tasmania, Hobart, Tasmania, Australia
[4]Department of Ocean and Coastal Science and Center for Computation and Technology, Louisiana State University, Baton Rouge, Louisiana, USA
[5]Scripps Institution of Oceanography, University of California San Diego, CA, USA
[6]Centre for Fisheries Ecosystems Research, Fisheries & Marine Institute, Memorial University, St. John's, NL, Canada
[7]Department of Biological Sciences, University of Cape Town, Rondebosch, Cape Town, 7701, South Africa
[8]NOAA/OAR/Geophysical Fluid Dynamics Laboratory, Princeton, NJ, United States
[9]SAIC@NOAA/NWS/NCEP Environmental Modeling Center, 5830 University Research Court, College Park, MD 20740
[10]School of Mathematical Sciences, Queensland University of Technology, Brisbane, QLD, Australia
[11]Department of Biology, Dalhousie University, Halifax, Nova Scotia, Canada, B3H 4R2
[12]Institute for Research for Development, UMR 248 MARBEC, France.
[13]Key Laboratory of Water Cycle and Related Land Surface Processes, Institute of Geographic Sciences and Natural Resources Research, Chinese Academy of Sciences, Beijing 100101, China
[14]Institute for Environmental Decisions, ETH Zurich, Zurich, Switzerland
[15]Vrije Universiteit Brussel, Department of Hydrology and Hydraulic Engineering, Brussels, Belgium
[16]Wyss Academy for Nature, University of Bern, Bern, Switzerland
[17]Climate and Environmental Physics and Oeschger Center for Climate Change Research, University of Bern, Bern, Switzerland
[18]NASA Goddard Institute for Space Studies, New York, NY 10025, USA
[19]Columbia University, Climate School, New York, NY 10025, USA

[20]Climate and Global Dynamics Laboratory National Center for Atmospheric Research Boulder, CO
80302, USA
[21]Senckenberg Leibniz Biodiversity and Climate Research Centre (SBiK-F), Frankfurt am Main,
Germany.
[22]Department of Mathematics, University of Exeter, Exeter UK
[23]Met Office Hadley Centre, Fitzroy Road, Exeter, UK
[24]Geography Department, University of Exeter, Exeter, UK
[25]College of Environmental and Resource Sciences, Zhejiang University, Hangzhou, China
[26]Faculty of Natural Sciences, Universidad del Rosario, Bogotá, Colombia
[27]International Center for Climate and Environment Sciences, Institute of Atmospheric Physics, Chinese
Academy of Sciences, Beijng, China
[28]School of Geography, University of Nottingham, Nottingham, UK
[29]Institute of Physical Geography, Goethe University Frankfurt, Frankfurt am Main, Germany
[30]Department of Geography and Geospatial Sciences, Kansas State University, Manhattan, Kansas,
USA
[31]Environmental Resilience Institute, University of Virginia, Charlottesville, Virginia 22903, USA
[32]Catalan Institute for Water Research (ICRA), 17003 Girona, Spain
[33]Universitat de Girona, Girona, Spain
[34]Uppsala University, Norbyvägen 18 D, 752 36 Uppsala, Sweden
[35]Center for Limnology, University of Wisconsin-Madison, Madison, Wisconsin 53706, USA
[36]KU Leuven, Department of Earth and Environmental Sciences, Leuven, Belgium
*Correspondence to:* Katja Frieler (katja.frieler@pik-potsdam.de)
**Abstract.** This paper describes the rationale and the protocol of the first component of the third
simulation round of the Inter-Sectoral Impact Model Intercomparison Project (ISIMIP3a,
www.isimip.org) and the associated set of climate-related and direct human forcing data (CRF and
DHF, respectively). The observation-based climate-related forcings for the first time include high-
resolution observational climate forcings derived by orographic downscaling, monthly to hourly coastal
water levels, and wind fields associated with historical tropical cyclones. The DHFs include land use
patterns, population densities, information about water and agricultural management, and fishing
intensities. The ISIMIP3a impact model simulations driven by these observation-based climate-related
and direct human forcings are designed to test to what degree the impact models can explain observed
changes in natural and human systems. In a second set of ISIMIP3a experiments the participating
impact models are forced by the same DHFs but a counterfactual set of atmospheric forcings and
coastal water levels where observed trends have been removed. These experiments are designed to
allow for the attribution of observed changes in natural, human and managed systems to climate
change, rising $CH_4$ and $CO_2$ concentrations, and sea level rise according to the definition of the Working
Group II contribution to the IPCC AR6.
**1 Introduction**
The Inter-Sectoral Impact Model Intercomparison Project ISIMIP (www.isimip.org) provides a common
scenario framework for cross-sectorally consistent climate impact simulations currently covering the
following sectors: agriculture (global; in cooperation with AgMIP's Global Gridded Crop Model
Intercomparison Project (GGCMI)), water (global and regional), lakes (global and regional), biomes
(global), forest (regional), fisheries and marine ecosystems (global and regional), terrestrial biodiversity
(global), fire (global), permafrost (global), peat (global), coastal systems (global), energy (global), health
(temperature-related mortality; water-borne diseases; vector-borne diseases; and food security and
nutrition) (global and local), and labour productivity (global and local). The impact model simulations
are made freely available, allowing for all types of follow-up analysis. The consistent design of the
simulations does allow for the comparison of climate impact simulations within each sector. However,
it also enables the bottom-up integration of impacts across sectors. Thus, it provides a unique basis for
the estimation of the effects of climate change on, e.g., the economy, displacement and migration,
health, or water quality resolving the mechanisms along different impact channels and fully exploiting
the process-understanding represented in the biophysical impact models.
Initialised in 2012, ISIMIP is organised in individual modelling rounds. The decision about their design
and the development of the associated simulation protocols has been developed into an iterative
process between stakeholders and users of ISIMIP data, the sectoral coordinators representing
participating modelling teams, the Scientific Advisory Board, and the Cross-Sectoral and Coordination
Team at PIK  (ISIMIP Coordination Team, Sectoral Coordinators, Scientific Advisory Board, 2018).
Since its second round the ISIMIP protocols comprise an 'a' part describing impact model simulations
that cover the historical period forced by observational climate-related and direct human forcings
(evaluation set-up), and a 'b' part dedicated to impact simulations based on simulated climate-related
forcings including future projections. This paper describes the ISIMIP3a simulation framework only
where the DHF described here are also used for the historical simulations within ISIMIP3b. Compared
to ISIMIP2a the evaluation set-up based on observational forcing data has been extended to now
include additional years up to 2021 and sensitivity experiments using high resolution historical climate
forcing data to quantify associated improvements of impact simulations (see section **3.1**). Besides, the
set of historical observation-based direct human forcings has been updated compared to previous
ISIMIP simulation rounds (see **Table 1**). For the first time, and closely connected to the evaluation set-
up, ISIMIP3a now also includes an 'impact attribution' scenario set-up designed to address the question
"To what degree have observed changes in the climate-related systems contributed to observed
changes in natural, human or managed systems compared to direct human influences?" Here, changes
in climate-related systems mean climate change itself, changes in atmospheric $CO_2$ and $CH_4$
concentration, and sea level changes. The attribution question can both refer to the impacts of individual
events (e.g. to what extent has long-term climate change contributed to the observed extent of a specific
river flood?) and to long-term changes (e.g. to what extent have long-term climate change and
increasing $CO_2$ fertilisation contributed to an observed change in crop yields?). The IPCC AR5 (Cramer
et al., 2014) and AR6 ((O'Neill et al., 2022; Hope et al., 2022) have established a framework for impact
attribution according to which an 'observed impact of climate change or change in any other climate-
related system' is defined as the difference between the observed state of the human, natural or
managed system and a counterfactual baseline that characterises the system's behaviour in the
absence of changes in the climate-related systems. This counterfactual baseline may be stationary or
vary in response to direct human influences such as changes in land use patterns, agricultural or water
management or population distribution and economic development affecting exposure and vulnerability

to weather-related hazards. While the definition is established for about a decade at least, the number of studies addressing impact attribution based on this basic definition is still relatively small compared to the number of studies addressing climate attribution, i.e. the question to what degree anthropogenic emissions of climate forcers, in particular greenhouse gases, have induced changes in the climate-related systems. While climate attribution is mainly confronted by the challenge of separating the anthropogenically forced changes from the internal variability of the climate-related systems, the focus of climate impact attribution is on separating the impacts of observed changes in these climate-related systems from the effects of other direct (human) drivers of changes in the considered natural, human or managed systems. 'Observed changes in the climate-related systems' does not necessarily imply 'changes induced by anthropogenic climate forcing', but only means 'any long-term trend' in line with the IPCC definition of climate change (see Glossary of the AR5 (IPCC, 2014) and AR6 (Matthews et al., 2021)).

Impact attribution studies usually face the problem that the counterfactual baseline assuming no long-term changes in the climate-related systems cannot be observed (see (Hansen et al., 2016) for examples). However, impact models such as the ones participating in ISIMIP are well suited to simulate this baseline. As the impact models usually account not only for the changes in climate or the climate-related systems but also for direct human forcings such as land use and irrigation changes, changes in water and agricultural management, population distributions etc. (see **Table 1** for a comprehensive list of direct human forcings provided within ISIMIP3a) they are ideal tools to address the attribution question: In line with the IPCC definition it requires the comparison of a factual simulation based on the observed variations in the climate-related and direct human drivers to a counterfactual simulations where only the climate-related forcings are replaced by counterfactual versions where long-term trends have been removed. While the factual simulations correspond to the evaluation runs within ISIMIP3a (see section **2.1**), the protocol now also includes the counterfactual simulations based on the newly generated counterfactual data sets derived from observational data of climate and coastal water levels (see sections **2.2** for the associated concept and scenario design and **Table 3** for a comprehensive list of the counterfactual climate and sea level forcing data that are described in more detail in section **3.1** and **3.3,** respectively). To allow for an attribution of 'observed changes in natural, human, and managed systems' in contrast to an attribution of simulated changes it has to be demonstrated that the processes represented in the impact model can explain the observed changes in the affected system, i.e. it has to be shown that the model forced by observed changes in the climate-related systems and accounting for the historical development of direct (human) forcings is able to reproduce the observed changes in the affected system. In this way the attribution exercise is closely linked to the ISIMIP3a evaluation exercise. Thereby, models can either explicitly represent known changes in non-climate drivers such as known adjustments of fertiliser input or growing seasons (explicit accounting for non-climate drivers) or implicitly account for their potential contributions by e.g., allowing for non-climate related temporal trends in empirical models as often done in empirical approaches (implicit accounting for non-climate drivers).

While the default attribution experiment in ISIMIP3a is designed for the attribution of observed changes in human, natural, and managed systems to observed change in the climate-related systems in

combination (in the current ISIMIP3a setting this is changes in atmospheric climate forcing in
combination with changes in atmospheric CO2 and CH4 concentrations, see **Table 3**), the protocol
also includes a sensitivity experiments that allow for the quantification of the influence of increasing
CO2 concentrations separately and for an attribution of observed changes in natural, human and
managed systems to historical changes in atmospheric CO2 concentrations only (see section **2.1**).
Here, we consistently define 'an observed impact of a change in any component of the historical forcing
as the difference between the observed state of the system to a counterfactual world where only this
specific component of the forcing has not changed. So the 'observed impact of increasing CO2
concentrations' is approximated by the difference between a full forcing run and a run where CO2
concentrations are held constant. This is different from the 'CO2 only' experiment considered within
TRENDY (Trends in the land carbon cycle, (Sitch et al., 2015, Protocol - TRENDY, 2023)) where the
pure effect of increasing CO2 concentrations on the terrestrial carbon cycle (e.g. net biome production)
is estimated by simulations where the Dynamic Global Vegetation Models (as participating in the biomes
sector of ISIMIP) are forced by the observed increases in CO2 concentrations but a time-invariant "pre-
industrial" climate and land use mask. In the above sense, other ISIMIP3a experiments can also be
considered counterfactual baseline experiments that allow for the attribution of observed changes in
human, natural, or managed systems to changes in the direct human forcings as a whole (DHF set to
zero or fixed at 1901 and 2015 levels) or to changes in individual components such as changes in water
management, irrigation patterns, and riverine influx of nutrients into the ocean (see section **2.1** and
**Table 2**). The attribution to changes in direct human forcings is e.g. similar to the comparison of the full
forcing run within TRENDY to the 'CO2 and climate only' run where climate change and atmospheric
CO2 concentrations are prescribed according to observations but land use changes are held constant
to quantify the contribution of this direct human forcing to observed changes in the carbon cycle for the
annual report of the Global Carbon Project (e.g. (Friedlingstein et al., 2022)). However, in this paper
the term 'impact attribution' is used as a short form of 'attribution of observed changes in natural, human
and managed systems to observed changes in the climate-related systems' which is the focus of the
ISIMIP3a experiments. In other cases the driver to which the changes are attributed is explicitly named.
In addition to ISIMIP3a, there are other model intercomparison projects that address different kinds of
attribution questions such as Land Use Model Intercomparison Project (LUMIP, (Lawrence et al., 2016))
and Detection and Attribution Model Intercomparison Project (DAMIP, (Gillett et al., 2016)) embedded
into the sixth phase of the Coupled Model Intercomparison Project (CMIP6). While the phase 2 LUMIP
experiments include historical climate model simulations to quantify the contribution of historical land
use changes to observed climate change, the AMIP protocol include a counterfactual 'no anthropogenic
climate forcing' baseline to attribute observed changes in climate to anthropogenic climate forcings.
The development of the protocol was coordinated by the ISIMIP-Cross-Sectoral Science Team (CSST)
at the Potsdam Institute for Climate Impact Research (PIK) and involved the sectoral coordinators,
participating modelling teams, and the Scientific Advisory Board. The process was initiated by a
proposal for the main research questions to be addressed and an associated scenario set-up
accounting for suggestions collected in a stakeholder engagement process (Lejeune et al., 2018).
Following ISIMIP's mission and implementation document (ISIMIP Coordination Team, Sectoral
Coordinators, Scientific Advisory Board, 2018), the basic proposal was approved by the ISIMIP strategy
group at the cross-sectoral ISIMIP workshop in Potsdam, September 2018 (Outcomes of the ISIMIP
Strategy Group Meeting, 2023). Thereby the CSST and the sectoral coordinators were tasked to
translate the decisions into a cross-sectorally consistent simulation protocol and to generate, pre-
process or collect the required climate-related and direct human forcing data. The provided forcing data
sets (e.g. the climate variables or components of atmospheric composition or types of land use) is very
much demand driven. The data we describe here represent a core set that is sufficient for the range of
models participating so far (see ISIMIP output data table (ISIMIP Output Data Table, 2023) that also
provides information about the input data used by the individual models) but may be extended if there
were further demands. This paper presents the results of this process and the motivation and reasoning
behind the individual steps for ISIMIP3a, while a follow-up paper will provide the same information for
ISIMIP3b dedicated to impact projections based on climate model simulations(Frieler, submitted 2023).
It provides the point of reference for modelling teams interested in participating in ISIMIP3a but also for
users of the impact simulation data, which become freely accessible according to the ISIMIP terms of
use (ISIMIP terms of use, 2023). The paper is accompanied by a simulation protocol (ISIMIP3
simulation protocol, 2023) providing all technical details such as file and variable naming conventions
and sector-specific lists of output variables to be reported by the participating modelling teams. The
ISIMIP3 simulation round was officially started on 21st February 2020[1] with the release of the
associated protocol. Since then, the protocol has already received some updates through the addition
of output variables, correction of errors, and inclusion of new sectors. This paper refers to the protocol
version of 14th January 2023. However, the protocol may still receive updates similar to the ones
mentioned above. Impact modellers interested in contributing to ISIMIP should therefore refer to
(ISIMIP3 simulation protocol, 2023) for the most up to date version for planned impact model
simulations. The protocol landing page (protocol.isimip.org) includes a unique version identifier (the
commit hash) that links to the latest protocol version on github for traceability.

In the second round of ISIMIP the observation-based model evaluation part (ISIMIP2a) was temporally
separate from the climate model-based second part (ISIMIP2b, (Frieler et al., 2017). This has led to
inconsistencies in the models and model versions contributing to ISIMIP2a and ISIMIP2b. Also, not all
models providing future projections within ISIMIP2b also provided model evaluation runs for ISIMIP2a.
To avoid this problem and ensure that each model's set of future projections is accompanied by
associated historical simulations allowing for model evaluation, in the third simulation round (ISIMIP3),
the ISMIP3a and ISIMIP3b protocols were released together and participating in ISIMIP3 means
contributing to ISIMIP3a and ISIMIP3b using the same impact model versions.

In the following section **2** of this paper, we provide the comprehensive list of all ISIMIP3a model
evaluation and sensitivity experiments (see **Table 2** within section **2.1**) and the counterfactual 'no
climate change' experiments (see **Table 4** within section **2.2**) describe the rationale behind the scenario

---

[1] announced via email to the ISIMIP mailing list from 21st February 2020

set-ups. Detailed description of the climate-related forcing data sets (see CRF section of **Table 1** in
section **2.1** and **Table 3** in section **2.2**) are provided in the third section: atmospheric climate data (see
section **3.1)**; tropical cyclone data (see section **3.2**); coastal water levels (see section **3.3)**, and the
ocean data (see section **3.4**). Section **4** presents the ISIMIP3a direct human forcing data sets (see DHF
section of **Table 1**), comprising population data (see section **4.1**), gross domestic product (see section
**4.2**), land use and irrigation patterns (see section **4.3**), fertiliser inputs (see section **4.4**), land
transformations (see section **4.5**), nitrogen deposition (see section **4.6**), crop calendar (see section
**4.7**), dams and reservoirs (see section **4.8**), fishing intensities (see section **4.9**), regional forest
management (see section **4.10**), and desalination (see section **4.11**).

**2 Experiments and underlying rationale**

ISIMIP3a includes a core ('default') set of experiments that are specified by a specific set of underlying
climate-related forcings and direct human forcings that have to be indicated in the file names when
submitting simulation data to the ISIMIP repository. In the following we first introduce these default
experiments by defining the combination of both types of forcing data sets. In the subheadings naming
the experiments the associated CRF and DHF specifiers to be used in the file names are indicated in
brackets where the third sensitivity specifier is set to 'default' (CRF specifier + DHF specifier, default).
The different combinations of the default sets of ISIMIP3a CRFs ('obsclim', 'counterclim') and DHFs
('histsoc', '2015soc', '1901soc', '1850soc', 'nat') are sketched in **Figure 1** and defined in more detail
below (see **Table 1** for the default 'obsclim' CRF and the default DHFs and **Table 3** for the 'counterclim'
CRF). Some of the forcing data sets are mandatory: i.e. if impact models account for the forcing, the
specified dataset must be used; if an alternative input data set is used instead, the run cannot be
considered an ISIMIP simulation. We also provide 'optional' forcing data that could be used but are not
'mandatory' in the above sense (see second column of **Table 1** and **Table 3**). In addition, the protocol
includes a set of sensitivity experiments that are described as deviations from the default runs and
labelled by the baseline CRF and DHF settings and the third specifier then indicating the deviation from
this default setting instead of being set to 'default'. The ISIMIP3a sensitivity runs include experiments
with high-resolution climate forcing ('30arcsec', '90arcsec', '300arcsec', or '1800arcsec'), fixed levels
of atmospheric $CO_2$ concentrations ('1901co2'), a scenario assuming no water management
('nowatermgt'), simulations excluding the occurrence of wildfires ('nofire'), keeping irrigation patterns at
1901 levels ('1901irr'), and assuming fixed 1955 riverine inputs of freshwater and nutrients into the
ocean ('1955-riverine-input') (see **Table 2**). **Table 2** and **Table 4** providing the comprehensive list of all
'obsclim' and 'counterclim'-based experiments, respectively, also indicate the priority of the experiments
where '1st priority' means that modellers should focus on this set of experiments if their capacities were
limited and they wanted to limit the set of experiments. However, this is just an indication trying to ensure
the generation of a small set of experiments that is covered by as many impact models as possible. If
an impact modeller can only do part of the first priority set-up or has to start from second priority
simulations these fragmented data sets can also be submitted to the ISIMIP3a repository.

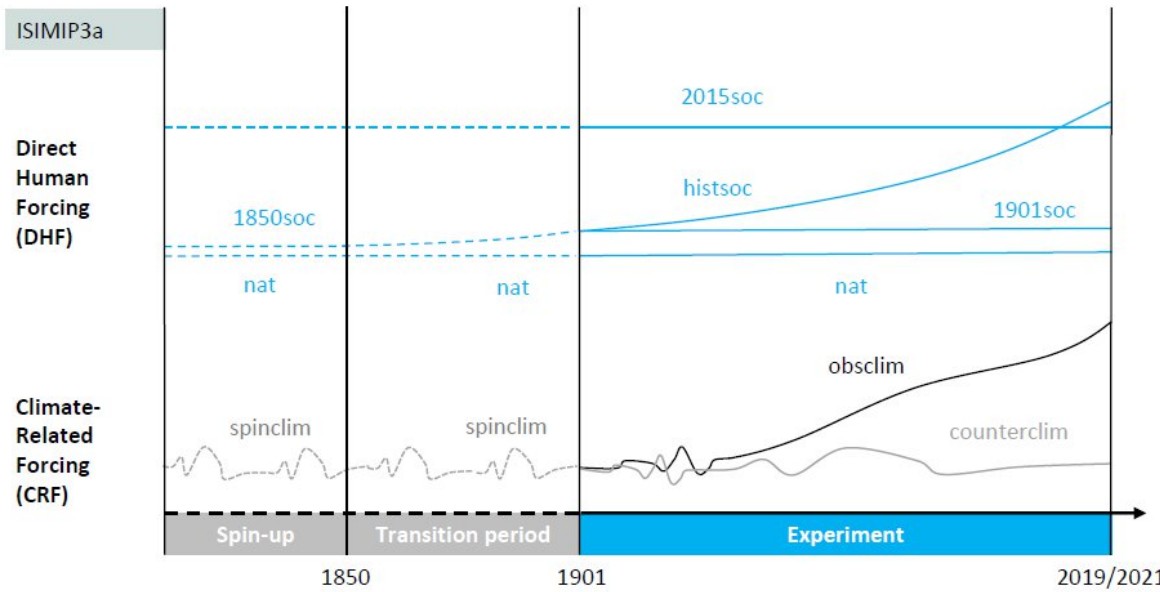

**Figure 1: ISIMIP3a scenario design:** Illustration of the default ISIMIP3a forcing data sets. Each experiment is
defined by a combination of a CRF data set with a DHF data set. The considered combinations are listed in **Table**
**2** and **Table 4** and the underlying rationale is described in section **2.1** (evaluation runs based on 'obsclim' defined
in **Table 1**) and section **2.2** (attribution runs based on 'counterclim' defined in **Table 3**). **Table 1** also lists all data
sets defining the 'histsoc' DHF. Solid lines indicate the part of the experiments that should be reported while the
dashed lines illustrate the different spin-up procedures for the models that require a spin-up. Note that the oceanic
climate-related forcing for the *marine ecosystems and fisheries* sector is only available for 'obsclim' and the period
1961-2010, i.e. the actual experiments only start from the year 1961. The associated spin-up procedure and the
simulations set-up for a transition period are not illustrated in the Figure but described below for the 'obsclim +
histsoc, default', 'obsclim + nat, default', 'obsclim + histsoc, 60arcmin', and 'obsclim + nat, 60arcmin' experiments
considered in this sector.

**2.1 Model evaluation and sensitivity experiments based on observed CRFs ('obsclim')**
The experiments described in this section are all based on observational (factual) climate data, coastal
water levels, and atmospheric CO2 as well as CH4 concentrations including observed trends. The only
exception are the sensitivity experiments where CO2 concentrations are fixed at 1901 levels
('1901co2'). However, as these experiments only deviate in this one aspect from the factual CRF they
are also described by the 'obsclim' CRF specifier but the '1901co2' sensitivity specifier to indicate the
deviation. So all experiments described in this section share the common 'obsclim' CRF specifier in the
file names. In contrast, all experiments described in section 2.2 can be identified by the 'counterclim'
specifier in the names of the output files containing the impact model simulations.

**2.1.1 Default evaluation experiments based on observed CRFs ('obsclim')**
In this first part of section **2.1** we describe the default ISIMIP3a experiments (sensitivity specifier in the
file names set to 'default') that are based on the standard observed climate-related forcings ('obsclim',
see CRF part of **Table 1**) in combination with different assumptions regarding direct human forcings
('histsoc', '2015soc', '1901soc', and 'nat') illustrated in **Figure 1**.

**Standard evaluation experiment (obsclim + histsoc; default).** The first set of observation-based
simulations is dedicated to impact model evaluation, i.e., to test our ability to reproduce and explain
observed long-term changes or variations in impact indicators such as crop yields, river discharge,
changes in natural vegetation carbon, vegetation types, and peatland moisture conditions. To this end,
we provide the climate-related ('obsclim'), direct human ('histsoc'), and static geographical forcings
listed in **Table 1.** They are described in more detail in sections **3** and **4**.
For impact model simulations that require a spin-up to e.g. balance carbon stocks, 100 years of climate
data ('spinclim') are provided that represent stable 1900 climate conditions. The spinclim data is
equivalent to the first 100 years of the counterfactual climate data that are described in section **3.1**. If
more than 100 years of spin-up are needed, the spinclim data can be repeated as often as needed. For
the spin-up, $CO_2$ concentrations and direct human forcing should be kept constant at 1850 levels. To
get to the historical reporting period starting in 1901, modellers should simulate a transition period from
1850 to 1900 using spinclim climate data and the observed increase in $CO_2$ concentrations and
historical changes in socioeconomic forcings (from 1850-1900).
The temporal coverage of the evaluation experiment is limited to 1961-2010 in the *marine ecosystems*
*and fisheries* sector due to the availability of reanalysis-based oceanic forcing data (Liu et al., 2021).
As spin-up + transition period for the 'obsclim + histsoc, default' experiments starting in 1961 the models
should be run through six cycles of 1961-1980 '1955-riverine-input' CRFs (120 years, see **Table 1**)
assuming reconstructed fishing efforts from 1861-1960 and constant 1861 levels before during 1841-
1860 (see **Table 1** and **Figure 3** in section **4.9**). If more years of spin-up are required, additional cycles
of the 1961-1980 '1955-riverine-input' CRFs should be added, assuming constant 1861 fishing efforts.
**Table 1: Climate-related, direct human, and static geographic forcing data provided for the model**
**evaluation and sensitivity experiments within ISIMIP3a.** The CRFs are grouped according to the definition of
the default 'obsclim' CRF (30 arcmin for the atmospheric data and 15 arcmin for the oceanic data), the higher
resolution '30arcsec', '90arcsec', '300arcsec', '1800arcsec' atmospheric CRF, the lower resolution '60arcmin'
oceanic CRF, and the '1955-riverine-input' oceanic CRF for the sensitivity experiments. The listed set of DHFs
defines the 'histsoc' set-up.

| Forcing | Status | Source, description |
|---|---|---|
| Climate-Related Forcings ('obsclim') | | |
| Atmospheric forcings | | |
| Standard observation-based atmospheric climate forcing | mandatory | GSWP3-W5E5, 20CRv3-W5E5, 20CRv3-ERA5, 20CRv3, see section **3.1** |
| Local atmospheric climate forcing for lake locations | mandatory | Atmospheric data extracted from the data sets above for 72 lakes that have been identified within the *lake* sector as locations (grid cells of the ISIMIP 0.5° grid) where models can be calibrated based on observed temperature profiles |

| | | and hypsometry (Golub et al., 2022, https://www.isimip.org/gettingstarted/input-data-bias-adjustment/isimip3-local-lake-sites/). |
|---|---|---|
| Tropical cyclone tracks, as well aswind and precipitation fields | mandatory | Tracks from IBTrACS database (period 1950-2021; (Knapp et al., 2010). Wind and precipitation fields calculated by Holland(Holland, 1980, 2008), see section **3.2** |
| Lightning | mandatory | Satellite-based (1995-2014) climatology of monthly flash rates (number of strokes km-2 d-1 on 0.5° grid (Cecil, 2006) |
| Oceanic forcings | | |
| Standard observation-based oceanic forcing data | mandatory | GFDL MOM6/COBALTv2 simulations driven by reanalysis-based atmospheric forcing (Liu et al., 2021) see section **3.4** |
| Regional oceanic climate forcing for regional *marine ecosystems* and *fisheries* sector | mandatory | Extraction from data set above for 21 regional marine ecosystems associated with the interests identified by the modelling groups (https://www.isimip.org/gettingstarted/input-data-bias-adjustment/isimip3-ocean-regions/). The extraction has been done for individual layers (ocean surface or bottom) and a subset of the variables that have been integrated along the ocean column (see Table 8). |
| Coastal water levels | | |
| Coastal water levels | mandatory | Hourly coastal water levels with long-term trends, see section **3.3** |
| Atmospheric composition | | |
| Atmospheric $CO_2$ concentration | mandatory | 1850-2005: (Meinshausen et al., 2011); 2006-2021: Global annual CO2 from NOAA Global Monthly Mean $CO_2$; (Lan et al., 2023; Büchner and Reyer, 2022) |
| Atmospheric $CH_4$ concentration | mandatory | 1850-2014: (Meinshausen et al., 2017); 2015-2021: (Büchner and Reyer, 2022; Lan et al., 2023) |
| Climate-Related Forcings for sensitivity experiments (30arcsec, 90arcsec, 300arcsec, 1800arcsec, | | |

| 60arcmin, and 1955-riverine-input), identical to 'obsclim' except for: | | |
|---|---|---|
| **Atmospheric forcings** (30arcsec, 90arcsec, 300arcsec, 1800arcsec) | | |
| High resolution observation-based atmospheric forcing data | mandatory | see section **3.1** for a description of the CHELSA method applied to downscale the W5E5 observation-based atmospheric data to 30''. The data is then upscaled to 90'' (~3 km), 300'' (~10 km) and 1800'' = 0.5° (~60 km) to provide the forcings for additional sensitivity experiments. |
| **Oceanic forcings** (60arcmin) | | |
| Low resolution observation-based oceanic forcing data | mandatory | GFDL MOM6/COBALTv2 simulations (1961 - 2010) driven by reanalysis-based atmospheric forcing (Liu et al., 2021) upscaled to 1°, see section **3.4** |
| **Oceanic forcings** (1955-riverine-input) | | |
| Observation-based oceanic forcing data but assuming climatological 1951 to 1958 levels of riverine input | mandatory | GFDL MOM6/COBALTv2 simulations (1961 - 2010) driven by reanalysis-based atmospheric forcing (Liu et al., 2021), but fixed climatological 1951 to 1958 levels of freshwater and nutrients inputs, see section **3.4** |
| **Direct Human Forcing ('histsoc')** | | |
| Population data | mandatory | see section **4.1** |
| GDP data | mandatory | see section **4.2** |
| Land use and irrigation | mandatory | HYDE-based irrigated and rainfed cropland downscaled to up to 15 crops, managed pasture and grassland, and urban areas, see section **4.3** |
| N-fertiliser inputs | mandatory | see section **4.4** |
| Wood harvest | optional | Historical annual country-level wood harvesting data based on the LUH v2 Harmonization Data Set (del Valle et al., 2022; Hurtt et al., 2011, 2020, Land use harmonization, 2023), see section **4.5** |
| Land transformation | mandatory | Historical annual land-use transformation data, based on the LUH v2 Harmonization Data Set (Hurtt et al., 2011, 2020, |

| | | Land use harmonization, 2023), see section **4.5** |
|---|---|---|
| N-deposition | optional | (Yang and Tian, 2020; Tian et al., 2018) |
| Crop calendar | optional | Observation-based representation of recent average planting and maturity dates not accounting for changes over time (Jägermeyr et al., 2021a), see section **4.7** |
| Dams and reservoirs | optional | see section **4.8** |
| Lake and reservoir surface area | optional | Total lake and reservoir area fractions (percentage of grid cell) calculated from the HydroLAKES v1.0 (Messager et al., 2016) and GRanDv1.3 databases (Lehner et al., 2011b) mapped to 0.5 degrees resolution. Areas increase with time because of the increasing number of reservoirs documented in GRanDv1.3. Reservoirs from 2017 onwards are kept constant. This data set differs from the lake surface areas provided as static geographic forcing (see below) which describe the surface area of one representative lake per grid cell and does not change over time. |
| Water abstraction | optional | For modelling groups that do not have their own representation, we provide files containing the multi-model mean of domestic and industrial water withdrawal and consumption generated by the WaterGAP, PCR-GLOBWB, and H08 models (1850-2021). This data is based on ISIMIP2a 'varsoc' simulations for 1901-2005 and extended by SSP2-based simulations from the Water Futures and Solutions project up to 2021 (Wada et al., 2016b). Years before 1901 have been filled with the value for year 1901. |
| Marine fishing effort | mandatory | Observation-based reconstruction of fishing effort spanning 1841-2010  (Rousseau et al., 2022) based on (Rousseau et al., submitted 2023); see section **4.9** The climate-related forcing for the *marine ecosystems and fisheries* sector is only available for 1961-2010, but the spin-up procedure also requires fishing efforts for the earlier years (see description of the procedure for the 'obsclim + histsoc; default' scenario above). |
| Forest management | mandatory | Observed stem numbers, thinning type, planting numbers from and common management practices for 9 forest sites |

| | | in Europe (Reyer et al., 2020b),(Reyer et al., 2023), see section **4.10** |
|---|---|---|
| Static geographic forcing | | |
| Lake volume at different depths | optional | The gridded data set describes the volume at different depths of one hypothetical lake representing the typical characteristics of all real lakes in the grid cell according to the GLOBathy (Khazaei et al., 2022; Messager et al., 2016) and HydroLAKES v1.0 (Khazaei et al., 2022; Messager et al., 2016) datasets (Golub et al., 2022). Each hypsographic curve consists of 11 data pairs. Level refers to the depth of the lake taking the lake bottom as the reference. Volume is the volume at the corresponding level. |
| Lake area at different depths | optional | The gridded data set describes the lake area at different depths of one hypothetical lake representing the typical characteristics of all real lakes in the grid cell according to the GLOBathy (Khazaei et al., 2022; Messager et al., 2016)and HydroLAKES (Khazaei et al., 2022; Messager et al., 2016) datasets (Golub et al., 2022). Each hypsographic curve consists of 11 data pairs. Level refers to the depth of the lake taking the lake bottom as the reference. |
| Lake elevation | optional | The gridded data set provides the elevation above sea level for the representative lakes described above. The information is derived from HydroLAKES v1.0 (Messager et al., 2016). |
| Maximum lake depth | optional | Gridded data set that provides the maximum depth for the representative lakes described above and derived from GLOBathy (Khazaei et al., 2022). We recommend using the area or volume hypsographic curves described above as inputs for your lake model. Use this file only if your lake model does not accept a full hypsographic curve as an input. |
| Lake depth | optional | Gridded data set that provides the mean depth for the representative lakes as calculated from GLOBathy and HydroLAKES v1.0 (Khazaei et al., 2022; Messager et al., 2016). We recommend using the area or volume |

|  |  | hypsographic curves described above as inputs for your lake model. Use this file only if your lake model does not accept a full hypsographic curve as an input. |
|---|---|---|
| Lake volume | optional | Gridded data set of volume (km$^3$) for representative lakes described above as calculated from GLOBathy and HydroLAKES v1.0 (Khazaei et al., 2022; Messager et al., 2016). We recommend using the area or volume hypsographic curves described above as inputs for your lake model. Use this file only if your lake model does not accept a full hypsographic curve as an input. |
| Lake surface area | optional | Gridded data set of surface area for the representative lakes described above as calculated from GLOBathy and HydroLAKES v1.0 (Khazaei et al., 2022; Messager et al., 2016). As opposed to the "Lake and reservoir surface area" listed above under "Direct human forcing", this data set refers to one specific lake associated with each grid cell, and the corresponding surface area does not change over time.<br>We recommend using the area or volume hypsographic curves described above as inputs for your lake model. Use this file only if your lake model does not accept a full hypsographic curve as an input. |
| HydroLAKES ID | optional | HydroLAKES reference to relate HydroLAKES and GLOBathy database fields to the representative lakes described above. This dataset contains IDs of the 41449 representative lakes used in ISIMIP, which are a subset of the about 1.4 million lakes contained in the HydroLAKES and GLOBathy database. |
| HydroLAKES IDs for big lakes | optional | This dataset is analogous to the one above, but only contains IDs of 93 large lakes. It can be used to produce global plots with conspicuous large lakes. To be used together with the file storing the big lakes mask. |
| Big lakes mask | optional | This dataset indicates the 0.5° grid cells actually occupied by each of the 93 large lakes, which can be larger than a single grid cell. It can be used to produce global plots with conspicuous large lakes. To be used together with the big |

| | | lakes IDs in the dataset above. |
|---|---|---|
| Drainage direction map for river routing | optional | Includes for each grid cell a basin number, flow direction, and slope. Source: ISIMIPddm30 (Müller Schmied, 2022) based on DDM30 (Döll and Lehner, 2002) |
| Soil data | optional | Gridded soil characteristics have been generated within the Global Soil Wetness Project (GSWP3) (Dirmeyer et al., 2006; van den Hurk et al., 2016, Global soil wetness project phase 3 — GSWP3 documentation, 2023) and have already been provided within ISIMIP2a.<br><br>Alternatively, we also provide maps of the dominant soil types (i.e., the type covering the largest fraction of the cell of the topmost soil layer) within each ISIMIP grid cell and the dominant soil types on the agricultural land within each ISIMIP grid cell. Both maps were derived from the Harmonized World Soil Database (HWSD Version 1.1, 2009) assuming that soil types are evenly distributed within the ISIMIP grid cells. We have used version 1.12 of the HWSD data at high resolution (30 arcsec). Information about the fraction of agricultural land within each ISIMIP 0.5°×0.5° grid cell was taken from MIRCA2000 (Portmann et al., 2010). If there is no soil information for an ISIMIP grid cell, e.g. due to differing land-sea-masks, the information from neighbouring cells is used. For further details please see GGCMI-HWSD (2023). |
| Land-sea mask | optional | We provide the binary land-sea mask of the W5E5 dataset. It is a conservative land mask where grid cells that in reality cover both land and ocean are counted as ocean. Thus, climate conditions over the land grid cells of this land-sea mask can be safely assumed to represent climate conditions over land rather than a mix of climate conditions over land and ocean. This refers to all climate datasets based on W5E5, i.e. GSWP3-W5E5 and 20CRv3-W5E5 of ISIMIP3a and the ISIMIP3b climate forcing that has been bias-adjusted using W5E5. The mask is also provided in a version without Antarctica. In addition, the generic land-sea mask from ISIMIP2b is provided to be used for global water |

| | | |
|---|---|---|
| | | simulations in ISIMIP3. It marks more grid cells as land than the main mask described above (Lange and Büchner, 2020). |
| Sea floor depth | optional | Grid cell level ocean depth in metres of GFDL-MOM6-COBALT2 data in 0.25 and 1° horizontal resolution |
| Binary country mask | optional | Binary country map on a 0.5° x 0.5° latitude-longitude grid |
| Fractional country mask | optional | Fractional country map on the ISIMIP 0.5° x 0.5° grid. This is the map that has been used to calculate the national data for ISIpedia (isipedia.org) and to e.g. prepare the national population and GDP data provided within ISIMIP3 (see sections **4.1** and **4.2**). |
| Large Marine Ecosystem masks | mandatory | Binary masks available at 0.25°, 0.5°, and 1° resolution (Sherman, 2017). |
| Regional Marine Ecosystem masks | optional | Binary masks describing the 21 ocean regions for the regional modelling activities in the fisheries and marine ecosystems available at 0.25° and 1° resolution. These masks have been used for the ocean forcing data extractions (see CRF part of this table). |


**Fixed 2015 direct human forcing (obsclim + 2015soc; default).** To allow for the quantification of the
effect of historical changes in direct human forcings, ISIMIP3a also contains an experiment where all
direct human forcings are held constant at year 2015 levels. The difference between the evaluation run
described above and this baseline simulation can be considered the impact of changes in direct human
forcings. In this sense the experiment allows for the attribution of observed changes in the natural,
human, and managed systems to changes in DHF after 2015. In addition, the simulated changes in
models' output variables can be considered the 'pure effects of climate-related forcings', conditional on
present-day socio-economic conditions. The experiment is also introduced because not all impact
models can account for varying direct human forcings but rather assume fixed 'present day' conditions.
All modelling teams are asked to do this experiment even if they are able to account for varying direct
human forcings to generate one set of impact simulations that can be integrated across all participating
models from different sectors or where all simulations from one sector can be compared. If a spin-up is
required, it should be based on the 'spinclim' data as described above but fixed 2015 direct human
forcings.

**Fixed 1901 direct human forcing baseline (obsclim + 1901soc; default).** Fixing direct human
forcings at 1901 levels is an alternative approach to quantify i) the effects of direct human forcings when
comparing these baseline simulations to the evaluation run and ii) the 'pure effect of observed change
in climate-related systems', conditional on socio-economic conditions observed before the onset of this
change. As such the experiment is the counterfactual baseline when aiming for the attribution of
observed changes in natural, human, and managed systems to observed changes in direct human
forcings instead of the attribution to observed changes in the climate-related systems based on the
analogous 'counterfactual + histsoc, default' experiment described in section **2.2**. Both experiments
consider changes in direct human forcings or climate-related systems from 1901 levels, respectively.
Because of the low levels of direct human forcings in 1901, this experiment is similar to the sector-
specific 'nat' experiment that includes no direct human forcings whatsoever (see below). However,
while the fully naturalised 'nat' run is suitable for the dynamic vegetation models from the *biomes* sector
that simulate land cover by vegetation on their own, models in other sectors need land cover as an
input. As this information is not available for pristine conditions, we introduce the 1901soc scenario
such that models in the *water* sector can use land cover data approximately representative of 1901
conditions to describe a situation with minor human influences. If a spin-up is required, it should be
based on the 'spinclim' data as described above but fixed 1901 direct human forcings.

**No direct human forcing baseline (obsclim + nat; default).** To estimate the full effect of 2015 levels
of DHF we also introduce a baseline 'nat' experiment that does not consider any DHFs but a natural
state of the world. Then the difference to the 'obsclim + 2015soc, default' experiment can be considered
the effect of 2015 levels of DHF. The comparison to the 'obsclim + histsoc, default' experiment allows
for the attribution of observed changes in the natural, human, and managed systems to historical
changes in the DHF. Trends in the 'obsclim + nat; default' run only represent the impacts historical
changes in the climate-related forcings would have had on an otherwise natural state of the world.
While the '1901soc' conditions may be similar to 'nat' conditions, trends in the 'obsclim + 1901soc;
default' run may not only be induced by historical changes in the CRFs but could also represent lagged
responses to changes in DHFs during the transition period. The 'nat' experiment can also be used to
quantify the natural carbon sequestration potential of natural vegetation without any management or
land-use as an important counterfactual baseline to assess the additionality of carbon sequestration
measures. The 'nat' experiment is sector-specific for the *biomes, peat* and *marine ecosystems and*
*fisheries* sectors. If a spin-up is required in the *biomes* and *peat* sector, it should be based on the
'spinclim' data as described above but assuming no direct human forcings. In the *marine ecosystems*
*and fisheries* sector the spin-up should be based on the '1955 riverine input' CRF as described for
'obsclim + histsoc, default' section but assuming no DHF, i.e. no fishing efforts.

**2.1.2 Sensitivity experiments based on observed CRFs ('obsclim')**
This second part of section **2.1** is dedicated to the different sensitivity experiments described as
deviations from the default cases described in section **2.1.1**. Instead of the 'default' specifier, all
experiments described here are labelled by a sensitivity specifiers indicating their deviation from the
default cases. The experiments listed here are not explicitly depicted in **Figure 1**.

**High and low resolution sensitivity experiments (obsclim + histsoc; 30arcsec, 90arcsec, 300arcsec, 1800arcsec, and 60arcmin).** To test whether high resolution atmospheric climate data improve the climate impact model simulations, we also provide observational atmospheric forcing data at 30'' ('30arcsec'), 90'' ('90arcsec'), and 300'' ('300arcsec') resolution as well as atmospheric forcings at the original 1800'' resolution but derived from the 30'' (~1 km) data ('1800arcsec'). In addition, the oceanic data (original resolution of 0.25°) is upscaled to 1° to also test the sensitivity of the impact simulations to this modification ('60arcmin').

The 30'' atmospheric data (1979-2016) is derived from a topographic downscaling of the observational W5E5 data (resolution of 0.5°) that particularly corrects for systematic effects induced by orographic details not represented in global reanalyses (CHELSA-W5E5, see section **3.1**). The data set comprises daily mean precipitation, daily mean surface downwelling shortwave radiation, daily mean near-surface air temperature, daily maximum near surface air temperature, daily minimum near surface air temperature (see **Table 5**). We additionally provide simple approaches to downscale surface downwelling longwave radiation, near-surface relative humidity, air pressure and near-surface wind speed (see section **3.1**). Given the considerable storage capacities required by daily 1 km x 1 km data and constraints on data handling and download, we also aggregate the CHELSA-W5E5 data to 90'' (~3 km), 300'' (~10 km) and 1800'' = 0.5° (~60 km) to determine which resolution is required to improve the impact model simulations compared to observed impact indicators. The evaluation of these historical sensitivity experiments will inform future downscaling activities for the GCM climate forcing data including future projections. The '1800arcsec' experiment is included as a reference, as the aggregated CHELSA-W5E5 data differ from the standard W5E5 data at the same resolution (see section **3.1**). So far the experiments have been added to the agriculture, lakes, global and regional water, regional forests, terrestrial biodiversity, and labour protocol. However, they may be added to other sectors, too. The inclusion of the experiment is only constrained by the restricted set of variables included in CHELSA-W5E5. We do not provide spin-up data for the experiments. This means that models requiring a spin-up currently cannot perform the experiments. We will work on a solution on demand.

In contrast to the experiment testing the sensitivity of the impact simulations to a higher resolution of the atmospheric CRFs, the associated sensitivity experiment for the *marine ecosystems and fisheries* sector is not based on higher but on lower resolution oceanic data. While the default 'obsclim' oceanic forcing data is derived by interpolating the observation-based historical ocean simulations from a tri-polar 0.25° grid to a regular 0.25° grid (see section **3.4**), the CRFs for the sensitivity experiment are derived by aggregating the default 'obsclim' data to a regular 1.0° grid ('60arcmin'). Evaluating the 1.0° resolution is of interest because this is the resolution of the oceanic forcing data in ISIMIP3b. The low resolution simulations could either start from the end of the simulations of the transition period of the associated higher resolution runs ('obsclim + histsoc; default') or starting conditions could be newly generated by following the 'spin-up + transition' procedure of 'obsclim + histsoc; default' experiment but using the low-resolution '1955-riverine-input' CRF from the years 1961-1980.

**Low resolution sensitivity experiment (obsclim + nat; 60arcmin).** This sensitivity experiment for the *marine ecosystems and fisheries* sector is analogous to the 'obsclim + nat; default' experiment

described further above, but using the lower-resolution oceanic CRF ('60arcmin'). The difference
between this experiment and the 'obsclim + histsoc; 60arcmin' sensitivity experiment can be considered
the effect of the historical changes in DHF as estimated using lower-resolution CRF, and comparison
with the same difference in the default experiments then indicates how the estimate of this effect
depends on the resolution of the oceanic forcing. The simulations could either start from the end of the
simulations of the transition period of the associated higher resolution runs ('obsclim + nat; default') or
starting conditions could be newly generated by following the 'spin-up + transition' procedure of 'obsclim
+ nat, default' experiment but using the low-resolution '1955-riverine-input' CRF from the years 1961-
457  1980.

**CO$_2$ sensitivity experiments (obsclim + histsoc, obsclim + 2015soc, or obsclim + 1901soc;**
**1901co2).** To quantify the pure effect of the historical increase in atmospheric CO$_2$ concentrations on
vegetation leaf gas exchange and follow-on effects on carbon stocks, water use efficiency, vegetation
distribution etc., we introduced three sensitivity experiments where atmospheric CO$_2$ concentrations
are held constant at 1901 levels (= 296.13 ppm) in contrast to the default 'obsclim + histsoc', 'obsclim
+ 2015soc', or 'obsclim + 1901soc' experiments, respectively, where atmospheric CO$_2$ concentrations
are assumed to increase according to observations. The effect is known as CO$_2$ fertilisation through an
increase of the photosynthesis rate of plants and limited leaf transpiration (increase in water use
efficiency) enabling a more efficient uptake of carbon by the plants. Comparing the 'obsclim + histsoc,
default' experiment to the 'obsclim + histsoc, 1901soc' experiment can be considered as attributing
historical changes in natural, human, and managed systems to historical changes in CO2
concentrations as a single component of the changes in climate-related systems. The experiment is
included into the protocols of the *agriculture, terrestrial biodiversity, biomes, fire, lakes (global and*
*local), permafrost, peat and water (global and regional)* sector. A potentially required spin-up should be
identical to the spin-up for the associated default experiments using the transition period 1850-1900 to
reach the 1901 CO$_2$ level.
**Water management sensitivity experiment (obsclim + histsoc, obsclim + 2015soc; nowatermgt).**
In this "no water management" experiment, models are run assuming no irrigation, no human water
abstraction, no dams or reservoirs, and no seawater desalination, while other direct human forcings
such as land use changes are considered according to 'histsoc' or '2015soc'. By comparison to the
default experiments, the simulations allow for a quantification of the pure effects of dedicated water
management measures on, e.g., discharge. When comparing 'obsclim + histsoc, nowatermgt' to
'obsclim + histsoc, default' this can be considered attributing observed changes in natural, human, or
managed systems to (changes in) water management. The sensitivity experiment has been introduced
into the *global and regional water* sector protocols. If a spin-up is required, it should be done similar to
the spin-up for the associated default experiments but assuming "no water management".
**Irrigation sensitivity experiment (obsclim + histsoc, 1901irr).** In this "no irrigation expansion"
experiment, models are run assuming irrigation extent and irrigation water use efficiencies fixed at the
year 1901, while other direct human forcings such as land use changes and water management
categories are considered according to 'histsoc' or '2015soc'. By comparison to the default experiments,
the simulations allow for a quantification of the pure effects of historical irrigation expansion (i.e. the
attribution of historical changes in natural, human, or managed systems to changes in irrigation
compared to 1091). The sensitivity experiment has been introduced into the *global water and biome*
sector protocols. If a spin-up is required, it should be done similar to the spin-up for the associated
default experiments but assuming "no irrigation expansion". This experiment is designed such that its
outcomes are comparable to those of the Irrigation Impacts Model Intercomparison Project (IRRMIP;
https://hydr.vub.be/projects/irrmip), in which Earth System Models simulate irrigation influences on the
Earth system.

**No-fire sensitivity experiment (obsclim + histsoc; nofire).** In this 'nofire' experiment, fire is switched
off in the model simulations. In comparison to the default 'obsclim + histsoc' simulations, the historical
effects of fires on, e.g., carbon fluxes and vegetation distributions can be determined. The sensitivity
experiment has been introduced into the *fire, biomes, permafrost, and peat* protocols. The required
spin-up should be done similar to the spin-up for the associated default experiments but assuming no
fire activities.

**Fixed 1955 riverine input into the ocean sensitivity experiment (obsclim + histsoc; obsclim +**
**nat; 1955-riverine-input).** In this '1955-riverine-input' experiment, riverine input into the ocean (amount
of freshwater and nutrients) is held constant at 1955 levels. In comparison to the default 'obsclim +
histsoc' simulation, the experiment allows for the quantification of the impacts of historical climate-
induced variations in freshwater influx in combination with the climate and directly human induced
changes in nutrient inputs (attribution of observed changes in marine ecosystems and fisheries to long
term changes in riverine freshwater and nutrient inputs). The riverine inputs in the 'obsclim + nat; 1955-
riverine-input' experiment are identical to the ones in the 'obsclim + histsoc; 1955-riverine-input', i.e.
the riverine inputs also account for the human contribution to the nutrient influx due to land use changes
and fertiliser inputs and are not 'naturalized'. Instead the 'nat' specifier in the marine ecosystems and
fisheries sector only means 'no fishing efforts'. Thus, the comparison to the naturalised default
experiment (obsclim + nat; default) not accounting for any fishing efforts  to the 'obsclim + nat; 1955-
riverine-input' experiment allows for a quantification of the contribution of climate-induced changes in
freshwater-influx to the overall impacts of climate change in combination with the contribution of the
effect of the human contribution to nutrient inputs at 1955 levels. The sensitivity experiment has been
introduced into the marine ecosystems and fisheries protocol. A potentially required spin-up should be
done similar to the spin-up for the associated default experiments but assuming riverine inputs fixed at
1955 levels.

**Table 2: ISIMIP3a evaluation and sensitivity experiments**

| Experiment | Short description | Period: Historical 1901-2019 |
|---|---|---|
| **model evaluation** <br><br> histsoc <br><br> **1st priority** | **CRF:** Observed climate change, $CO_2$ and $CH_4$ levels, and coastal water levels | **obsclim** |
| | **DHF:** Varying direct human influences according to observations | **histsoc** |
| **model evaluation** <br><br> 2015soc <br><br> **1st priority** | **CRF:** Observed climate change, $CO_2$ and $CH_4$ levels, and coastal water levels | **obsclim** |
| | **DHF:** Fixed 2015 levels of direct human forcing for the entire time period | **2015soc** |
| **model evaluation** <br><br> 1901soc <br><br> **2nd priority** | **CRF:** Observed climate change, $CO_2$ and $CH_4$ levels, and coastal water levels | **obsclim** |
| | **DHF:** Fixed 1901 levels of direct human forcing for the entire time period | **1901soc** |
| **model evaluation** <br><br> nat <br><br> **2nd priority** | **CRF:** Observed climate change, $CO_2$ and $CH_4$ levels, and coastal water levels | **obsclim** |
| | **DHF:** No direct human influences | **nat** |
| **$CO_2$ sensitivity** <br><br> histsoc <br><br> **2nd priority** | **CRF:** Observed climate change, $CH_4$ concentrations and coastal water levels, fixed $CO_2$ concentration at 1901 level | **obsclim** <br><br> **Sensitivity experiment: 1901co2** |
| | **DHF:** Varying direct human influences according to observations | **histsoc** |
| **$CO_2$ sensitivity** <br><br> 2015soc | **CF:** Observed climate change, $CH_4$ concentrations and coastal water levels, fixed $CO_2$ concentration at 1901 level | **obsclim** <br><br> **Sensitivity experiment: 1901co2** |

| 2nd priority | DHF: Fixed 2015 levels of direct human forcing for the entire time period | 2015soc |
|---|---|---|
| CO₂ sensitivity<br><br>1901soc<br><br>2nd priority | CRF: Observed climate change, CH₄ concentrations and coastal water levels, fixed CO₂ concentration at 1901 level | obsclim<br><br>Sensitivity experiment: 1901co2 |
| | DHF: Fixed 1901 levels of direct human forcing for the entire time period | 1901soc |
| Water management sensitivity<br><br>histsoc<br><br>2nd priority | CRF: Observed climate change, coastal water levels, and CO₂ and CH₄ concentrations | obsclim |
| | DHF: No accounting for water management but representation of other direct human influences such as land use changes according to "histsoc" | histsoc<br><br>Sensitivity experiment: nowatermgt |
| Water management sensitivity<br><br>2015soc<br><br>2nd priority | CRF: Observed climate change, coastal water levels, and CO₂ and CH₄ concentrations | obsclim |
| | DHF: No accounting for water management but representation of other direct human influences such as land use patterns according to "2015soc" | 2015soc<br><br>Sensitivity experiment: nowatermgt |
| Irrigation sensitivity<br><br>histsoc<br><br>2nd priority | CRF: Observed climate change, coastal water levels, and CO₂ and CH₄ concentrations | obsclim |
| | DHF: Fixed year-1901 irrigation areas and water use efficiencies but representation of other direct human influences such as land use changes according to "histsoc" | histsoc<br><br>Sensitivity experiment: 1901irr |
| No-fire sensitivity | CRF: Observed climate change, coastal water levels, CO₂ and CH₄concentrations | obsclim |

| | | |
|---|---|---|
| histsoc<br><br>**2nd priority** | **DHF:** Varying direct human influences according to observations | **histsoc**<br><br>**Sensitivity experiment: nofire** |
| **Riverine influx sensitivity**<br><br>histsoc<br><br>**2nd priority** | **CRF:** Observation-based oceanic forcing data, but with constant riverine nutrient and freshwater influx. | **obsclim**<br><br>**Sensitivity experiment: 1955-riverine-input** |
| | **DHF:** Varying direct human influences according to observations | **histsoc** |
| **Riverine influx sensitivity**<br><br>nat<br><br>**2nd priority** | **CRF:** Observation-based oceanic forcing data, but with constant riverine nutrient and freshwater influx. | **obsclim**<br><br>**Sensitivity experiment: 1955-riverine-input** |
| | **DHF:** No direct human influences | **nat** |
| **H i g h - r e s o l u t i o n sensitivity, 1km**<br><br>histsoc<br><br>**2nd priority** | **CRF:** Observed high-resolution climate forcing (30''), coastal water levels, and $CO_2$ and $CH_4$ concentrations. For this experiment only 1979-2016 is covered | **obsclim**<br><br>**Sensitivity experiment: 30arcsec** |
| | **DHF:** Varying direct human influences according to observations | **histsoc** |
| **H i g h - r e s o l u t i o n sensitivity, 3km**<br><br>histsoc<br><br>**2nd priority** | **CRF:** Observed high-resolution climate forcing (90''), coastal water levels, and $CO_2$ and $CH_4$ concentrations. For this experiment only 1979-2016 is covered | **obsclim**<br><br>**Sensitivity experiment: 90arcsec** |
| | **DHF:** Varying direct human influences according to observations | **histsoc** |
| **H i g h - r e s o l u t i o n sensitivity, 12km** | **CRF:** Observed high-resolution climate forcing (360''), coastal water levels, and $CO_2$ and $CH_4$ concentrations. For this experiment only 1979- | **obsclim**<br><br>**Sensitivity** |

| histsoc 2nd priority | 2016 is covered | experiment: 360arcsec |
|---|---|---|
| | **DHF:** Varying direct human influences according to observations | **histsoc** |
| **H i g h - r e s o l u t i o n sensitivity, 60km** histsoc **2nd priority** | **CRF:** Observed climate forcings aggregated from high-resolution data, coastal water levels, $CO_2$ and $CH_4$ concentrations. For this experiment only 1979-2016 is covered | **obsclim** **Sensitivity experiment: 1800arcsec** |
| | **DHF:** Varying direct human influences according to observations | **histsoc** |
| **L o w - r e s o l u t i o n sensitivity, 1° in the ocean** histsoc **2nd priority** | **CRF:** Observation-based oceanic forcing data | **obsclim** **Sensitivity experiment: 60arcmin** |
| | **DHF:** Varying direct human influences according to observations | **histsoc** |
| **L o w - r e s o l u t i o n sensitivity, 1° in the ocean** nat **2nd priority** | **CRF:** Observation-based oceanic forcing data | **obsclim** **Sensitivity experiment: 60arcmin** |
| | **DHF:** No direct human influences | **nat** |


**2.2 Counterfactual baseline simulations for impact attribution ('counterclim')**


The second set of impact model simulations within ISIMIP3a is dedicated to the attribution of historical
changes in natural, managed, and human systems to long-term changes in climate-related systems,
i.e. the atmosphere, ocean and cryosphere as physical or chemical systems (see section **1**). In
ISIMIP3a, we address attribution to changes in the climate system itself, e.g., trends in atmospheric
temperature and precipitation, and changes in coastal water levels, and atmospheric $CO_2$
concentrations. The provided counterfactual forcing data comprises daily atmospheric climate derived
from the ISIMIP observational climate datasets (see section **3.1**); daily counterfactual coastal water
levels derived from the ISIMIP historical coastal water level dataset (see section **3.3**); and constant
1901 atmospheric $CO_2$ and $CH_4$ concentrations (see **Table 3**). So far, we do not address attribution to
long-term changes in i) the ocean (e.g. temperature or ocean acidification changes), ii) the cryosphere
(e.g. glacier mass loss), and iii) tropical cyclone characteristics (e.g. trends in associated heavy
precipitation or wind speeds) other than the effects mediated through sea level rise. **Table 3** lists the
climate-related forcings defining the 'counterclim' experiments. The 'counterclim' climate-related
forcings are combined with the observed direct human forcing to facilitate the attribution experiments
listed in **Table 4** and explained below.
**Table 3: ISIMIP3a counterfactual climate-related forcings ('counterclim')**

| Forcing | Status | Source, description |
|---|---|---|
| Climate-related forcings (counterclim) | | |
| Atmospheric forcings | | |
| Counterfactual 'no-climate change' atmospheric climate forcing | mandatory | Detrended versions of the GSWP3-W5E5, 20CRv3-W5E5, 20CRv3-ERA5, 20CRv3 data sets derived by the Attrici method, see section **3.1** |
| Local atmospheric climate forcing for lake location | mandatory | Atmospheric data extracted from the data sets above for 72 lakes that have been identified within the *lake* sector as locations (grid cells of the ISIMIP 0.5° grid) where models can be calibrated based on observed temperature profiles and hypsometry (depth and area). |
| Tropical cyclone tracks and windfields | mandatory | We do not provide 'no climate change' TC tracks and windfields but the original tracks from the IBTrACS database (Knapp et al., 2010); period 1841-2021) windfields calculated by Holland model (Holland, 2008, 1980) should be used in combination with the counterfactual water levels to estimate the impacts of sea level rise on TC induced damages, losses or replacement, see section **3.2** |
| Lightning | mandatory | We do not provide 'no climate change' lightning data. Instead the original Flash Rate Monthly Climatology (Cecil, 2006) should be used in the |

| | | |
|---|---|---|
| | | 'counterclim' set-up. |
| Oceanic forcings | | |
| Oceanic forcing data | - | We do not provide any counterfactual oceanic forcings, i.e. there is no 'no climate change' experiment proposed for the *marine ecosystems and fisheries* sector. |
| Coastal water levels | | |
| Coastal water levels | mandatory | Counterfactual monthly (1901 - 1978) and hourly (1979 - 2015) coastal water levels where long-term trends have been removed, see section **3.3** |
| Atmospheric composition or fluxes | | |
| Atmospheric $CO_2$ concentration | mandatory | 1901 levels ($[CO_2]$ = 296.13 ppm) of observed atmospheric $CO_2$ concentrations according to (Meinshausen et al., 2011) |
| Atmospheric $CH_4$ concentration | mandatory | 1901 levels of atmospheric $CH_4$ concentrations ($[CH_4]$ = 928.80 ppb), according to (Meinshausen et al., 2017) |


**Standard attribution experiment using counterfactual climate-related forcings and observed**
**variations of direct human forcings (counterclim + histsoc; default).** This is the twin experiment to
the default 'obsclim+histsoc' evaluation experiment. It uses the 'counterclim' climate-related forcings
as described in **Table 3** while all direct human forcings are the same as the ones used in the evaluation
experiment ('histsoc'). As the corresponding evaluation experiment aims to ensure that impact models
can fully capture the historical variations including its long-term trends, this experiment is best suited
for impact attribution. It is therefore the standard impact attribution experiment that each sector should
strive to follow.

**Fixed 2015 direct human forcing attribution experiment (counterclim + 2015soc; default).** This
is the twin experiment to the 'obsclim+2015soc' experiment. It uses the 'counterclim' climate-related
forcings as described in **Table 3** and constant direct human forcings at 2015 levels ('2015soc'). Impact
attribution using this experiment has caveats because the twin 'obsclim+2015soc' experiment is not
built to fully explain the historical observations including its trends. Impact attribution building on this
experiment therefore needs to find other means to ensure that the impact model correctly captures the
response to changes in the climate-related systems. It may e.g. build on the assumption that fixed direct
human forcings do not change the models' sensitivity to historical climate change. The impact models
that cannot account for varying historical direct human forcings can take up the attribution task through
this experiment.

**Fixed 1901 direct human forcing attribution experiment (counterclim + 1901soc; default).** This is
the twin experiment to the 'obsclim+1901soc' experiment. It allows for a quantification of the combined
effect of changes in all forcings (climate-related and direct human) during the historical period when
compared to the default evaluation experiment ('obsclim+histsoc'). It also allows for a quantification of
the effect of varying direct human drivers when compared to the 'counterclim+histsoc' experiment and
the effect of the 2015 to 1901 difference in direct human forcing if compared to the
'counterclim+2015soc' experiment, conditional on counterclim climate-related forcings.

**No direct human forcing attribution experiment (counterclim + nat; default)** This is the twin
experiment to the default 'obsclim+nat' experiment. It allows for a quantification of the effect of climate
change under conditions of absent direct human forcings but a natural state of the world. The 'nat'
experiment is included in the *biomes* sector protocol.

**Table 4: ISIMIP3a attribution experiments**

| Experiment | Short description | Period: Historical 1901-2019 |
|---|---|---|
| **counterfactual climate** <br> histsoc <br> **1st priority** | **CRF:** Detrended observational atmospheric climate forcing, detrended observed coastal water level forcings, and other CRF as listed in **Table 3** | **counterclim** |
| | **DHF:** Varying direct human influences according to observations | **histsoc** |
| **counterfactual climate** <br> 2015soc <br> **1st priority** | **CRF:** Detrended observational atmospheric climate forcing, detrended observed coastal water level forcings, and other CRF as listed in **Table 3** | **counterclim** |
| | **DHF:** Fixed 2015 levels of direct human forcing for the entire time period | **2015soc** |

| counterfactual climate 1901soc 2nd priority | **CRF:** Detrended observational atmospheric climate forcing, detrended observed coastal water level forcings, and other CRF as listed in **Table 3** | **counterclim** |
| | **DHF:** Fixed 1901 levels of direct human forcing for the entire time period | **1901soc** |
| counterfactual climate nat 2nd priority | **CRF:** Detrended observational atmospheric climate forcing, detrended observed coastal water level forcings, and other CRF as listed in **Table 3** | **counterclim** |
| | **DHF:** No direct human influences | **nat** |


**3 Climate-related forcing data**



**3.1 Observational atmospheric climate forcing data (factual + counterfactual)**



The data sets described in this section all contain the variables listed in **Table 5** at the resolution
indicated there. While section **3.1.1** described the standard atmospheric climate forcing as one
component of the default 'obsclim' CRF used within the evaluation experiments (see section **2.1.1**),
section **3.1.2** describes the derivation of the high resolution data used within the 'obsclim'-based
sensitivity experiments (see section **2.1.2**), and section **3.1.3** provides a description of the basic
approach and the references for the derivation of the counterfactual atmospheric climate forcings
used for the 'counterclim' experiments described in section **2.2**.

**Table 5:** Atmospheric climate variables provided as part of the climate-related forcing

| Variable | Variable specifier | Unit | Resolution | Datasets |
| --- | --- | --- | --- | --- |
| Near-Surface Relative Humidity | **hurs** | % | 0.5° grid, daily | GSWP3-W5E5 (factual and counterfactual, 1901-2019), 20CRv3-W5E5 (factual and counterfactual, 1901-2019), 20CRv3-ERA5 (factual and counterfactual, 1901-2021), 20CRv3 (factual and counterfactual, 1901-2015) |
| Near-Surface Specific Humidity | **huss** | kg kg-1 | 0.5° grid, daily | GSWP3-W5E5 (factual and counterfactual, 1901-2019), 20CRv3-W5E5 (factual and counterfactual, 1901-2019), 20CRv3-ERA5 |

| | | | | (factual and counterfactual, 1901-2021), 20CRv3 (factual and counterfactual, 1901-2015) |
|---|---|---|---|---|
| Precipitation (including snowfall) | **pr** | kg m-2 s-1 | 0.5° grid, daily | GSWP3-W5E5 (factual and counterfactual, 1901-2019), 20CRv3-W5E5 (factual and counterfactual, 1901-2019), 20CRv3-ERA5 (factual and counterfactual, 1901-2021), 20CRv3 (factual and counterfactual, 1901-2015) |
| | | | 30" grid, 90" grid, 300" grid, 1800" grid; daily | CHELSA-W5E5 (factual, 1979-2016) |
| Snowfall | **prsn** | kg m-2 s-1 | 0.5° grid, daily | GSWP3-W5E5 (factual only, 1901-2019, 0.5°) |
| Surface Air Pressure | **ps** | Pa | 0.5° grid, daily | GSWP3-W5E5 (factual and counterfactual, 1901-2019), 20CRv3-W5E5 (factual and counterfactual, 1901-2019), 20CRv3-ERA5 (factual and counterfactual, 1901-2021), 20CRv3 (factual and counterfactual, 1901-2015) |
| Surface Downwelling Longwave Radiation | **rlds** | W m-2 | 0.5° grid, daily | GSWP3-W5E5 (factual and counterfactual, 1901-2019), 20CRv3-W5E5 (factual and counterfactual, 1901-2019), 20CRv3-ERA5 (factual and counterfactual, 1901-2021), 20CRv3 (factual and counterfactual, 1901-2015) |
| Surface Downwelling Shortwave Radiation | **rsds** | W m-2 | 0.5° grid, daily | GSWP3-W5E5 (factual and counterfactual, 1901-2019), 20CRv3-W5E5 (factual and counterfactual, 1901-2019), 20CRv3-ERA5 (factual and counterfactual, 1901-2021), 20CRv3 (factual and counterfactual, 1901-2015) |

| | | | 30" grid, 90" grid, 300" grid, 1800" grid; daily | CHELSA-W5E5 (1979-2016) |
|---|---|---|---|---|
| Near-Surface Wind Speed | **sfcwind** | m s-1 | 0.5° grid, daily | GSWP3-W5E5 (factual and counterfactual, 1901-2019), 20CRv3-W5E5 (factual and counterfactual, 1901-2019), 20CRv3-ERA5 (factual and counterfactual, 1901-2021), 20CRv3 (factual and counterfactual, 1901-2015) |
| Near-Surface Air Temperature | **tas** | K | 0.5° grid, daily | GSWP3-W5E5 (factual and counterfactual, 1901-2019), 20CRv3-W5E5 (factual and counterfactual, 1901-2019), 20CRv3-ERA5 (factual and counterfactual, 1901-2021), 20CRv3 (factual and counterfactual, 1901-2015) |
| | | | 30" grid, 90" grid, 300" grid, 1800" grid; daily | CHELSA-W5E5 (1979-2016) |
| Daily Maximum Near-Surface Air Temperature | **tasmax** | K | 0.5° grid, daily | GSWP3-W5E5 (factual and counterfactual, 1901-2019), 20CRv3-W5E5 (factual and counterfactual, 1901-2019), 20CRv3-ERA5 (factual and counterfactual, 1901-2021), 20CRv3 (factual and counterfactual, 1901-2015) |
| | | | 30" grid, 90" grid, 300" grid, 1800" grid; daily | CHELSA-W5E5 (factual and counterfactual, 1979-2016) |
| Daily Minimum Near-Surface Air | **tasmin** | K | 0.5° grid, daily | GSWP3-W5E5 (factual and counterfactual, 1901-2019), 20CRv3-W5E5 (factual and counterfactual, 1901-2019), 20CRv3-ERA5 |

| Temperature | | | | (factual and counterfactual, 1901-2021), 20CRv3 (factual and counterfactual, 1901-2015) |
|---|---|---|---|---|
| | | | 30" grid, 90" grid, 300" grid, 1800" grid; daily | CHELSA-W5E5 (1979-2016) |


### 3.1.1 Default factual data

As one component of the default 'obsclim' CRFs, we provide four observational datasets specifically
generated for the evaluation experiments of ISIMIP3a: GSWP3-W5E5, 20CRv3-W5E5, 20CRv3-ERA5,
and 20CRv3. All four datasets have daily temporal and 0.5° spatial resolution and cover the variables
listed in **Table 5**. Their temporal coverage varies, with GSWP3-W5E5 and 20CRv3-W5E5 covering
1901-2019, while 20CRv3-ERA5 covers 1901-2021 and 20CRv3 covers 1901-2015. Instead of
excluding datasets that do not cover the most recent years, we focused on including datasets that start
in 1901, to allow for a common spin-up procedure (described in section **2.1** for the 'obsclim + histsoc;
default' experiment), in order to support models that need to spin up, e.g., their carbon pools under
stable climate-related and direct human forcings before they can do the actual experiments.

The GSWP3-W5E5 dataset is based on W5E5 v2.0 (Lange et al., 2021), which is also used as the
observational reference dataset for the bias adjustment of climate input data for ISIMIP3b that will be
described in an ISIMIP3b protocol paper(Frieler, submitted 2023). W5E5 v2.0 combines WFDE5 v2.0
((Cucchi et al., 2020) with data from the latest version of the European Reanalysis (ERA5; (Hersbach
et al., 2020) over the ocean. WFDE5 v2.0 is generated with the WATCH Forcing Data methodology
that includes bias adjustment of all variables (Cucchi et al., 2020). Since W5E5 v2.0 only covers the
years 1979 to 2019, it was extended backward in time to the year 1901. For this extension, we used
version 1.09 of the Global Soil Wetness Project phase 3 (GSWP3) dataset (Kim, 2017), bias-adjusted
to W5E5 v2.0 in order to reduce discontinuities at the 1978–1979 transition. The method used for this
bias adjustment was ISIMIP3BASD v2.5 (Lange, 2019, 2021). The GSWP3 dataset is a dynamically
downscaled and bias-adjusted version of the Twentieth Century Reanalysis version 2 (20CRv2; (Compo
et al., 2011)). For a detailed description of the GSWP3-W5E5 dataset and its constituents, see (Mengel
et al., 2021).

Unfortunately, for some variables, GSWP3 shows discontinuities at every turn of the month. The month-
by-month bias adjustment applied in its creation is responsible for this artefact (Rust et al., 2015). In
order to overcome this issue, which also affects GSWP3-W5E5, we additionally provide 20CRv3-
W5E5, a dataset where W5E5 v2.0 is backward-extended using ensemble member 1 of the Twentieth
Century Reanalysis version 3 (20CRv3; (Slivinski et al., 2019, 2021), interpolated to 0.5° and then bias-
adjusted to W5E5 v2.0 using ISIMIP3BASD v2.5. The 20CRv3-W5E5 data are continuous at every turn
of the month thanks to the application of ISIMIP3BASD v2.5 in running-window mode (see section **3.1**).
Since GSWP3 is based on 20CRv2, the 20CRv3-W5E5 dataset can be considered an update of
GSWP3-W5E5.

Two more climate input datasets are provided in ISIMIP3a in order to facilitate climate input data-related
quantifications of uncertainty in the associated impact assessments. Those datasets are not based on
W5E5 to account for trend and variability artefacts in W5E5 that are related to the climatological infilling
procedures used to deal with gaps in the station observations employed for the bias adjustment of
ERA5 for the production of WFDE5 (for a detailed description of this caveat see
https://data.isimip.org/caveats/20/). The first of the additional ISIMIP3a climate input datasets is
20CRv3-ERA5, which was created in the same way as 20CRv3-W5E5, but using ERA5 instead of
W5E5 for the time period 1979-2021, and also as the bias adjustment target for the time period 1901-
1978. Finally, we also provide the 'raw' 20CRv3 data, i.e., ensemble member 1 of 20CRv3, interpolated
to 0.5° but not bias-adjusted to any other dataset. This dataset is included since it was generated with
only one method and did not need to be combined with another dataset to fully cover the 20th century.

**3.1.2 High resolution atmospheric factual data (CHELSA-W5E5)**
This dataset is provided to facilitate the high resolution sensitivity experiment described in section **2.1.2**.
It covers the global land area at 30'' (~1 km) horizontal and daily temporal resolution from 1979 to 2016
for the variables precipitation (pr), surface downwelling shortwave radiation (rsds), and daily mean,
minimum and maximum near-surface air temperature (tas, tasmin, tasmax). CHELSA-W5E5 v1.0
(Karger et al., 2022b) is a downscaled version of the W5E5 v1.0 dataset, where the downscaling is
done with the Climatologies at High resolution for the Earth's Land Surface Areas (CHELSA) v2.0
algorithm (Karger et al., 2017, 2021, 2022a).

This algorithm applies topographic adjustments based on surface altitude (orog) information from the
Global Multi-resolution Terrain Elevation Data 2010 (GMTED2010; (Danielson and Gesch, 2011). The
algorithm is applied day by day. CHELSA-W5E5 tas is obtained by applying a lapse rate adjustment to
W5E5 tas, using differences between CHELSA-W5E5 orog and W5E5 orog in combination with
temperature lapse rates from ERA5. Those lapse rates are calculated based on atmospheric
temperature, $T$, at 950 hPa and 850 hPa, and the geopotential height, $z$, of those pressure levels. The
lapse rate used for the adjustment is calculated as the daily mean of hourly values of $(T\_850 - T\_950)/$
$(z\_850 - z\_950)$. The variables tasmax and tasmin are downscaled in the same way, using the same
lapse rate value.
Precipitation downscaling uses daily mean zonal and meridional wind components from ERA5 to
approximate the orographic wind effect on small-scale precipitation patterns (differences between
windward and leeward precipitation rates) and combines that with the height of the planetary boundary
layer to estimate the total orographic effect on precipitation intensity. Using that, precipitation from
W5E5 is downscaled such that precipitation fluxes are preserved at the original 0.5° resolution of W5E5.
More details are given in (Karger et al., 2021).
Surface downwelling shortwave radiation, rsds, at 30 arcsec resolution is strongly influenced by
topographic features such as aspect or terrain shadows, which are less pronounced at 0.5° resolution.
The downscaling algorithm combines such geometric effects with orographic effects on cloud cover for
an orographic adjustment of rsds. Geometric effects are considered by computing 30'' clear-sky
radiation estimates using the method described in (Karger et al., 2022a) and a simplified, uniform
atmospheric transmittance of 80%. These effects include shadowing from surrounding terrain, diffuse
radiation, and terrain aspect. To include how orographic effects on cloud cover influence rsds, the clear-
sky radiation estimates are adjusted using downscaled ERA5 total cloud cover. The cloud cover
downscaling uses ERA5 cloud cover at all pressure levels and the orographic wind field following the
methods described in (Brun et al., 2022b). Finally, the clear-sky radiation estimates adjusted for cloud
cover are rescaled such that they match W5E5 rsds, B-spline interpolated to 30''.
We provide the original CHELSA-W5E5 data with a horizontal resolution of 30'' = 0.5' (~1 km) as well
as spatially aggregated versions with resolutions of 1.5' (~3 km, aggregation factor 3), 5.0' (~10 km,
aggregation factor 10) and 30.0' = 0.5° (~60 km, aggregation factor 60). The aggregation to 0.5° is
necessary since the aggregated CHELSA-W5E5 data differ from the default GSWP3-W5E5 and
20CRv3-W5E5 data provided in the 'obsclim' set-up for 1979-2016. This has two reasons. First, the
downscaled data are based on W5E5 v1.0 whereas GSWP3-W5E5 and 20CRv3-W5E5 are based on
W5E5 v2.0. Secondly, for all variables except pr, the CHELSA downscaling algorithm produces data
that differs from the original data when it is upscaled (spatially aggregated) back to the original
resolution.
We do not provide a counterfactual version of the high resolution climate forcing.
The CHELSA method is not yet available for all variables included in the standard forcing data. Relative
humidity, surface wind, air pressure, and longwave radiation can not yet be downscaled by the
approach. To allow modellers to start the sensitivity experiments already now, we provide an alternative
downscaling approach as described below. We use observational data with the required higher spatial
resolution but lower temporal resolution to generate the high resolution daily relative humidity and
surface wind speeds. Air pressure is derived by on orographic correction of the linearly interpolated sea
level pressure and surface downwelling longwave radiation is derived from high-resolution
temperatures derived by CHELSA and relative humidity. The code required to generate the data is
freely available (Malle, 2023).
For daily mean near-surface relative humidity (hurs) the provided downscaling algorithm combines
monthly 30'' CHELSA-BIOCLIM+ data (Brun et al., 2022b, a) with daily W5E5 data. In a first step we
regrid daily 0.5° W5E5 hurs to the target grid (30") by bilinear interpolation. We assume relative humidity
to follow a beta-distribution and logit-transform both regridded monthly-averaged W5E5 ($hurs_{mon}^{W5E5}$) and
monthly CHELSA-BIOCLIM+ ($hurs_{mon}^{CHELSA}$) relative humidity data. The difference ($\Delta hurs_{mon}$) is then
added to daily regridded and logit-transformed W5E5 hurs of the respective month, and the final raster
is obtained by back-transforming the sum:
$$hurs_{dly} = \frac{1}{(1+exp^{-h})} \quad , (1)$$
where
$$h = log(\frac{hurs_{dly}^{W5E5}}{1 - hurs_{dly}^{W5E5}}) + \Delta hurs_{mon} \quad , (2)$$
$$\Delta hurs_{mon} = log(\frac{hurs_{mon}^{CHELSA}}{1 - hurs_{mon}^{CHELSA}}) - log(\frac{hurs_{mon}^{W5E5}}{1 - hurs_{mon}^{W5E5}}) \quad . (3)$$
To include orographic effects into daily mean near-surface wind speed (*sfcwind*) we follow the approach
of (Brun et al., 2022b), and use an aggregation of the Global Wind Atlas 3.0 data (Badger et al.,
n.d.)Technical University of Denmark(Badger et al., n.d.) in combination with daily 0.5° sfcwind from
W5E5. We first regrid both the Global Wind Atlas data and the W5E5 sfcwind data to the target grid of
30" using bilinear interpolation. The Global Wind Atlas data product ($sfcWind_{cli}^{GWA}$) represents average
wind speeds for 2008 to 2017. We therefore average daily regridded W5E5 data over this time period
($sfcWind_{cli}^{W5E5}$). We assume surface wind speeds follows a Weibull distribution and log-transform both
datasets before computing the difference $\Delta sfcWind_{cli}$, whereby a small positive constant (c) was added
to all data points before applying the transformation to avoid the problem that log(0) is undefined. We
add this difference layer ($\Delta sfcWind_{cli}$) to each log-transformed daily W5E5 raster, and back-transform
the sum to obtain the final daily mean near-surface wind speed raster:
$$sfcWind_{dly} = exp^{(log(sfcWind_{dly}^{W5E5} + c) + \Delta sfcWind_{cli})} - c \quad , (4)$$
where
$$\Delta sfcWind_{cli} = log(sfcWind_{cli}^{GWA} + c) - log(sfcWind_{cli}^{W5E5} + c) \quad . (5)$$

Daily mean surface air pressure (ps) is calculated using the barometric formula:
$$ps_{dly} = psl_{dly}^{W5E5} \times exp^{-(g \times orog \times M)/(T_0 \times R)} \quad , (6)$$
with $psl_{dly}^{W5E5}$ being the regridded 0.5° W5E5 daily mean sea-level pressure (bilinear interpolation to 30"),
*g* the gravitational acceleration constant (9.80665 m/s²), *orog* the altitude at which air pressure is
calculated (CHELSA-W5E5 orog, m), *M* the molar mass of dry air (0.02896968 kg/mol), *R* the universal
gas constant (8.314462618 J/(mol K)) and $T_0$ the sea level standard temperature (288.16 K).

For Surface Downwelling Longwave Radiation (*rlds*) we follow (Fiddes and Gruber, 2014) as well as
(Konzelmann et al., 1994), and account for orographic effects by reducing the clear-sky component of
all-sky emissivity with elevation. We assume cloud emissivity remains unchanged when moving from
coarse to fine resolution. First, we compute clear-sky emissivity components both for the 0.5° W5E5
grid and the target 30'' grid ($\epsilon_{clear}^{W5E5}$, $\epsilon_{clear}^{highres}$ respectively):
$$\epsilon_{clear}^{highres/W5E5} = 0.23 + x1(pV_{dly}^{highres/W5E5}/ \ tas_{dly}^{highres/W5E5})^{1/x2} \ , (7)$$
where x1 = 0.43 and x2 = 5.7 and $pV_{dly}^{highres/W5E5}$ is water vapour pressure as a function of relative
humidity at the respective resolution (see (Fiddes and Gruber, 2014). By using 0.5° W5E5 *rlds* and *tas*
data and inverting the Stefan-Boltzmann equation we obtain all-sky emissivity:
$$\epsilon_{allsky}^{W5E5} = rlds_{dly}^{W5E5}/ \ (\sigma \times (tas_{dly}^{W5E5})^4 \ ) \ , (8)$$
with $\sigma$ being the Stefan-Boltzmann constant (5.67 x $10^{-8}$ $Js^{-1}$ $m^{-2}$ $K^{-4}$). In a next step, the cloud-based
component of emissivity ($\Delta\epsilon_{dly}^{W5E5}$) can be estimated as the difference between all-sky and clear-sky
emissivity, which is then regridded to the target grid via bilinear interpolation.
$$\Delta\epsilon_{dly}^{W5E5} = \epsilon_{allsky}^{W5E5} - \epsilon_{clear}^{W5E5} \ (9)$$
In a last step we obtain elevation-corrected longwave radiation ($rlds_{dly}$) by adding $\Delta\epsilon_{dly}^{W5E5}$ to the high-
resolution clear-sky emissivity ($\epsilon_{clear}^{highres}$) and applying the Stefan-Boltzmann law again:
$$rlds_{dly} = (\epsilon_{clear}^{highres} + \Delta\epsilon_{dly}^{W5E5} \ ) \times \sigma \times (tas_{dly}^{highres})^4 \ (10)$$
As soon as the CHELSA approach is extended to also cover the missing variable we plan to also provide
these data and test for the sensitivity of the impact simulations to these two alternative downscaling
methods.

**3.1.3 Default counterfactual data.**
To simulate the baseline 'no climate change' state of a human or natural system that is required for
impact attribution, we provide a detrended version of the observational factual forcing data using the
ATTRICI approach (ATTRIbuting Climate Impacts, Mengel et al., 2021). The method identifies the long-
term shifts in the factual daily climate variables that are correlated to global mean temperature change
assuming a smooth annual cycle of the associated scaling coefficients for each day of the year. The
observed trends since 1901 are then removed from the observational data by projecting the observed
data onto the estimated distributions assuming a fixed 1901 level of global warming. The projection is
done through quantile mapping, a method borrowed from the bias adjustment literature. In this way we
preserve the internal variability of the observed data in the sense that factual and counterfactual data
for a given day have the same rank in their respective statistical distributions. The impact model
simulations forced by the counterfactual climate inputs therefore allow for quantifying the contribution
of the observed climate change (no matter from where the trends originate) to observed long-term
changes in impact indicators but also for quantifying the contribution of the observed trend in climate to
the magnitude of individual impact events.
**3.2 Tropical cyclone (TC) data (factual)**

**Figure 2: Tropical cyclone storm track (a and b, line with arrows),derived maximum wind speeds (a,**
**coloured shades) and accumulated rainfall totals (b, coloured shades)** of Hurrican Harvey that made landfall
in Texas (USA) in August 2017. The wind speeds are according to the Holland wind profile (Holland, 1980, 2008),
and the rainfall is according to the TCR model (Zhu et al., 2013). The colouring in (b) follows the "Tropical Cyclone
Severity Scale" (Bloemendaal et al., 2021).
**Table 6: Tropical cyclone information provided as part of the ISIMIP3a climate-related forcing**

| Variable | Variable specifier | Unit | Resolution | Datasets |
|---|---|---|---|---|
| Time associated with a given location of the storm centre | **time** | hours since 1950-01-01 00:00 | along-track, at least 3-hourly | IBTrACS (1950-2021, postprocessed) |
| Latitudinalcoordinate of storm centre (as defined by the reporting agencies) | **lat** | degrees north | along-track, at least 3-hourly | IBTrACS (1950-2021, postprocessed) |
| Longitudinal coordinate of storm centre (as defined by the reporting agencies) | **lon** | degrees east | along-track, at least 3-hourly | IBTrACS (1950-2021, postprocessed) |
| Ocean basin: NA/SA (North/South Atlantic), EP/WP/SP (East/West/South Pacific), NI/SI (North/South Indian Ocean) | **basin** | two-letter abbreviation | along-track, at least 3-hourly | IBTrACS (1950-2021, postprocessed) |
| Central pressure | **pres** | hPa | along-track, at least 3-hourly | IBTrACS (1950-2021, postprocessed) |
| Environmental pressure (pressure of the outermost closed isobar) | **penv** | mbar | along-track, at least 3-hourly | IBTrACS (1950-2021, postprocessed) |
| Maximum 1-minute sustained wind speed | **windspatial max** | knots | along-track, at least 3-hourly | IBTrACS (1950-2021, postprocessed) |
| Radius of maximum wind speeds | **rmw** | nautical miles | along-track, at least 3-hourly | IBTrACS (1950-2021, postprocessed) |
| Radius of the outermost closed isobar | **roci** | nautical miles | along-track, at least 3-hourly | IBTrACS (1950-2021, postprocessed) |

| Wind speed on the 850 hPa pressure level | **u850** **v850** | ms^-1 | along-track, at least 3-hourly | IBTrACS (1950-2021, postprocessed) |
|---|---|---|---|---|
| Temperature on the 600 hPa pressure level | **t600** | K | along-track, at least 3-hourly | IBTrACS (1950-2021, postprocessed) |
| 1-minute sustained wind speed | **wind** | ms^-1 | hourly on a 300 arc-seconds (~10 km) grid | according to the Holland wind profile (Holland, 1980, 2008) and the Emanuel-Rotunno wind profile (Emanuel and Rotunno, 2011) |
| Gridded lifetime maximum 1-minute sustained wind speed | **windlifetime max** | ms^-1 | per storm on a 300 arc-seconds (~10 km) grid | according to the Holland wind profile (Holland, 1980, 2008) and the Emanuel-Rotunno wind profile (Emanuel and Rotunno, 2011) |
| National territory exposed to wind speeds of at least 34, 48, 64, 96 knots | **34knarea** **48knarea** **64knarea** **96knarea** | km^2 | per storm and country | according to the Holland wind profile (Holland, 1980, 2008) and to the Emanuel-Rotunno wind profile (Emanuel and Rotunno, 2011) |
| Number of people exposed to wind speeds of at least 34, 48, 64, 96 knots | **34knpop** **48knpop** **64knpop** **96knpop** | count | per storm and country | according to the Holland wind profile (Holland, 1980, 2008) and to the Emanuel-Rotunno wind profile (Emanuel and Rotunno, 2011) and assuming temporally varying (histsoc) or fixed 2015 (2015soc) population distributions |

| | | | | (see section **4.1**). |
|---|---|---|---|---|
| Economic assets exposed to wind speeds of at least 34, 48, 64, 96 knots | **34knassets** **48knassets** **64knassets** **96knassets** | Int$ PPP 2005 | per storm and country | Windfields according to the Holland wind profile (Holland, 1980, 2008) and Emanuel-Rotunno wind profile (Emanuel and Rotunno, 2011) and assuming temporally varying (histsoc) or fixed 2015 (2015soc) asset distributions (see section **4.2**). |
| Total rainfall | **rain** | mm | hourly on a 300 arc-seconds (~10 km) grid | according to the Holland wind profile (Holland, 1980, 2008) and to the Emanuel-Rotunno wind profile (Emanuel and Rotunno, 2011) |
| Maximum 24-hourly rainfall total during the whole storm duration | **max_rain** | mm | per storm on a 300 arc-seconds (~10 km) grid | according to the Holland wind profile (Holland, 1980, 2008) and to the Emanuel-Rotunno wind profile (Emanuel and Rotunno, 2011) |

As additional CRF, we provide historical TC tracks (information about the observed location of minimal
pressure), with associated gridded wind and rain fields (see variable names and units in **Table 6** and
the maps of maximum wind speed and accumulated rainfall totals for the example of hurricane Harvey
in **Figure 2**). In addition to this purely CRF, we also provide wind exposure in terms of i) shares of
national territory affected by extreme winds speeds, ii) national shares of people exposed to extreme
winds speeds, and iii) national shares of economic assets affected by extreme winds speeds as derived
from the estimated wind fields and historical population and GDP distributions (see below). **Table 6**
provides a comprehensive list of all variables, their meaning and resolution as well as their source.

**TC Tracks (position of storm centre, central pressure, environmental pressure, radius of maximum wind speed and the outermost closed isobar).** We provide processed track information of historical TCs from 1950 to 2021. The information is derived from IBTrACS, the most comprehensive global dataset of historical TC activity (Knapp et al., 2010) that provides information about the location of the storm centre, the pressure at the centre and at the outermost closed isobar as well as the maximum 1-minute sustained wind speed as reported by the WMO Regional Specialised Meteorological Centers (RSMCs) and by agencies in Shanghai and Hong Kong. For recent events and most reporting agencies, IBTrACS also contains observational information about the radius from the centre where maximum wind speed is attained and the radius of the outermost closed isobar. Information is provided in at least 6-hourly time steps. Usually temporal resolution reaches three hours or even less. The latest version (v04r00) of IBTrACS is continuously updated with near real time data taken from regional meteorological agencies. The data is marked as provisional before it is replaced by so-called best track data up to two years after the events. IBTrACS contains data from 1842 to present, but coverage by the WMO RSMCs starts much later for some of the basins (around 1850 for the North Atlantic and South Indian, in 1905 for the South Pacific, in 1950 for the North Pacific, and in 1990 for the Northern Indian basin). Data quality is globally consistent starting from the mid 1970s when satellite observations became available.

The data set we provide uses best track data from 1950 to 2021. For each TC in IBTrACS, we merge the data of different reporting agencies into a single track data set with information about the following variables: time, location of the storm centre, ocean basin, central pressure, maximum 1-minute sustained wind speed, environmental pressure, radius of maximum wind speeds, and radius of the outermost closed isobar (see Table 8). Several processing steps are applied to ensure consistency and completeness of the data: For each storm, the variables that are not reported by the officially responsible WMO RSMC for this storm are taken from the next agency in the following list that did report this variable for this storm: the US agencies (NHC, JTWC, CPHC), Japanese Meteorological Agency, Indian Meteorological Department, MeteoFrance (La Reunion), Bureau of Meteorology (Australia), Fiji Meteorological Service, New Zealand MetService, Chinese Meteorological Administration, Hong Kong Observatory. Thus, for different storms, the same variable might be taken from different agencies. As sustained wind speeds are reported at different averaging intervals by different agencies, we use multiplicative factors to rescale all wind speeds to 1-minute sustained winds (Knapp and Kruk, 2010). All variables are extracted at the highest temporal resolution where time and location information is available in IBTrACS. Temporal reporting gaps within a variable are linearly interpolated so that the temporal resolution is at least 3-hourly. After interpolation, time steps where neither central pressure nor maximum wind speeds are available, are discarded. Tracks with less than two valid time steps are discarded. If at least one of central pressure or maximum wind speed is available, one variable is estimated from the other using statistical wind-pressure relationships. Missing RMW and ROCI values are estimated from the central pressure using statistical relationships. Finally, missing environmental

pressure values are filled with basin-specific defaults (1010 hPa for the Atlantic and Eastern Pacific,
1005 hPa for the Indian Ocean and Western Pacific, and 1004 hPa for the South Pacific).
We provide two additional along-track variables that are taken from the European Reanalysis (ERA5;
(Hersbach et al., 2020), and that are needed for the computation of precipitation (see below): The
temperature at the storm centre on the 600 hPa pressure level, and the wind speed on the 850 hPa
pressure level, averaged over the 200-500 km annulus around the storm centre.
**Gridded maps of (maximum) wind speeds.** We derive two different gridded wind field products from
an extrapolation of the observed TC track information to gridded estimates of surface wind speeds (1-
minute sustained winds at 10 metres above ground), at a spatial resolution of 300 arc-seconds
(approximately 10 km). The two products are based on circular wind fields from different radial wind
profiles. The first is a semiempirical model that estimates the full wind profile from the central pressure
variable based on the gradient wind balance assumption (Holland, 1980, 2008). The second, more
physics-based model uses the less-reliable maximum wind speed variable to derive the wind profile
from the boundary layer angular momentum balance (Emanuel and Rotunno, 2011). This wind profile
represents the storm's inner core very well, but tails off too sharply in the outer region (Chavas and Lin,
2016). However, for high-impact events, the core is the most relevant storm region, and outer wind
profiles are not analytically solvable, incurring considerable computational expense when applied to a
large track set.
In both cases, the circular wind fields are combined with translational wind vectors that arise from the
TC movement, assuming that the influence of translational wind decreases with distance from the TC
centre (Cyclone Database Manager, 2023). We use the highest available temporal resolution (up to 3-
hourly) provided in IBTrACS and interpolate it to 1-hourly resolution before applying the parametric
wind field models. In a postprocessing step, we also calculate the maximum value of wind speeds over
the duration of the TC event ('max_wind').
The approach by Holland has been successfully applied in socioeconomic risk and impact analyses
(Peduzzi et al., 2012; Geiger et al., 2018; Eberenz et al., 2021). The Emanuel-Rotunno approach has
been used for storm surge simulations (Krien et al., 2017; Marsooli et al., 2019; Gori et al., 2020; Yang
et al., 2021), and as the basis for the rain field model that we describe below (Feldmann et al., 2019).
**Wind Exposure.** As an extension of the tropical cyclone exposure data set TCE-DAT (Geiger et al.,
2018), we provide national shares of people and economic assets exposed to 1-minute sustained winds
above 34, 48, 64, and 96 knots for each storm. In addition to that, shares of national territory affected
by 1-minute sustained winds above 34, 48, 64, and 96 knots are provided. To estimate the exposed
population and assets we use the 'histsoc' population and GDP distributions described in section **4.1**
and section **4.2**, respectively. The GDP values are converted to assets by applying the decadal (2010-
2019) mean of national capital stock to GDP ratios from the Penn World Table version 10.0 (Feenstra
et al., 2015). We also provide exposed population and assets assuming fixed 2015 population and
asset distributions.
**Precipitation.** We are also planning to provide rainfall fields, following a physics-based model that
simulates convective TC rainfall by relating the precipitation rate to the total upward velocity within the
TC vortex (Zhu et al., 2013). The approach has been successfully applied in rainfall risk assessments
in the US (Feldmann et al., 2019; Gori et al., 2022). The rain rate will be  simulated for all events in the
IBTrACS database at 0.5-hourly temporal and 300 arc-seconds (approximately 10 km) spatial resolution
within a 1500 km radius around the storm centre. We provide the derived rainfall totals at hourly
resolution as well as the maximum 24-hourly rainfall total during the entire storm duration since this
variable is frequently used for rainfall risk assessment studies (Fagnant et al., 2020).
Different TC wind profiles can be used as an input for the rain field model (Lu et al., 2018; Xi et al.,
2020). We will provide the rainfall fields for the two wind profile models by Holland and Emanuel-
Rotunno that we also use for the wind fields described above.

**3.3 Coastal water levels (factual + counterfactual)**

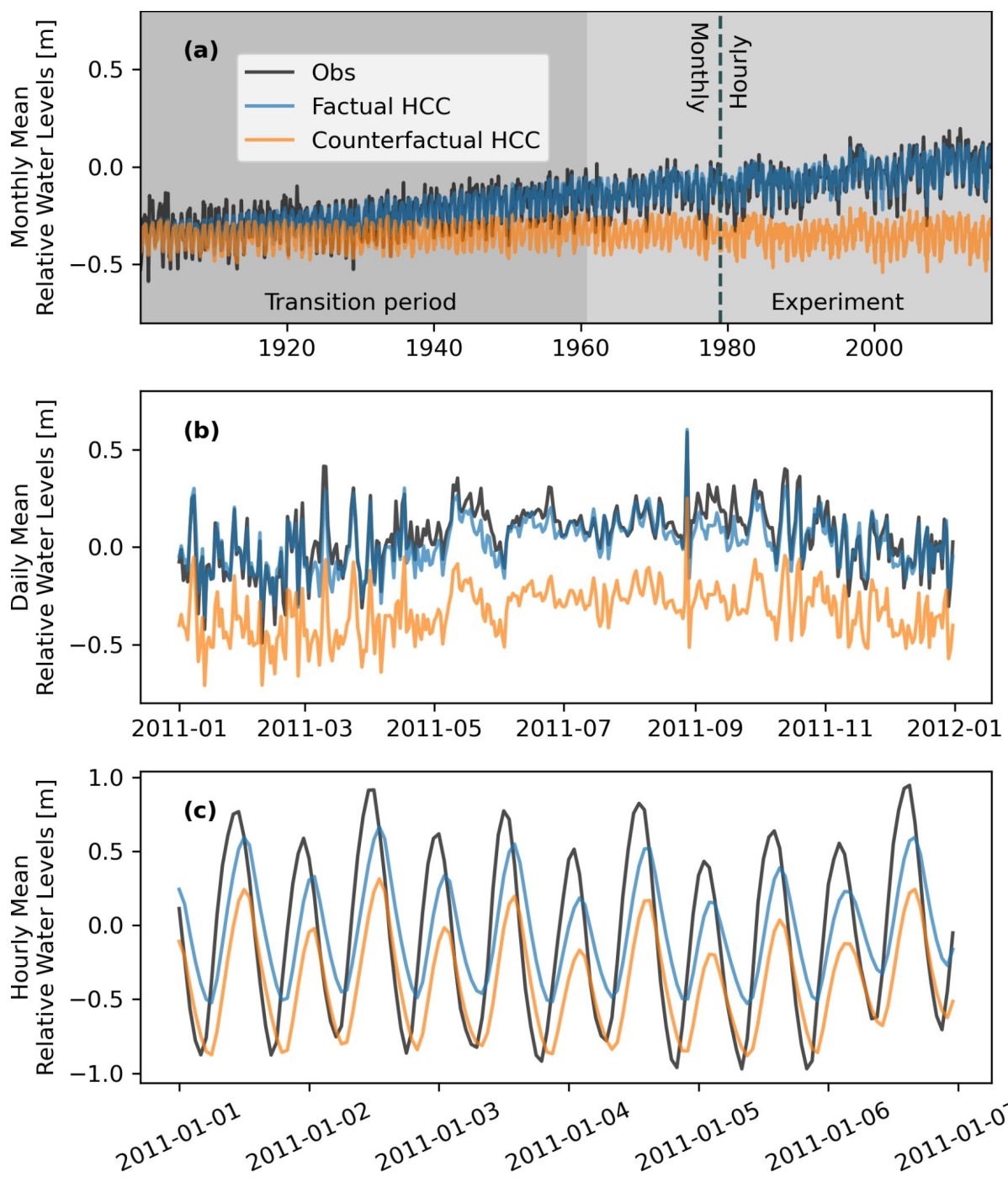

**Figure 3: Observed and reconstructed coastal relative water levels** at New York, USA. The counterfactual baseline represents water levels without long-term trend since 1900. Water levels are aggregated to monthly means in panel (a) and daily means in the year 2011 in panel (b) while panel c shows part of the data in hourly resolution. The reconstructed water levels are available as monthly mean values from 1901 to 1978 and as hourly mean values from 1979 to 2015.

**Table 7: Information about coastal water levels provided as ISIMIP3a climate-related forcing.**

| Variable | Variable specifier | Unit | Resolution | Datasets |
|---|---|---|---|---|
| Coastal water levels | **cwl** | m | custom coastal grid; monthly from 1901 to 1978 and hourly from 1979 to 2015 | HCC obsclim and counterclim (Treu et al.2023) |


To enable the quantification of impacts of historical relative sea level rise on coastal systems we provide
observation-based coastal water levels building on the HCC dataset (Hourly Coastal water levels with
Counterfactual; Treu et al.2023). In contrast to absolute sea levels, relative sea levels are measured
against a land-based reference frame (tide gauge measurements). This means that they are not only
determined by thermal expansion, loss of land ice, or dynamical processes influenced by climate
change, but also by vertical land movements (Wöppelmann and Marcos, 2016) induced by, e.g., glacial
isostatic adjustments (Caron et al., 2018; Whitehouse, 2018) or human interventions such as ground
water abstraction (Wada et al., 2016a). HCC encompasses factual and counterfactual coastal water
levels along global coastlines from 1901 to 1978 on monthly resolution and from 1979 to 2015 on hourly
resolution (see **Figure 3**). The counterfactual coastal water levels are derived from the factual dataset
by removing the trend in relative sea level since 1900. The detrending preserves the timing of historical
extreme sea-level events similar to the counterfactual atmospheric climate forcing described in section
**3.1** (see **Figure 3,** panel B). Hence, the data can be used for an event-based attribution of, e.g.,
observed flooding to observed relative sea-level rise with pairs of impact simulations driven with the
factual and counterfactual datasets. It is important to highlight that 'attribution to observed changes in
relative water levels' does not imply attribution to anthropogenic climate forcing because such observed
changes may include trends that are not driven by human greenhouse gas emissions. Important
sources for such trends are the ongoing adjustments of ice sheets, glaciers and the earth crust to
climate conditions before industrialization (Slangen et al. 2016) and the land subsidence due to water,
gas and oil extraction (Nicholls et al. 2021). In the following the derivation of the data is described in
more detail.

**Default factual data.** To capture the impacts of extreme water levels we provide hourly observation-
based coastal water levels as forcing data. To this end we combine the Coastal Dataset for the
Evaluation of Climate Impact (CoDEC) dataset (Muis et al., 2020) that describes high frequency
variation of sea level along global coastlines with a recent reconstruction of observed long-term sea-
level rise (Dangendorf et al., 2019). The CoDEC hourly data builds on a shallow-water model with fixed
ocean density driven by ERA5 wind and atmospheric pressure fields. The CoDEC data thus starts only
in the year 1979 and does not include variations due to ocean density changes and multi-year trends
from observed sea-level rise or vertical land movement. In contrast, the hybrid reconstructions (HR)
dataset from (Dangendorf et al., 2019) represents sea-level change since 1900 on a monthly timescale,
including density variations and multi-year trends. Long term sea-level change in HR is based on fitting
theoretically known and modelled spatial-temporal fields of individual contributing factors of sea level

change to a set of observations of sea level change from tide gauges. The individual contributing factors are theoretically known cryospheric fingerprints from two ice sheets, 18 major glacier regions, glacial isostatic adjustment from 161 Earth rheological models and dynamic changes of sea surface height modelled by six global climate models. Short term sea-level variations are represented in HR by extending the spatio-temporal patterns from satellite altimetry back to the year 1900 using tide gauge records. We create the HCC dataset by low-pass filtering the HR dataset and high-pass filtering the CoDEC dataset before summing them. Vertical land motion is subsequently added to yield relative changes of water levels along global coastlines. HCC shows improved agreement with tide gauge records on hourly to monthly time scales when compared to CoDEC due to the inclusion of density variations. This is most apparent for lower latitudes. The performance on interannual time scales is equal to (Dangendorf et al., 2019).

**Default counterfactual data.** To estimate the effects of historical sea-level rise on coastal systems, we provide a counterfactual sea-level dataset as forcing for coastal impact models (Treu et al. 2023). To this end the long term trend in the HCC data (1900-2015) was identified by a simple quadratic model in time and subtracted from the factual HCC data. The quadratic model assumes a constant acceleration of sea-level rise over time. Analysis of sea level rise acceleration shows variation throughout the last century with an acceleration phase in the early century followed by a deceleration and then again acceleration until today (Dangendorf et al., 2019). By design, this variation is not included in our quadratic trend estimate. In general, we expect our trend estimation to largely exclude natural variability from the trend due to the low dimensionality of the trend model and the long data period. This is a desired outcome and preserves the natural variability in the counterfactual. Extreme sea-level events have the same timing in the counterfactual and the factual dataset, facilitating event-based impact attribution.

**3.4 Ocean data (factual)**

**Default factual data.** For the fisheries and marine ecosystem models, we provide a number of physical and biogeochemical variables for the period 1961 to 2010 at different depth levels in the ocean (see **Table 8)**. Since direct measurements of these variables are very scarce (Sarmiento and Gruber, 2006, WOCE Atlas, 2023), the only way to obtain a globally (or even regionally) complete and consistent forcing dataset is to use numerical models. Global ocean models, which also serve as oceanic components of Earth System models, often simulate many or all of the required variables. To let observations at least indirectly enter the oceanic forcing data for ISIMIP3a, we provide outputs from an ocean model run that is forced by an observation-based reanalysis product of atmospheric forcing (Liu et al., 2021). Compared to the oceanic forcing (Stock et al., 2014) provided to generate the ISIMIP2a simulations for the *marine ecosystems and fisheries* sector (Tittensor et al., 2018), this new dataset is based on the latest GFDL-MOM6 and COBALTv2 physical and biogeochemical ocean models running on a tripolar 0.25° grid and using the JRA-55 reanalysis (Tsujino et al., 2018) as the surface forcing, in contrast to the inter-annual forcing dataset of (Large and Yeager, 2009), which was previously used to

drive GFDL-MOM4. The simulations also account for dynamic, time-varying river freshwater and nitrogen inputs that were simulated based on GFDL's land-watershed model LM3-TAN (Land Model version 3 with Terrestrial and Aquatic Nitrogen; (Lee et al., 2019), adjusted using observations from the Global Nutrient Export from WaterSheds (NEWS) database (Seitzinger et al., 2006). To create the default 'obsclim' climate-related forcings for the fisheries and marine ecosystem models these ocean model simulation data have been interpolated to a regular 0.25° grid while vertical resolution is preserved. In contrast to the atmospheric data, oceanic CRF are provided at monthly temporal resolution.

**Low resolution factual data.** To test to what degree a lower spatial resolution of the climate-related forcings affects the impact model simulations, the oceanic climate-related forcings have also been aggregated to one degree resolution as input for the 'obsclim + histsoc, 60arcmin' sensitivity experiment.

**CRF for the '1955-riverine-input' sensitivity experiment.** The '1955-riverine-inputs' sensitivity experiment builds on 0.25 degree GFDL-COBALT2 simulation forced by the JRA-55 reanalysis, but without time-varying riverine inputs. Instead the influx of freshwater and nutrients are fixed at mean 1951 to 1958 levels as described in the "control run" introduced by (Liu et al., 2021). The data is interpolated to a regular 0.25 degree grid in the same way as the default 'obsclim' CRFs.

We currently do not provide counterfactual versions of the ocean data forcing, though options are being explored.

**Table 8: ISIMIP3a oceanic climate-related forcing.** Variables with suffixes -bot, -surf, and -vint were obtained from the seafloor, the top layer of the ocean, and vertical integration, respectively.

| Variable | Variable specifier | Unit | Resolution | Datasets |
|---|---|---|---|---|
| Mass concentration of total phytoplankton expressed as chlorophyll | **chl** | kg m-3 | 0.25° and 1° grid, 35 levels (m from the surface), monthly | GFDL-COBALT2 simulation forced by the JRA-55 reanalysis, accounting for climate-driven changes in riverine inputs ('default') or assuming fixed levels of riverine inputs ('1955-riverine-input'). Standard salt water density of 1035 kg m-3 applied when converting from mass to volumetric unit, i.e. µg kg-1 to kg m-3 |

| | | | | |
|---|---|---|---|---|
| Downward flux of organic particles expressed as organic carbon at ocean bottom | **expc-bot** | mol m-2 s-1 | 0.25° and 1° grid, monthly | GFDL-COBALT2 simulation forced by the JRA-55 reanalysis, accounting for climate-driven changes in riverine inputs ('default') or assuming fixed levels of riverine inputs ('1955-riverine-input'). Derived from nitrogen detritus flux at ocean bottom (fndet_btm) by multiplying with fixed N-C ratio of 6.625.<br><br>Extractions for individual grid cells available in ASCII format for regional models (see **Table 1**). |
| Particulate organic carbon content in the upper 100 m | **intpoc** | kg m-2 | 0.25° and 1° grid, monthly | GFDL-COBALT2 simulation forced by the JRA-55 reanalysis, accounting for climate-driven changes in riverine inputs ('default') or assuming fixed levels of riverine inputs ('1955-riverine-input'). Derived by aggregating bacterial, detritus, diazotroph, large+small phytoplankton, large+medium+small zooplankton nitrogen biomass and multiplying by a fixed N-C ratio of 6.625.<br><br>Extractions for individual grid cells available in ASCII format for regional models (see **Table 1**). |
| Net primary organic carbon production by all types of phytoplankton in grid cell column | **intpp** | mol m-2 s-1 | 0.25° and 1° grid, monthly | GFDL-COBALT2 simulation forced by the JRA-55 reanalysis, both accounting for climate-driven changes in riverine inputs ('default') or assuming fixed levels of riverine inputs ('1955-riverine-input'). Derived by aggregating net primary productions by diatoms, |

| | | | | diazotrophs and pico-phytoplankton and under the assumption of a fixed N-C ratio of 6.625.<br><br>Extractions for individual grid cells available in ASCII format for regional models (see **Table 1**). |
|---|---|---|---|---|
| Net primary organic carbon production by diatoms in grid cell column | **intppdiat** | mol m-2 s-1 | 0.25° and 1° grid, monthly | GFDL-COBALT2 simulation forced by the JRA-55 reanalysis, both accounting for climate-driven changes in riverine inputs ('default') or assuming fixed levels of riverine inputs ('1955-riverine-input'). Derived under the assumption of a fixed N-C ratio of 6.625.<br><br>Extractions for individual grid cells available in ASCII format for regional models (see **Table 1**). |
| Net primary organic carbon production of carbon by diazotrophs in grid cell column | **intppdiaz** | mol m-2 s-1 | 0.25° and 1° grid, monthly | GFDL-COBALT2 simulation forced by the JRA-55 reanalysis, both accounting for climate-driven changes in riverine inputs ('default') or assuming fixed levels of riverine inputs ('1955-riverine-input'). Derived under the assumption of a fixed N-C ratio of 6.625.<br><br>Extractions for individual grid cells available in ASCII format for regional models (see **Table 1**). |

| Net Primary Mole Productivity of Carbon by Picophytoplankton in grid cell column | **intpppico** | mol m-2 s-1 | 0.25° and 1° grid, monthly | GFDL-COBALT2 simulation forced by the JRA-55 reanalysis, both accounting for climate-driven changes in riverine inputs ('default') or assuming fixed levels of riverine inputs ('1955-riverine-input'). Derived under the assumption of a fixed N-C ratio of 6.625. |
|---|---|---|---|---|
| Mixed Layer Ocean Thickness defined by a Sigma Theta difference (= density difference) of 0.125 kg m-3 compared to the surface | **mlotst-0125** | m | 0.25° and 1° grid, monthly | GFDL-COBALT2 simulation forced by the JRA-55 reanalysis, both accounting for climate-driven changes in riverine inputs ('default') or assuming fixed levels of riverine inputs ('1955-riverine-input') |
| Dissolved oxygen concentration; vertically resolved, at the bottom or at the surface, respectively | **o2, o2-bot, o2-surf** | mol m-3 | 0.25° and 1° grid, 35 levels (m from the surface), monthly | GFDL-COBALT2 simulation forced by the JRA-55 reanalysis, both accounting for climate-driven changes in riverine inputs ('default') or assuming fixed levels of riverine inputs ('1955-riverine-input'). <br><br> Extractions for individual grid cells of the bottom and surface layer available in ASCII format for regional models (see **Table 1**). |
| pH; vertically resolved, at the bottom or at the surface, respectively | **ph, ph-bot, ph-surf** | 1 | 0.25° and 1° grid, 35 levels (m from the surface), ocean bottom and surface fields, monthly | GFDL-COBALT2 simulation forced by the JRA-55 reanalysis, both accounting for climate-driven changes in riverine inputs ('default') or assuming fixed levels of riverine inputs ('1955-riverine-input') where pH is derived from ion concentrations H+ as pH = $-\log_{10}(H+)$. |

| | | | | |
|---|---|---|---|---|
| | | | | Extractions for individual grid cells of the bottom and surface layer available in ASCII format for regional models (see **Table 1**). |
| Total phytoplankton carbon concentration; vertically resolved or integrated over the grid cell column, respectively | **phyc, phyc-vint** | mol m-3 | 0.25° and 1° grid, 35 levels (m from the surface) and vertically integrated, monthly | GFDL-COBALT2 simulation forced by the JRA-55 reanalysis, both accounting for climate-driven changes in riverine inputs ('default') or assuming fixed levels of riverine inputs ('1955-riverine-input'). Aggregated from diatom, diazotroph and pico-phytoplankton. Standard salt water density of 1035 kg m-3 and fixed N-C ratio of 6.625 applied when converting from mass to volumetric unit, i.e. mol kg-1 to mol m-3. Extractions for individual grid cells of the vertically integrated data set are available in ASCII format for regional models (see **Table 1**). |
| Concentration of diatoms expressed as carbon in sea water; vertically resolved or integrated over the grid cell column, respectively | **phydiat, phydiat-vint** | mol m-3 | 0.25° and 1° grid, 35 levels (m from the surface) and vertically integrated, monthly | GFDL-COBALT2 simulation forced by the JRA-55 reanalysis, both accounting for climate-driven changes in riverine inputs ('default') or assuming fixed levels of riverine inputs ('1955-riverine-input'). Standard salt water density of 1035 kg m-3 and fixed N-C ratio of 6.625 applied when converting from mass to volumetric unit, i.e. mol kg-1 to mol m-3. Extractions for individual grid cells of the vertically integrated data set are available in ASCII format for |

| | | | | |
|---|---|---|---|---|
| | | | | regional models (see **Table 1**). |
| Concentration of diazotrophs expressed as carbon in sea water; vertically resolved or integrated over the grid cell column, respectively | **phydiaz, phydiaz-vint** | mol m-3 | 0.25° and 1° grid, 35 levels (m from the surface) and vertically integrated, monthly | GFDL-COBALT2 simulation forced by the JRA-55 reanalysis, both accounting for climate-driven changes in riverine inputs ('default') or assuming fixed levels of riverine inputs ('1955-riverine-input'). Standard salt water density of 1035 kg m-3 and fixed N-C ratio of 6.625 applied when converting from mass to volumetric unit, i.e. mol kg-1 to mol m-3. |
| Mole concentration of picophytoplankton expressed as carbon in sea water; vertically resolved or integrated over the grid cell column, respectively | **phypico, phypico-vint** | mol m-3 | 0.25° and 1° grid, 35 levels (m from the surface) and vertically integrated, monthly | GFDL-COBALT2 simulation forced by the JRA-55 reanalysis, both accounting for climate-driven changes in riverine inputs ('default') or assuming fixed levels of riverine inputs ('1955-riverine-input'). Standard salt water density of 1035 kg m-3 and fixed N-C ratio of 6.625 applied when converting from mass to volumetric unit, i.e. mol kg-1 to mol m-3. |
| Net downward shortwave radiation at sea water surface | **rsntds** | W m-2 | 0.25° and 1° grid, monthly | From JRA-55 reanalysis |
| Sea ice area fraction | **siconc** | % | 0.25° and 1° grid, monthly | From JRA-55 reanalysis |

| Sea water salinity; vertically resolved, at the bottom, or at the surface, respectively | **so, so-bot, so-surf** | 0.001 | 0.25° and 1° grid, 35 levels (m from the surface), ocean bottom and surface fields, monthly | GFDL-COBALT2 simulation forced by the JRA-55 reanalysis, both accounting for climate-driven changes in riverine inputs ('default') or assuming fixed levels of riverine inputs ('1955-riverine-input'). Extractions for individual grid cells of the surface and bottom layer are available in ASCII format for regional models (see **Table 1**). |
|---|---|---|---|---|
| Sea water potential temperature | **thetao** | °C | 0.25° and 1° grid, 35 levels (m from the surface), monthly | GFDL-COBALT2 simulation forced by the JRA-55 reanalysis, both accounting for climate-driven changes in riverine inputs ('default') or assuming fixed levels of riverine inputs ('1955-riverine-input') |
| Ocean model cell thickness | **thkcello** | m | 0.25° and 1° grid, 35 levels (m from the surface), constant | GFDL-COBALT2 simulation forced by the JRA-55 reanalysis, both accounting for climate-driven changes in riverine inputs ('default') or assuming fixed levels of riverine inputs ('1955-riverine-input') |
| Sea water potential temperature at sea floor (bottom) | **tob** | °C | 0.25° and 1° grid, monthly | GFDL-COBALT2 simulation forced by the JRA-55 reanalysis, both accounting for climate-driven changes in riverine inputs ('default') or assuming fixed levels of riverine inputs ('1955-riverine-input'). Extractions for individual grid cells are available in ASCII format for regional models (see **Table 1**). |

| Sea surface temperature | **tos** | °C | 0.25° and 1° grid, monthly | GFDL-COBALT2 simulation forced by the JRA-55 reanalysis, both accounting for climate-driven changes in riverine inputs ('default') or assuming fixed levels of riverine inputs ('1955-riverine-input'). Extracted from uppermost ocean layers potential temperatures. Extractions for individual grid cells are available in ASCII format for regional models (see **Table 1**). |
|---|---|---|---|---|
| Sea water zonal velocity | **uo** | m s-1 | 0.25° and 1° grid, 35 levels (m from the surface), monthly | GFDL-COBALT2 simulation forced by the JRA-55 reanalysis, both accounting for climate-driven changes in riverine inputs ('default') or assuming fixed levels of riverine inputs ('1955-riverine-input') |
| Sea water meridional velocity | **vo** | m s-1 | 0.25° and 1° grid, 35 levels (m from the surface), monthly | GFDL-COBALT2 simulation forced by the JRA-55 reanalysis, both accounting for climate-driven changes in riverine inputs ('default') or assuming fixed levels of riverine inputs ('1955-riverine-input') |
| Concentration of zooplankton of meso size expressed as carbon in seawater; vertically resolved or integrated over the grid cell column, respectively | **zmeso, zmeso-vint** | mol m-3 | 0.25° and 1° grid, 35 levels (m from the surface) and vertically integrated, monthly | GFDL-COBALT2 simulation forced by the JRA-55 reanalysis, both accounting for climate-driven changes in riverine inputs ('default') or assuming fixed levels of riverine inputs ('1955-riverine-input'). Aggregated from large and medium zooplankton. Standard salt water density of 1035 kg m-3 and fixed N-C ratio of 6.625 applied when converting from |

| | | | | mass to volumetric unit, i.e. mol kg-1 to mol m-3.

Extractions for individual grid cells of the vertically integrated data set are available in ASCII format for regional models (see **Table 1**). |
|---|---|---|---|---|
| Concentration of zooplankton of micro scale expressed as carbon in seawater; vertically resolved or integrated over the grid cell column, respectively. | **zmicro, zmicro-vint** | mol m-3 | 0.25° and 1° grid, 35 levels (m from the surface) and vertically integrated, monthly | GFDL-COBALT2 simulation forced by the JRA-55 reanalysis, both accounting for climate-driven changes in riverine inputs ('default') or assuming fixed levels of riverine inputs ('1955-riverine-input'). Standard salt water density of 1035 kg m-3 and fixed N-C ratio of 6.625 applied when converting from mass to volumetric unit, i.e. mol kg-1 to mol m-3.

Extractions for individual grid cells of the vertically integrated data set are available in ASCII format for regional models (see **Table 1**). |
| Total Zooplankton Carbon Concentration; vertically resolved or integrated over the grid cell column, respectively | **zooc, zooc-vint** | mol m-3 | 0.25° and 1° grid, 35 levels (m from the surface) and vertically integrated, monthly | GFDL-COBALT2 simulation forced by the JRA-55 reanalysis, both accounting for climate-driven changes in riverine inputs ('default') or assuming fixed levels of riverine inputs ('1955-riverine-input'), aggregated from large, medium and micro zooplankton. Standard salt water density of 1035 kg m-3 and fixed N-C ratio of 6.625 applied when converting from mass to volumetric unit, i.e. mol kg-1 to mol m-3.

Extractions for individual grid cells |

| | | | | of the vertically integrated data set are available in ASCII format for regional models (see **Table 1**). |
|---|---|---|---|---|


## 4 Direct human forcings


## 4.1 Population data

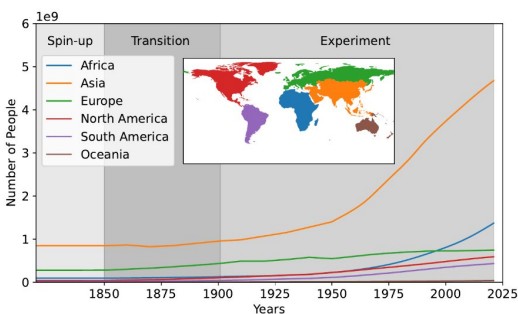 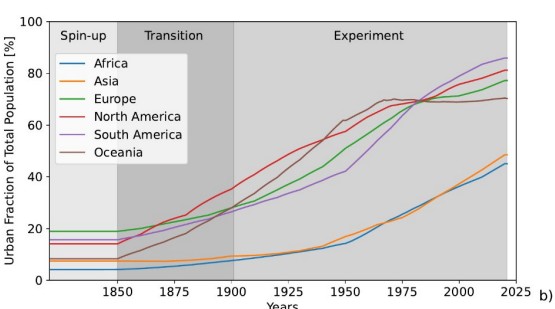


**Figure 4: Historical evaluation of population for different continents.** Total number of people living in the region (panel a) and urban population as a fraction of the total population per region (panel b).


**Table 9: Population data provided as part of the ISIMIP3a direct human forcing.**

| Variable | Variable specifier | Unit | Resolution | Datasets |
|---|---|---|---|---|
| National population | **pop** | Number of people in millions | annual | UN 2019 WPP database (2023): census-based from 1950 to 2020 + "medium-variant" forecast provided for 2021 |
| Gridded total population | **total-population** | Number of people | 0.5°x 0.5°, annual | HYDE3.3 data for 1950-2020 constantly extended to 2021 and adjusted to match the national UN numbers described above (see text below) |
| Gridded rural population | **rural-population** | Number of people | 0.5°x 0.5°, annual | HYDE3.3 data for 1950-2020 constantly extended to 2021 and rescaled by the same national scaling factors as the total population |
| Gridded urban population | **urban-population** | Number of people | 0.5°x 0.5°, annual | HYDE3.3 data for 1950-2020 constantly extended to 2021 and rescaled by the same national scaling factors as the total population |


For ISIMIP3a we provide consistent gridded and national population data (see **Table 9**) by rescaling
the gridded data to match the national aggregates. **Figure 4** shows the temporal evolution of total and
urban population for different continents.

**National data.** Annual national population data are taken from the 2019 UN World Population Prospects
(WPP) database for the period from 1950 – 2021 (United Nations, 2019). The 2019 revision of the WPP
provides census-based population numbers from 1950 through 2020. For the year 2021, we use the
"medium-variant" of the probabilistic forecast also provided by the WPP. The forecast accounts the
past experience of each country, while reflecting uncertainty about future changes based on the past
experience of other countries under similar conditions (see United Nations, 2019 for details). For
countries not covered in the database, estimates are taken from the MissingIslands dataset (Arujo et
al., 2021) to finally provide population data for 249 countries.

**Gridded data.** We provide gridded population data that is based on HYDE v3.3 (Klein Goldewijk, 2022).
Just like the original dataset we provide total, rural and urban population per grid cell. The original HYDE
3.3 data was on a 1/12°×1/12° grid and has been interpolated to ISIMIP's 0.5°×0.5° grid. Furthermore,
the land-sea distinction was modified to comply with the ISIMIP country mask (see **Table 1**). Before the
year 1950 HYDE provides data every ten years, the intermediate years have been filled by linear
interpolation. Also, the original HYDE data ends in 2020. So to cover the whole ISIMIP3a time frame
the final year 2020 has been duplicated as 2021. In this way annual coverage of 1850 to 2021 has been
achieved.
Data for all grid cells of a country, as defined by the ISIMIP 0.5°×0.5° fractional country map (see **Table**
**1**), have been rescaled such that the country's total population matches the numbers provided in the
national population data. Since the national data only starts in 1950, all years prior to 1950 have been
rescaled by the national scaling factors of 1950. The urban and rural populations have been rescaled
by the same national scaling factors as the total population.

**4.2 Gross Domestic Product (GDP)**

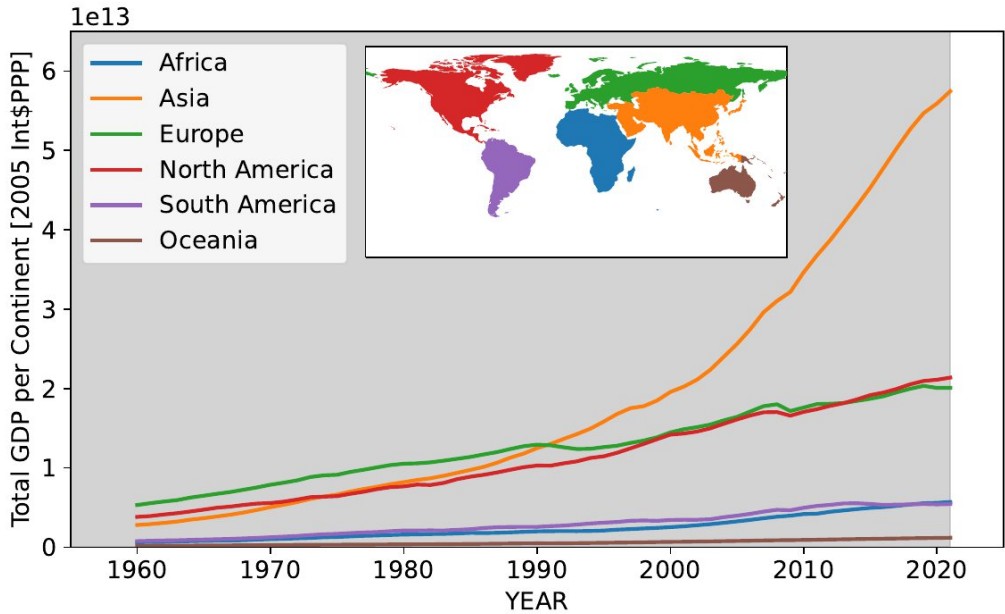

**Figure 5: Aggregated GDP (Int$ PPP 2005) for different continents.**

**Table 10:  GDP data provided as part of the ISIMIP3a direct human forcing.**

| Variable | Variable specifier | Unit | Resolution | Datasets |
|---|---|---|---|---|
| National Gross Domestic Product | **gdp** | Int$ PPP 2005 | annual | World Bank's World Development Indicator database (Anon, 2008) |
| Gridded Gross Domestic Product | **gridded-gdp** | Int$ PPP 2005 | annual | National GDP data downscaled to the 0.5° grid according to (Wang and Sun, 2022) |

Similar to the population data we also provide gridded and national GDP data (see **Table 10**). The downscaling of the national numbers is based on population and nightlight data (see below). In contrast to ISIMIP2a the gridded GDP and population data are now consistent such that previous artefacts in the derived GDP per capita could be eliminated (see below). **Figure 5** shows the historical increase in GDP for different continents.

**National GDP data.** Time series of per-capita GDP for the time period 1960-2021 are taken from the World Bank's World Development Indicator database (WDI) (Anon, 2008) and converted into constant 2005 Int$PPP, using deflators and PPP conversion factors from WDI. For countries not covered in the

WDI database, data from the MissingIslands dataset (Arujo et al., 2021) is used to allow covering 249
countries. Following a method developed by (Koch and Leimbach, 2023); the values for the year 2021
are derived from the IMF's World Economic Outlook short-term estimates of GDP per capita growth
(International Monetary Fund, 2021) that comprise estimates of the growth impacts of the Covid-19
shock.

**Gridded GDP data**. Gridded GDP data at 0.5 degree resolution are derived from the national GDP time
series by applying the LitPop method (Zhao et al., 2017; Eberenz et al., 2019), which uses the ISIMIP3a
gridded population based on HYDE v.3.3 and nighttime light (NTL) data to downscale national GDP
data for the period 1960-2021 to the ISIMIP 0.5°×0.5° grid.

As the disaggregation of GDP is not only based on population but also uses the NTL GDP per capita,
it is not constant within different countries. Deriving the gridded GDP data from the gridded population
data provided within ISIMIP3a ensures that the both data sets can be combined such that the associated
GDP per capita does no longer show the artefacts that have been found in the ISIMIP2a GDP per capita
(ISIMIP2a: suspicious gridded GDP per capita data; new functions in the isimip data repository; Forum
on      Scenarios      for      Climate      and      Societal      Futures,      2023).


**4.3 Land use and irrigation patterns**

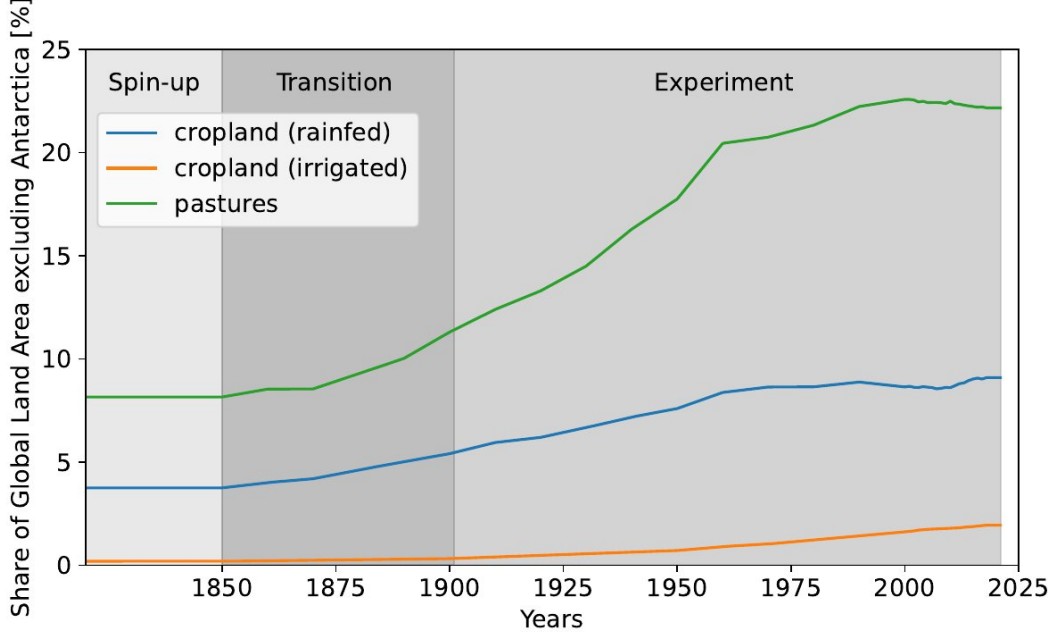

**Figure 6: Share of Global Land Area excluding Antarctica covered by rainfed cropland (green), irrigated**
**cropland (blue), and pasture (orange) [%].** The information is from the LUH v2 data set provided as direct human
forcing for ISIMIP3a (see details below).

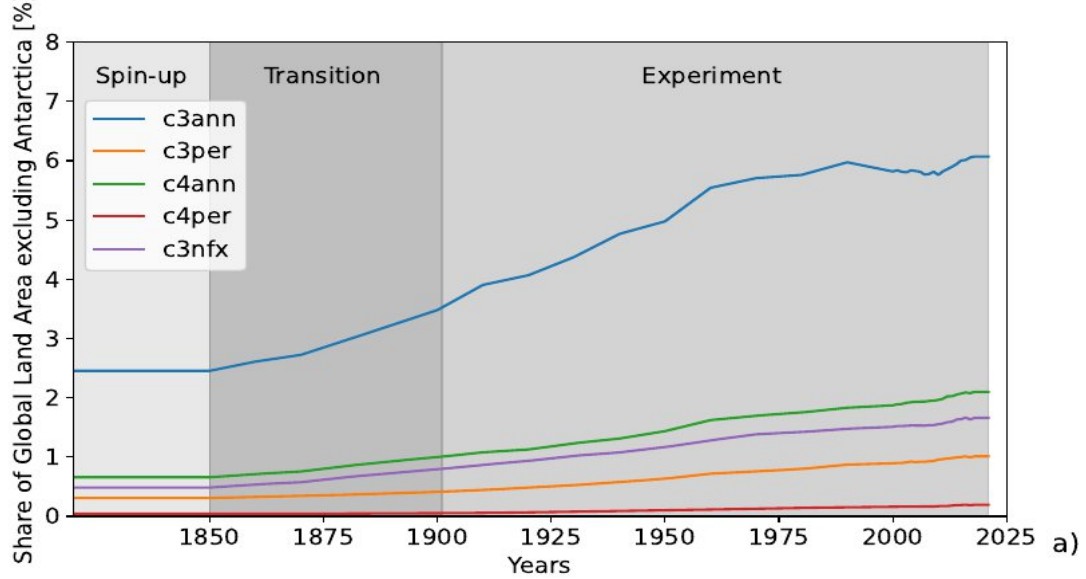

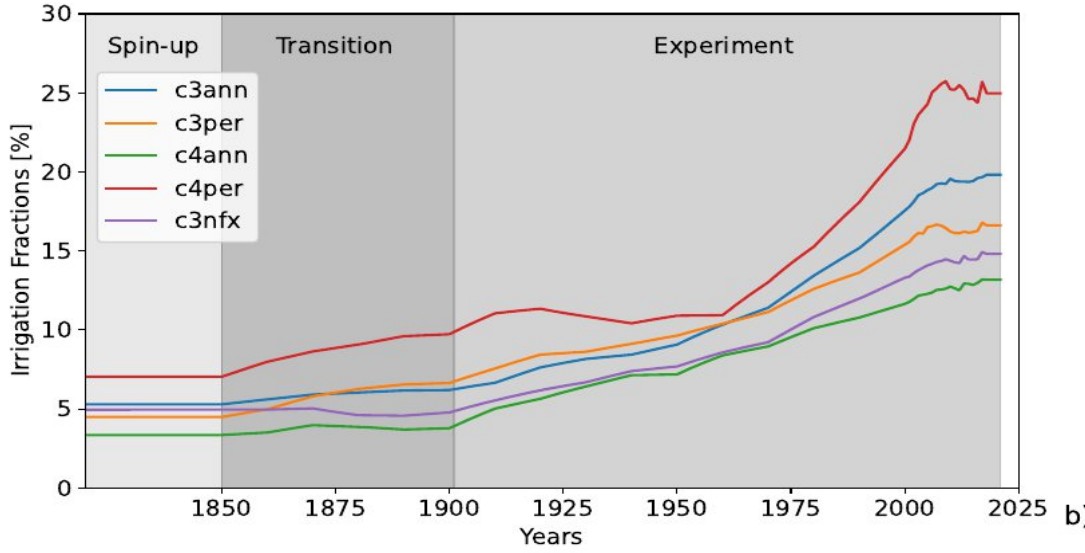


**Figure 7: Panel A: Share of Global Land Area excluding Antarctica covered by different groups of crops**
**(C3 annual (blue), C3 perennial (orange), C4 annual (green), C4 perennial (red), C3 nitrogen fixing (purple)).**
**Panel B: Fraction of irrigated land for the different groups of crops.** The information is from the LUH v2 data
set (see details on further disaggregation of the LUH v2 groups below).
**Table 11: Historical land use and irrigation patterns provided as part of the ISIMIP3a direct human forcing.**

| Variable | Variable specifier | Unit | Resolution | Datasets |
|---|---|---|---|---|
|  |  |  |  |  |

| Total crop land, rainfed cropland, irrigated cropland | **cropland_total, cropland_rainfed, cropland_irrigated** | unitless (share of area in a grid cell) | 0.5°×0.5°, annual | LUH2 v2 (Hurtt et al., 2020, Land use harmonization, 2023) |
|---|---|---|---|---|
| pastures | **pastures** | unitless (share of area in a grid cell) | 0.5°×0.5°, annual | sum of 'managed_pastures' and 'rangeland' from HYDE 3.2 (see below) |
| Managed pastures | **managed_pastures** | 1 (share of area in a grid cell) | 0.5°×0.5°, annual | first subcategory of 'pastures' from HYDE 3.2 (see above) |
| rangeland | **rangeland** | 1 (share of area in a grid cell) | 0.5°×0.5°, annual | second subcategory of 'pastures' from HYDE 3.2, more extensive management than 'managed pastures' (see above) |
| C3 annual rainfed cropland, C3 annual irrigated cropland | **c3ann_irrigated, c3ann_rainfed** | 1 (share of area in a grid cell) | 0.5°×0.5°, annual | LUH v2, for the disaggregation we consider C3 annual to be: rapeseed, rice, temperate cereals, temperate roots, tropical roots, sunflower, others C3 annual (see below) |

| C3 perennial cropland | **c3per_irrigated, c3per_rainfed** | 1 (share of area in a grid cell) | 0.5°×0.5°, annual | LUH v2 (this variable appears in the file only distinguishing 5 land use types and in the file with the downscaled 15 land use types. The provided values are identical) |
|---|---|---|---|---|
| C3 nitrogen-fixing rainfed cropland, C3 nitrogen-fixing irrigated cropland | **c3nfx_irrigated, c3nfx_rainfed** | 1 (share of area in a grid cell) | 0.5°×0.5°, annual | LUH v2 for the disaggregation we consider 'C3 nitrogen-fixing' to be: groundnut, pulses, soybean, others C3 nitrogen-fixing (see below) |
| C4 annual rainfed cropland, C4 annual irrigated cropland | **c4ann_irrigated, c4ann_rainfed** | 1 (share of area in a grid cell) | 0.5°×0.5°, annual | LUH v2, for the disaggregation we consider 'C4 annual' to be: maize, tropical cereals (see below) |
| C4 perennial rainfed cropland, C4 perennial irrigated cropland | **c4per_irrigated, c4per_rainfed** | 1 (share of area in a grid cell) | 0.5°×0.5°, annual | LUH v2 (this variable appears in the file only distinguishing 5 land use types and in the file with the downscaled 15 land use types. The provided values are identical), in the file with the 15 crops 'C4 perennial' is considered to be |

| | | | | sugarcane |
|---|---|---|---|---|
| Fraction of grid cell where maize is grown (rainfed and irrigated) | **maize_irrigated, maize_rainfed** | 1 (share of area in a grid cell) | 0.5°×0.5° annual | downscaled from LUH v2 data based on the crop distribution from (Monfreda et al., 2008). The method is described in (Frieler et al., 2017) |
| Fraction of grid cell where groundnut is grown (rainfed and irrigated) | **oil_crops_groundnut_irrigated, oil_crops_groundnut_rainfed,** | 1 (share of area in a grid cell) | 0.5°×0.5° annual | downscaled from LUH v2 data based on the crop distribution from (Monfreda et al., 2008). The method is described in (Frieler et al., 2017) |

| Fraction of grid cell where rapeseed is grown (rainfed and irrigated) | **oil_crops_rapeseed_irrigated, oil_crops_rapeseed_rainfed** | 1 (share of area in a grid cell) | 0.5°×0.5°, annual | downscaled from LUH v2 data based on the crop distribution from (Monfreda et al., 2008). The method is described in (Frieler et al., 2017) |
|---|---|---|---|---|
| Fraction of grid cell where soybean is grown (rainfed and irrigated) | **oil_crops_soybean_irrigated, oil_crops_soybean_rainfed** | 1 (share of area in a grid cell) | 0.5°×0.5°, annual | downscaled from LUH v2 data based on the crop distribution from (Monfreda et al., 2008). The method is described in (Frieler et al., 2017) |
| Fraction of grid cell where sunflower is grown (rainfed and irrigated) | **oil_crops_sunflower_irrigated, oil_crops_sunflower_rainfed** | 1 (share of area in a grid cell) | 0.5°×0.5°, annual | downscaled from LUH v2 data based on the crop distribution from (Monfreda et al., 2008). The method is described in (Frieler et al., 2017) |
| Fraction of grid cell where pulses are grown (rainfed and irrigated) | **pulses_irrigated, pulses_rainfed** | 1 (share of area in a grid cell) | 0.5°×0.5°, annual | downscaled from LUH v2 data based on the crop distribution from (Monfreda et al., 2008). The method is described in (Frieler et al., 2017) |
| Fraction of grid cell where rice is grown (rainfed and irrigated) | **rice_irrigated, rice_rainfed** | 1 (share of area in a | 0.5°×0.5°, annual | downscaled from LUH v2 data based on the crop distribution from |

| | | | | |
|---|---|---|---|---|
| | | grid cell) | | (Monfreda et al., 2008). The method is described in (Frieler et al., 2017) |
| Fraction of grid cell where temperate cereals are grown (rainfed and irrigated) | **temperate_cereals_irrigated, temperate_cereals_rainfed** | 1 (share of area in a grid cell) | 0.5°×0.5°, annual | downscaled from LUH v2 data based on the crop distribution from (Monfreda et al., 2008). The method is described in (Frieler et al., 2017) |
| Fraction of grid cell where temperate roots are grown (rainfed and irrigated) | **temperate_roots_irrigated, temperate_roots_rainfed** | 1 (share of area in a grid cell) | 0.5°×0.5°, annual | downscaled from LUH v2 data based on the crop distribution from (Monfreda et al., 2008). The method is described in (Frieler et al., 2017) |
| Fraction of grid cell where tropical cereals are grown (rainfed and irrigated) | **tropical_cereals_irrigated, tropical_cereals_rainfed** | 1 (share of area in a grid cell) | 0.5°×0.5°, annual | downscaled from LUH v2 data based on the crop distribution from (Monfreda et al., 2008). The method is described in (Frieler et al., 2017) |
| Fraction of grid cell where tropical roots are grown (rainfed and irrigated) | **tropical_roots_irrigated, tropical_roots_rainfed** | 1 (share of area in a grid cell) | 0.5°×0.5°, annual | downscaled from LUH v2 data based on the crop distribution from (Monfreda et al., 2008). The method is described in (Frieler et al., 2017) |

| | | | | |
|---|---|---|---|---|
| Fraction of grid cell where C3 annual crops other than rapeseed, rice, temperate cereals, temperate roots, tropical roots, and sunflower are grown (rainfed and irrigated) | **others_c3ann_irrigated, others_c3ann_rainfed** | 1 (share of area in a grid cell) | 0.5°×0.5°, annual | downscaled from LUH v2 data based on the crop distribution from (Monfreda et al., 2008). The method is described in (Frieler et al., 2017) |
| Fraction of grid cell where nitrogen fixing C3 crops other than groundnut, pulses, and soybean are grown (rainfed and irrigated) | **others_c3nfx_irrigated, others_c3nfx_rainfed** | 1 (share of area in a grid cell) | 0.5°×0.5°, annual | downscaled from LUH v2 data based on the crop distribution from (Monfreda et al., 2008). The method is described in (Frieler et al., 2017) |
| Urban areas | **urbanareas** | 1 (share of area in a grid cell) | 0.5°×0.5°, annual | LUH v2 |


Historical land use and irrigation patterns for ISIMIP3a simulations are taken from LUH v2 (Hurtt et al.,
2020, Land use harmonization, 2023). The data set is, up to 2018, identical to the data provided with
ISIMIP2b. The data are based on the HYDE 3.2 land use data set (Klein Goldewijk et al., 2017) and
have been constantly extended up to 2021, i.e., by copying the 2018 patterns into 2019, 2020, and
1078    2021.

The original HYDE 3.2 data distinguishes four categories of land use: rainfed and irrigated cropland,
managed pastures, and more extensively managed rangelands (see **Table 11**). The latter two
categories are combined to grazing lands (ISIMIP variable 'pastures', see **Figure 6**). In LUH v2 the crop
land information is further downscaled to five crop types: C3 annual plants, C3 perennial plants, C3
nitrogen fixing plants, C4 annual plants and C4 perennial plants (see global aggregates in **Figure 7**). In
the same vein as the HYDE case, the LUH v2 data set distinguishes between rainfed and irrigated
croplands. For the purpose of driving the ISIMIP impact models, the LUH v2 data was interpolated from
the original 0.25° × 0.25° to the standard ISIMIP 0.5° × 0.5° global grid. In a further downscaling step
the 5 crops land use data has been downscaled even further to 15 crop types (see global aggregates
in **Figure 7**). For this purpose the Monfreda land use dataset (Monfreda et al., 2008) has been used. It
describes the crop land areas of 175 crops in the year 2000, and we use this to downscale the 5 crops
categories into land use areas of 15 more specific crop types (maize, groundnut, rapeseed, soybeans,
sunflower, rice, sugarcane, pulses, temperate cereals (including wheat), temperate roots, tropical
cereals, tropical roots, others annual, others perennial, and others N-fixing). The ratios determined from
the year 2000 numbers have then been applied to all years. For further details please refer to (Frieler
et al., 2017).

The areas outside of the specified agricultural and urban land is considered 'natural vegetation' and not
prescribed further to not constrain the dynamical vegetation models.

 **4.4 Fertiliser input**

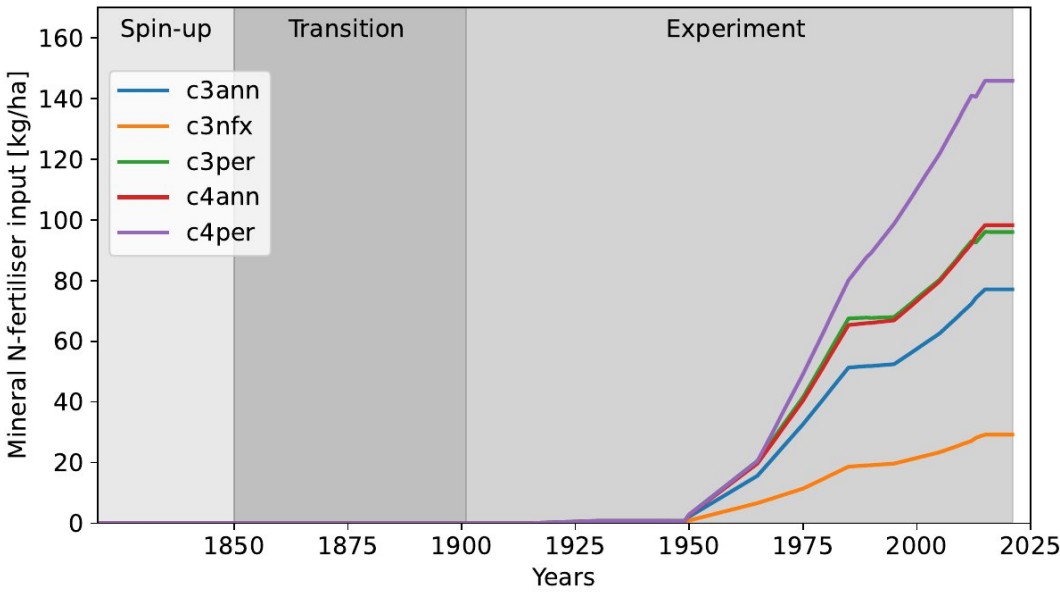

 **Figure 8: Mean mineral N-fertiliser input averaged across the land areas where the considered crop groups are grown.**

 **Table 12: Fertiliser inputs provided as part of the ISIMIP3a direct human forcing.**

| Variable | Variable specifier | Unit | Resolution | Datasets |
|---|---|---|---|---|
| Mineral N-fertiliser input for annual C3 crops C4annual, C4 perennial, C3 nitrogen fixing) | **fertl_c3ann,** | kg ha-1 yr-1 (crop season) | 0.5°×0.5°, annual | LUH v2 (Hurtt et al., 2020) |
| Mineral N-fertiliser input for perennial C3 crops | **fertl_c3per** | kg ha-1 yr-1 (crop season) | 0.5°×0.5°, annual | LUH v2 (Hurtt et al., 2020) |
| Mineral N-fertiliser input for annual C4 crops | **fertl_c4ann** | kg ha-1 yr-1 (crop season) | 0.5°×0.5°, annual | LUH v2 (Hurtt et al., 2020) |

| | | | | |
|---|---|---|---|---|
| Mineral N-fertiliser input for perennial C4 crops | **fertl_c4per** | kg ha-1 yr-1 (crop season) | 0.5°×0.5°, annual | LUH v2 (Hurtt et al., 2020) |
| Mineral N-fertiliser input for nitrogen-fixing C3 crops | **fertl_c3nfx** | kg ha-1 yr-1 (crop season) | 0.5°×0.5°, annual | LUH v2 (Hurtt et al., 2020) |


The LUH v2 data set also includes national application rates of industrial nitrogen fertiliser (Hurtt et al.,
2020). This does not include manure. The fertiliser data is not based on HYDE but was derived from
other sources. The data for the years 1915–1960 are based on (Smil, 2001), 1961–2011 are based on
a compilation by (Zhang et al., 2015) which in turn is based on FAOSTAT (FAO, 2016), and 2012–2015
are based on a projection by the International Fertilizer Association (IFASTAT, 2015). For the pure crop
runs within ISIMIP, where the considered crops are assumed to be grown everywhere without a land
use specification, the LUH v2 national fertiliser inputs are assumed to be applied everywhere within the
country. To calculate crop production, the LUH2 v2 land use patterns are applied in post-processing,
i.e. by multiplying the crop yields from the pure crop run with the land use patterns (fraction of the grid
cell where the crop has been grown).

**4.5 Land transformation**

**Table 13: Land transformation and wood harvest provided as part of the ISIMIP3a direct human forcing.**

| Variable | Variable specifier | Unit | Resolution | Datasets |
|---|---|---|---|---|
| Wood harvest area from primary forest land | **primf-harv** | Fraction of the national land area, kg in case of biomass | Annual, national sum | Based on LUH v2 v2h (Hurtt et al., 2011, 2020; del Valle et al., 2022, Land use harmonization, 2023) |

| Wood harvest area from primary non-forest land | **primn-harv** | Fraction of the national land area, kg in case of biomass | Annual, national sum | Based on LUH v2 v2h (Hurtt et al., 2011, 2020; del Valle et al., 2022) |
|---|---|---|---|---|
| Wood harvest area from secondary mature forest land | **secmf-harv** | Fraction of the national land area, kg in case of biomass | Annual, national sum | Based on LUH v2 v2h (Hurtt et al., 2011, 2020; del Valle et al., 2022) |
| Wood harvest area from secondary young forest land | **secyf-harv** | Fraction of the national land area, kg in case of biomass | Annual, national sum | Based on LUH v2 v2h (Hurtt et al., 2011, 2020; del Valle et al., 2022) |
| Wood harvest area from secondary non-forest land | **secnf-harv** | Fraction of the national land area, kg in case of biomass | Annual, national sum | Based on LUH v2 v2h (Hurtt et al., 2011, 2020; del Valle et al., 2022) |
| Wood harvest biomass carbon from primary forest land | **primf-bioh** | Fraction of the national land area, kg in case of biomass | Annual, national sum | Based on LUH v2 v2h (Hurtt et al., 2011, 2020; del Valle et al., 2022) |
| Wood harvest biomass carbon from primary non-forest land | **primn-bioh** | Fraction of the national land area, kg in case of biomass | Annual, national sum | Based on LUH v2 v2h (Hurtt et al., 2011, 2020; del Valle et al., 2022) |

| | | | | |
|---|---|---|---|---|
| Wood harvest biomass carbon from secondary mature forest land | **secmf-bioh** | Fraction of the national land area, kg in case of biomass | Annual, national sum | Based on LUH v2 v2h (Hurtt et al., 2011, 2020; del Valle et al., 2022) |
| Wood harvest biomass carbon from secondary young forest land | **secyf-bioh** | Fraction of the national land area, kg in case of biomass | Annual, national sum | Based on LUH v2 v2h (Hurtt et al., 2011, 2020; del Valle et al., 2022) |
| Wood harvest biomass carbon from secondary non-forest land | **secnf-bioh** | Fraction of the national land area, kg in case of biomass | Annual, national sum | Based on LUH v2 v2h (Hurtt et al., 2011, 2020; del Valle et al., 2022) |
| Not forest-related land transformations<br><br>All transitions from one type of land use to another | **<type 1>_to_<type 2>**<br><br>With type 1 and type 2 from the following list:<br><br>**secdf** (potentially forested secondary land),<br>**secdn** (potentially non-forested secondary land),<br>**urban** (urban land),<br>**c3ann** (C3 annual crops),<br>**c4ann** (C4 annual crops),<br>**c3per** (C3 perennial crops),<br>**c4per** (C4 perennial crops),<br>**c3nfx** (C3 nitrogen-fixing crops),<br>**pastr** (managed pasture)<br>**range** (rangeland) | Fraction of the grid cell | Annual | Based on LUH v2h (Hurtt et al., 2011, 2020, Land use harmonization, 2023);<br><br>Land is considered to be 'potentially forested' if the above ground biomass density (kg C m−2) of the potential vegetation as estimated by the Miami-LU model accounting for changes in cropland and grazing land is > 2 kg C m−2 |

|  |  |  |  | (Hurtt et al., 2020) |
|---|---|---|---|---|
|  |  |  |  |  |


These datasets are based on the LUH v2 Harmonization Data Set covering 850 to 2015 (Hurtt et al.,
2020, Land use harmonization, 2023). The wood harvest data were obtained by aggregating from the
original LUH v2 grid to the ISIMIP 0.5° × 0.5° grid (first-order conservative remapping) and then
aggregating to the national sums. Wood harvesting data are used in the vegetation models to mimic
wood removal as part of forest management and clearing, and has a strong influence on the carbon
balance. National data are provided so that models can use their internal routines to distribute the
harvesting within a country's forest area. The gridded land transformation data were obtained by
aggregating from the original LUH v2 grid to the ISIMIP 0.5° × 0.5° grid; these data always end a year
earlier than all other land use data, because a year in these data sets actually describes the changes
from the current to the next year. The data have been extended up to 2021 by copying the 2015 data
into the following  years (files end in 2020).

**4.6 Nitrogen Deposition**

**Table 14: Nitrogen deposition provided as part of the ISIMIP3a direct human forcing.**

| Variable | Variable specifier | Unit | Resolution | Datasets |
|---|---|---|---|---|
| Reduced nitrogen deposition | **nhx** | g N m-2 mon-1 | monthly | based on simulations from (Tian et al., 2018) |

| Oxidised nitrogen deposition | **noy** | g N m-2 mon-1 | monthly | based on simulations from (Tian et al., 2018) |


Reduced and oxidised nitrogen deposition (NHx, NOy) are based on simulations by the NCAR
Chemistry-Climate Model Initiative during 1850-2014 (Tian et al., 2018). Nitrogen deposition data was
interpolated to 0.5° by 0.5° using the nearest grid point method. Data in 2015-2021 are assumed to be
the same as that in 2014.

**4.7 Crop calendar**

**Table 15:** Crop calendar provided as optional representation of agricultural management. The information is
given for 18 crop types.

| Variable | Variable specifier | Unit | Resolution | Datasets |
|---|---|---|---|---|
| Planting day, separated for rainfed and irrigated crops where applicable | **planting_day** | day of year | 0.5°, time average, no variation in time | (Jägermeyr et al., 2021b) |
| Maturity day, separated for rainfed and irrigated crops where applicable | **maturity_day** | day of year | 0.5°, time average, no variation in time | (Jägermeyr et al., 2021b) |


Unfortunately, there is no global data set describing changes in growing seasons across the historical
period. Instead we provide a static crop calendar that has been developed within the AgMIP Global
Gridded Crop Model Intercomparison GGCMI and merges information from various observational data
sources (Jägermeyr et al., 2021b). It provides planting and maturity days for 18 different crops at the
ISIMIP standard 0.5° grid. Grid cells outside of currently cultivated areas are spatially extrapolated
(details below). For wheat and rice two growing seasons are provided while for all other crops the
calendar only specifies one main growing season. The reported growing seasons should not be
considered the growing seasons for one specific year but as 'representative growing season' across
the recent years. Within the crop models different crop varieties are represented by different heat units
required to reach physiological maturity. The crop calendar should be implemented by adjusting the
required heat units to the average of the annual sums of heat units between the specified planting and
maturity date over all growing seasons between 1979 and 2010.
If modellers use a temporal adjustment of cultivars by varying required heat units in response to socio-
economic development or historical climate change this is certainly allowed within the 'histsoc' set-up.
If cultivars are fixed according to the method described above this simulation will be considered a
'2015soc' simulation as long as other direct human drivers are also held constant at 2015 levels.
However, if, e.g., fertiliser inputs are varied over time according to provided forcing data (see section
**4.4**), the run will be considered a 'histsoc' run.
GGCMI is currently working on a temporally resolved global crop calendar at the same spatial resolution
based on various new data sources including agricultural ministries, census reports, phenological data
bases, experimental sites, etc. This data set will be published separately and could then be used to
inform 'histsoc' simulations.

**4.8 Dams and reservoirs**

**Table 16: Information about dams and reservoirs**

| Variable | Variable specifier | Unit | Resolution | Datasets |
|---|---|---|---|---|
| Unique ID for each point representing a dam and its associated reservoir. | **ID** | unitless numbers: 1-7320 from GRanD and J3-J26 from GeoDAR v1.2 | per dam | Global Reservoir and Dam Database (GRanDv1.3, data up to 2016; (Lehner et al., 2011a, b) and GeoDAR v1.2 (Wang et al., 2022) covering the period 2016-2020 |
| Name of the dam structure | **DAM_NAME** | unitless | per dam | GRanDv1.3, GeoDARv1.2 |
| Original longitudinal location of the dam | **LON_ORIG** | degree (°) | per dam | GRanDv1.3, GeoDARv1.2 |
| Original longitudinal location of the dam | **LAT_ORIG** | degree (°) | per dam | GRanDv1.3, GeoDARv1.2 |

| Longitude, adjusted to the ISIMIPddm30 0.5° grid cell centres | LON_DDM30 | degree (°) | per dam | Adjustment of original GRanDv1.3, GeoDARv1.2 data |
|---|---|---|---|---|
| Latitude, adjusted to the ISIMIPddm30 0.5° grid cell centres | LAT_DDM30 | degree (°) | per dam | Adjustment of original GRanDv1.3, GeoDARv1.2 data |
| Upstream area draining into the reservoir using ISIMIPddm30 | CATCH_SKM_DDM30 | km² | per dam | Derived from dam location and the ISIMIPddm30 drainage map. |
| Upstream area draining into the reservoir acc. to GRanD [km²] | CATCH_SKM_GRanD | km² | per dam | GRanDv1.3 |
| Representative maximum storage capacity of reservoir | CAP_MCM | $10^6$ m³ | per dam | GRanDv1.3, GeoDARv1.2 |
| Year of construction, completion, commissioning, etc. (not specified) | YEAR | year | per dam | GRanDv1.3, GeoDARv1.2 + complemented by internet research |
| Alternative year (may indicate multi-year construction, secondary dam, etc.) | ALT_YEAR | year | per dam | GRanD |
| Original, rounded location has been shifted with automatic mapping (FLAG_CORR=1) | FLAG_CORR | Unitless labels: 1 or 2 | per dam | Introduced when adjusting the locations to the ISIMIPddm30 |

| | | | | |
|---|---|---|---|---|
| If visual check or manual re-location has been applied (FLAG_CORR=2) | | | | 0.5° grid |
| Name of the river which the dam impounds | **RIVER** | unitless | per dam | GeoDARv1.2. For GRanD records, it can be found in the GRanD database |
| Country where the dam is located | **COUNTRY** | unitless | per dam | GeoDARv1.2. For GRanD records, it can be found in the GRanD database |
| Height of the dam.<br><br>If multiple heights are available, the foundation height was used. | **D_Hght_m** | m | per dam | GeoDARv1.2. For GRanD records, it can be found in the GRanD database |
| Maximum inundation area of the reservoir | **R_Area_km2** | km² | per dam | GeoDARv1.2. For GRanD records, it can be found in the GRanD database |
| Maximum inundation length of the reservoir | **R_Lgth_km** | km | per dam | GeoDARv1.2. For GRanD records, it can be found in the GRanD |

| | | | | database |
|---|---|---|---|---|
| Main purpose(s) of the dam | **PURPOSE** | no units | per dam | GeoDARv1.2. For GRanD records, it can be found in the GRanD database |
| Sources used to collect this dam's information | **SOURCE** | no units | per dam | GeoDARv1.2. For GRanD records, it can be found in the GRanD database. If filled out for GeoDAR records, it corresponds to the source for the year of construction/ commissioning |
| Other notes related to the mapping or re-location of dams to ISIMIPddm30 | **COMMENTS** | no units | per dam | |


In order to offer a consistent and common source of information about reservoirs and associated dams
for climate impact modellers (see **Table 16**), we joined the Global Reservoir and Dam Database of the
Global Water System Project (GRanD v1.3; (Lehner et al., 2011a, b) with a subset of the Georeferenced
global Dams And Reservoirs (GeoDAR v1.2) database (Wang et al., 2022), developed at Kansas State
University (KSU), and provided by Jida Wang ahead of publication, so that it could be provided when
launching ISIMIP3 in 2020. These additional dams have construction or projected finalisation dates
between 2016 and 2025, while GRanD v1.3 includes dams constructed up until 2017. In total, the
combined database now includes 7331 dams whose construction will be finished by 2025. It includes
dams that were constructed before the simulation period, but still exist (the first reported dam was
finished in the year 286). For the simulations described here, dams with (projected) construction dates
after 2020 are not considered; these will become relevant in the ISIMIP3b simulations, with exception
of the Grand Ethiopian Renaissance Dam, which we decided to include since its reservoir reached a
first stage of filling of 4.9 km$^3$ in July 2020 (BBC news: Nile dam row, 2020; Tractebel: Filling of the
reservoir of the Grand Renaissance Dam, 2020).

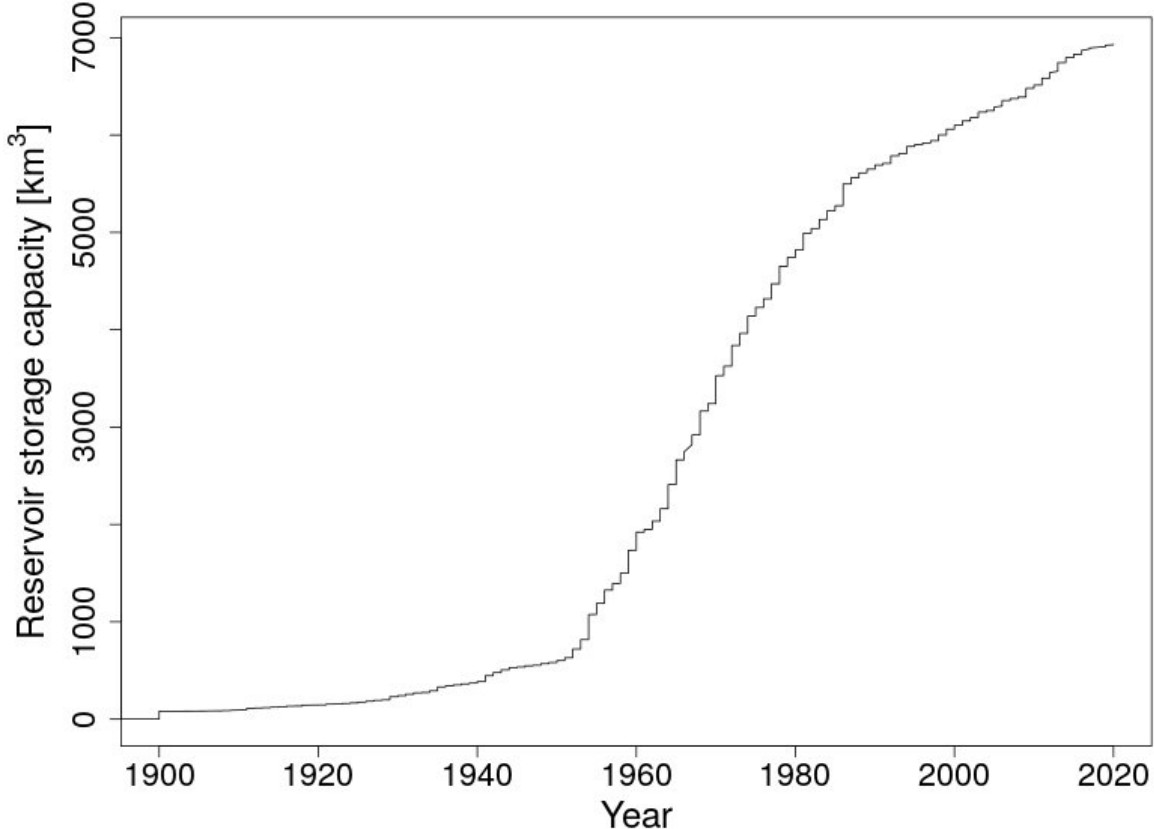

**Figure 9: Cumulative reservoir storage capacity between 1900 and 2020.** Reservoirs that are active before
the year 1901 have been assigned to the year 1900. Horizontal axis shows year of construction, completion, or
commissioning, reflecting ambiguity in available data.

The original GRanDv1.3 dam locations were mapped to the global 30-min drainage direction map
(ISIMIPddm30, (Müller Schmied, 2022) based on DDM30 (Döll and Lehner, 2002), by applying the
following algorithm:
Firstly, the locations have been rounded to the closest 0.5° grid cell centre. Then, the area of the
upstream catchment draining into the GRanD reservoirs (previous version of GRanDv1.3) in the
ISIMIPddm30 map have been calculated and compared against the ones reported in GRanD. All dams
with an upstream area bigger than 10000 km$^2$ in GRanD and more than 50% deviation from the GRanD
upstream area have been shifted to the 8 possible neighbouring cells. If any of these shifts resulted in
a smaller deviation from the GRanD upstream areas, the dam was moved to the grid cell resulting in
the smallest deviation in the upstream area.
Additionally, a visual validation and, where appropriate, manual relocation were applied with the aim to
find the best fitting grid cell from a hydrological perspective. Due to the low resolution of the model grid,
reservoirs might get wrongly assigned to e.g. the main stream (either before or after the confluence of
two rivers), even though the dam is located in a particular tributary according to the database.
In those cases, and based on visual GIS inspection, the best location was searched, e.g. by moving the
dam location one cell upstream to preserve the routing order and to avoid a different or much deviating
river basin in the ISIMIPddm30 stream network. In case a dam is not assigned to any river basin in the
ISIMIPddm30 (which can happen due to the difference in spatial resolution), the most suited location
according to the observed upstream area was selected. Because of limited capacity, this visual
validation procedure was applied only for dams present in the earlier GranDv1.1 version that have a
maximum storage capacity greater than 0.5 km$^3$ (1108 dams), as well as for all the 458 additional dams
in GRanDv1.3 and the 11 dams (excluding post-2020 dams) added from GeoDAR v1.2, and not for
several thousand smaller dams present in GranDv1.1. In total the reported dams have a global
cumulative storage capacity of approximately 6932 km$^3$ (**Figure 9**).
**4.9 Fishing intensities**

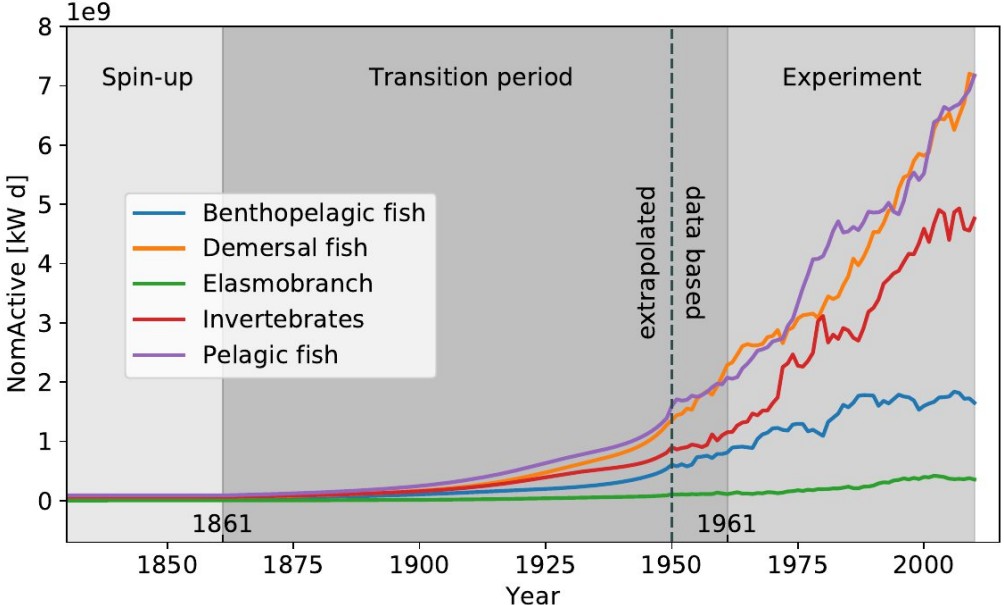

**Figure 10: Evolution of historical nominal active fishing effort** (NomActive) as provided for the spin-up,
transition period, and 'obsclim + histoc, default' ISIMIP3a experiment, separated by target functional group. The
groups represent an aggregation of 29 even finer categories covered by the data set (see **Table 17**).
**Table 17: Information about historical fishing intensities provided as DHF within ISIMIP3a**. For the spin-up
+ transition period required by models within the *marine ecosystems and fisheries* sector the forcing is provided
for 1841-2010 although the 'obsclim + histoc, default' experiment only starts in 1961.

| Variable | Variable specifier | Unit | Resolution | Datasets |
|---|---|---|---|---|
| Total nominal active fishing effort (i.e., accounting for total | **NomActive** | kW d (kilowatts of fleet | annual data spatially grouped by Exclusive Economic Zones | Reconstruction based on historical yearbook and  FAO |

| | | | | |
|---|---|---|---|---|
| power of the fleet but not including changes in the efficiency of fishing technology) separated by fishing sector, fleet, and target functional groups. | | power times days at sea) | (EEZ), (Sea Around Us Area Parameters and Definitions) and nested within Large Marine Ecosystems. Masks for the latter are provided as static geographic information (see **Table 1**). | compilations ((Rousseau et al., 2022) based on (Rousseau et al., submitted 2023). The reconstructions have been extended backwards to 1841 by constant 1861 values to cover the 120 years of spin-up required for the marine ecosystems and fisheries models |

The data set of reconstructed historical fishing efforts (Rousseau et al., 2022) serves as the DHF for the *marine ecosystems and fisheries* sector. The efforts are quantified for 'artisanal' and 'industrial' fishing (sector), 66 Large Marine Ecosystems (LME), 187 national Exclusive Economic Zones (EEZ) and 'high seas', 244 country identifiers from the Sea Around Us Project (SAUP), 16 different categories of applied gears (e.g. bottom trawls, longlines and purse seines), 29 target functional groups (see nominal active fishing effort for 5 aggregated categories in **Figure 10**), separately.

The original  annual time series spanning 1950-2015 were further extrapolated into the past to 1861 using generalised additive models (Rousseau et al., submitted 2023; see **Figure 10**). To cover the 'spin-up + transition' period from 1841-1960 the data set has been extended backwards by 1861 values. Forcing with this dataset allows for a comparison of simulated catches  against the congruent (Watson, 2019) reconstruction of historical fisheries catches (spanning the period 1869-2015; (Watson and Tidd, 2018). To permit integration into marine ecosystem models that capture different fishing sectors, fleets, and functional groups these data include nominal active fishing effort disaggregated by location (Exclusive Economic Zone/High Seas and Large Marine Ecosystem), fishing country, fishing gear, targeted functional group, and fishing sector (coastal artisanal and industrial). Impact modellers are allowed to distribute this effort across space, time, and target organisms in any method compatible with their models' structure. The fishing effort data does not include any information about changes in the efficiency of fishing technology over time (technological creep). Assumptions about these efficiencies are left to the individual modellers and usually determined in model calibration.

### 4.10 Forest management for *regional forest* sector

**Table 18:** Information about historical forest management provided as DHF for the *regional forest* sector within ISIMIP3a

| Variable | Variable specifier | Unit | Resolution | Datasets |
|---|---|---|---|---|
| Silvicultural system | **sysi** | na | stand | (Reyer et al., 2023) |
| Tree species | **species** | na | stand | (Reyer et al., 2023) |
| Harvest type | **harvtype** | na | stand | (Reyer et al., 2023) |
| Thinning type | **thintype** | % of basal area | stand | (Reyer et al., 2023) |
| Rotation length | **rotlength** | year | stand | (Reyer et al., 2023) |
| Thinning frequency | **thinfrequ** | year | stand | (Reyer et al., 2023) |
| Year of Management intervention | **manyear** | year | stand | (Reyer et al., 2023) |
| Type of management intervention | **mantype** | na | stand | (Reyer et al., 2023) |
| Regeneration species | **regen** | na | stand | (Reyer et al., 2023) |
| Planting density | **plantdens** | na | stand | (Reyer et al., 2023) |
| Planting age | **plantage** | year | stand | (Reyer et al., 2023) |
| Planting seedling height | **planthei** | m | stand | (Reyer et al., 2023) |
| Planting diameter at breast height | **plantdbh** | cm | stand | (Reyer et al., 2023) |
| Age when diameter at breast height is | **dbhage** | year | stand | (Reyer et al., 2023) |

| reached | | | | |
|---|---|---|---|---|
| Stem number | **stemno** | na | stand | (Reyer et al., 2020a) based on (Reyer et al., 2020b) |

For the *regional forest* sector, forest management is defined for nine forest sites in Europe, four in Germany (Peitz, KROOF, Solling-beech, Solling-spruce) as well in Czech Republic (Bily Kriz), Denmark (Sorø), France (Le Bray), Italy (Collelongo) and Finland (Hyytiälä) (Reyer et al., 2020b). Additionally, a set of forest site-specific forest management rules and planting numbers based on historical standard management practices of the area where the forest sites are located are defined and spelled out in concrete management schedules to enable modellers to simulate '2015soc' conditions (Reyer et al., 2023). The regional forest management data has not been harmonised to the global gridded wood harvest data provided for the biomes sector, because the data is very site-specific and the variation not resolved in the global data set.

**5 Conclusion**

The first part of the third simulation round of the Inter-sectoral Impact Model Intercomparison Project ISIMIP (ISIMIP3a) is intended to facilitate impact model evaluation and impact attribution experiments to significantly move forward our understanding of observed changes in natural and human systems and their respective drivers. Impact models as participating in ISIMIP encode our process knowledge on how several drivers (climate-related ones as well as direct human influences) come together to generate observed changes. As such, they are ideal tools for this task. The new ISIMIP3a simulation framework including the provision of the relevant forcing data is intended to unleash the power of a wide range of models from different sectors to quantify the contribution of observed changes in climate-related systems to observed environmental or societal changes.

As a first step towards impact attribution, the ISIMIP3a evaluation experiments will help to clarify how well the current generation of impact models can explain observed changes in impacted systems based on provided information about the different forcings. The performance of the models in reproducing observed variations and long-term changes in the impacted systems, certainly does not only depend on the models themselves but also on the availability and uncertainties associated with the climate-related and direct human forcings (see Table **1**). We capture part of this uncertainty by providing four different observational atmospheric climate forcing data and associated counterfactual forcings (see section **2.1**) and TC windfields derived from two different modelling approaches (see section **3.2**). Uncertainties in the direct human forcings are represented to the degree that the forcing data sets considered as 'optional' vary from model to model. In addition, the multi-model framework of ISIMIP allows for testing to what degree different process-representations may be better suitable to explain the observations than others.

High explanatory power is then a prerequisite for impact attribution through the ISIMIP3a attribution
experiments based on counterfactual climate-related forcings following the IPCC definition (O'Neill et
al., 2022).
The setup is the first that allows to easily and broadly address impact attribution across many impact
categories. This will fill an important gap as only few process-based impact models have been used in
this field despite their general suitability. The presented work can thus lay the ground for urgently
necessary works to inform climate litigation (Burger et al., 2020; Burger and Tigre, 2023), the loss and
damage debate (Mechler et al., 2018; Wyns, 2023), and last but not least also decisions about short
term adaptation measures. It will ultimately help to carve out the sensitivity of our ecosystems and
human societies to historical climate change, which is a precondition for robustly projecting future
climate impacts.
This paper aims to give an overview of the ISIMIP3a experiments and the provided climate-related and
direct human forcing data sets. It is intended to work as a catalogue where modellers can find all
relevant information about the data sets they need for the impact model simulations within ISIMIP3a.
As a community-driven initiative across multiple disciplines the selection of the best available forcing
data for ISIMIP builds on the expertise within the different sectoral communities.
We would like to improve or complement these data sets in a continuous process wherever possible.
So this paper can also be read as a call for contributing additional data that could i) be provided within
the current round (ISIMIP3) as optional data (see explanation in the introduction) that is not harmonised
within or across sectors or ii) as mandatory forcing for an upcoming simulation round. In particular, we
aim for temporally resolved historical growing seasons that have been shown to be critical to reproduce
observed crop yields (Jägermeyr and Frieler, 2018), counterfactual oceanic climate-related forcings,
counterfactual TC-related precipitation (Risser and Wehner, 2017; van Oldenborgh et al., 2017; Wang
et al., 2018; Patricola and Wehner, 2018), temporally resolved lightning data for the full set of considered
climate model simulations, and temporally resolved human drainage and restoration activities in
peatlands as one of the key controls over global peatland greenhouse gas emissions (Loisel et al.,
1316 2020).
**Author contribution:** KF lead the project and developed the concept with contributions from JS, MM,
CO, CPOR, JLB, CSH, CMP, TDE, KOC, CN, RH, DT, OM, JJ, GL, SC, EB, AGS, NS, JC, SH, CB, AG,
FL, SNG, HMS, FH, TH, RM, DP, WT, DMB, MB. JV supported the data generation and harmonisation
of the protocol across all sectors. SL provided atmospheric climate forcing data. MM provided coastal
water level data and atmospheric forcing data. MdRRL, JW and FY provided dam data. CO and IJS
provided GDP data. CPOR provided forest management data. DNK and JTM provided high resolution
climate forcing data. ST provided coastal water levels and counterfactual climate forcing data. YR
provided data on fishing efforts. CS and XL provided ocean forcing data. TV provided TC data. TW and
FS provided gridded GDP data. IV provided lake data. JJ provided growing seasons. CM provided soil
data. KF prepared the manuscript with contributions from all co-authors.

**Code and data availability:** All input data described is available for participating modelers with a
respective account at the DKRZ server. Data will be made publicly available, and most data is already
publicly available at https://data.isimip.org/. Availability is documented on www.isimip.org where the
way of accessing the data is described, as well. Model output is already partly available at
https://data.isimip.org/.
The ISIMIP Repository fulfills the Archive standards as stated in the "GMD code and data policy". The
Repository is hosted and maintained by the Potsdam Institute for Climate Impact Research (PIK). Data
can only be published or removed from the repository by the ISIMIP data team, that is monitored by the
ISIMIP steering committee according to the organisational structure of ISIMIP (ISIMIP organigram,
2020). DOI are used to refer to datasets in a persistent way. Whenever a dataset is replaced for any
reason a copy is kept on tape, and a new DOI is issued, while the old DOI is kept online with information
on how to retrieve the archived data. Detailed information can be found in the ISIMIP terms of use
(ISIMIP terms of use, 2023).

**Competing interests:** At least one of the (co-)authors is a member of the editorial board of
Geoscientific Model Development.

**Acknowledgements**
This article is based upon work from COST Action CA19139 PROCLIAS (PROcess-based models for
CLimate Impact Attribution across Sectors), supported by COST (European Cooperation in Science
and Technology; https://www.cost.eu). Funding from the EU Horizon 2020 research and innovation
program under grant agreement 821010 (CASCADES) supported the work of C.P.O.R., J.V. and
I.D.VdV., and the provisioning of the high resolution climate data, and supported the work of ST under
grant agreement No 820712 (RECEIPT). SL received funding from the German Research Foundation
(DFG, project number 427397136). The German Federal Ministry Ministry of Education and Research
(BMBF) supported the work under the research projects ISIAccess (16QK05), SLICE (01LA1829A),
QUIDIC (01LP1907A), CHIPS (01LS1904A), ISIpedia (01LS1711A). FL received funding from the
National Key Research and Development Program of China (project number 2022YFE0106500). JC
received funding from the National Key Research and Development Program of China (project number
2022YFF0801904). MB acknowledges funding from the Research Foundation—Flanders (FWO,
G095720N). SC and NS were supported by the National Environmental Research Council (NERC)
grant NE/R015791/1. SC, AGS, MB and NS acknowledge funding through NERC NE/V01854X/1
(MOTHERSHIP). CB was supported by the Newton Fund through the Met Office. The research by DNK
was funded through the 2019-2020 BiodivERsA joint call for research proposals, under the BiodivClim
ERA-Net COFUND program, and with the funding organisations Swiss National Science Foundation
SNF (project: FeedBaCks, 193907), and the Swiss National Science Foundation SNF (project: Adohris,
205530). RM was supported by the Alter-C project (PID2020-114024GB-C32/AEI
/10.13039/501100011033). CSH was supported by Open Philanthropy, NSF award 2218777, and
NOAA CPO. We also thank Jason Evans for his positive review, and an anonymous reviewer for an
extremely careful and comprehensive review that contributed significantly to the improvement of the
paper.

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
