# Peer review of "Scenario set-up and forcing data for impact model evaluation and impact attribution within the third round of the Inter-Sectoral Model Intercomparison Project (ISIMIP3a)"

_EGUsphere, 2023_

## Referee Comment (RC2)

**Anonymous review of Preprint egusphere-2023-281 Scenario set-up and forcing data for impact model evaluation and impact attribution within the third round of the Inter-Sectoral Model Intercomparison Project (ISIMIP3a) By Frieler et al.**

This long manuscript describes the set-up of ISIMIP3a, corresponding to historical simulations within the third round of the Inter-Sectoral Impact Model Intercomparison Project (ISIMIP). For a decade now, ISIMIP has been a very important contribution to the assessment of climate change impacts, gathering a huge community of modellers and users, so I recommend this paper to be published in GMD. Yet, I also recommend some revisions beforehand to clarify the set-up, make this long paper more easy to read, and also to enhance the scientific relevance of the paper, which looks a bit too much like a technical report in its present form. I worked hard to make full sense of this manuscript, and I hope the comments below can help the authors improve it. I also attach a annotated version of the submitted pdf.

**Introduction**

With the exception of the final paragraph devoted to the outline of the paper, the Introduction only addresses the ISIMIP framework. This part could be made more informative to non ISIMIP specialists by clarifying the different rounds (now third) with references, and how the 'b' runs differ from the 'a' runs (L101). The different sectors which are addressed by ISIMIP models could also be reminded.

The innovation of ISMIP3a compared to ISMIP2a is to offer a well-thought framework for « climate impact attribution », but this expression is not defined before section 2.2, p22 ; I suggest to bring this part into the Introduction, and to develop a bit this discussion and the references. There is only one, O'Neill et al. (2022), thus omitting the seminal papers of Cramer et al. (2014) and Hansen et al. (2016), and a few newer ones probably.

In doing so, I think the authors could better distinguish two drivers of the changes in human-impacted sectors since 1901: past climate change vs other anthropogenic pressures. If I understood correctly, ISIMIP3a intends to quantify how much of past trends can be attributed to past climate change ; this is the climate impact attribution, using factual and counterfactual climate related forcing (CRF). But ISIMIP3a also offers counterfactual direct human forcing (DHF), which are planned to be used in sensitivity experiments, but could also support attribution : contrasting factual and counterfactual DHFs should allow to quantify how much of past trends can be attributed to past DHF. This strategy has long been explored in the sector of hydrology and water resources at various scales, for various DHFs, with more or less quantification (e.g. Alkama et al., 2010; Sterling et al., 2013; Tramblay et al., 2021; Teuling et al., 2019; Vicente-Serrano et al., 2019) and ISIMIP3a would be a great opportunity to consolidate a comprehensive framework to attribute past trends to CRFs, DHFs, and their interactions. If too late to adapt the simulation set-up, this generalization could at least be discussed in Conclusion as a perspective. In this case, I suggest that the title is changed to « climate impact attribution » instead of « impact attribution ».

If space allows, I would also recommend to quickly compare ISIMIP to other important MIPs, especially the ones which address climate change impacts and climate impact attribution (IPCC since Cramer et al., 2014; LUMIP, sharing many DHF datasets with ISIMIP; maybe TRENDY, etc.).

**Paper structure and use of tables**

The reading is difficult because of the many long tables. We can distinguish three kinds of tables :

- Tables 1 and 3 describe the datasets, factual and counterfactual respectively ;
- Tables 2 and 4 describe the experiments: evaluation and sensivity expriments in Table 2, and climate impact attribution experiments in Table 4 ;
- Tables 5 to 18 detail the variables which are included in the datasets.

The latter ones are usually not cited in the text (true for 6, 7, 9, 11, 12, 13, 14, 15, 18, while Table 17 is only cited in the caption of Figure 10) and could conveniently be placed in annex or supplementary material without making the understanding more difficult, since the most important aspects of these tables are summarized by some text. The related figures, not cited either in the text for most of them, can either be kept in the main text or put in annex or supplementary, but should be cited.

The structure of section 2, which describes the experiments, could probably be made more efficient by better articulating the sub-sections, Figure 1, and Tables 1 to 4. Please consider the following suggestions :

- Put the first part of section 2 under a numbered subsection 2.1 devoted to the ISIMIP3a rationale, based on Figure 1.
- Split present subsection 2.1 « Model evaluation and sensitivity experiments » in two different subsections : one for the evaluation experiments, linked to Table 1 (datasets for factual experiments, and to the first part of Table 2), and one for the sensitivity experiments, as described in Table 2 starting at CO2 sensitivity under 'histsoc'. This part of Table 2 should refer to the subsections in which the used DHF datasets are described (within section 4).
- It wouldn't hurt to do the same for the CRF datasets. To this end, I also suggest to split section 3.1 « Observational atmopsheric climate forcing data (factual and counterfactual, starting at bottom of p 24) » in three different subsections : one for the factual (default) dataset, one for counterfactual dataset, and one for the high resolution factual datasets. The paragraphs already exist, so it's easy, and it would be useful to refer to the description of these datasets inside the tables.
- If the authors accept the above suggestions, section 2.2 on the climate impact attribution experiments would become 2.4, linked to Table 3 (counterfactual CRF) and Table 4 (attribution experiments).
- The last paragraph of the Introduction could also be reshaped to introduce the different kinds of experiments, and I wonder if Tables 1 to 4 wouldn't be worth being mentioned here. I'm not sure, however, that the list of all the subsections of section 3 (CRF) and 4 (DHF) is particularly useful here.

**Other comments**

- The link to get the ISIMIP3a protocol should be given in the paper
- What is the difference between « default » and « mandatory » and « 1st priority »? Another term in the tables is « optional » : does it mean it is not mandatory, is it « 2nd priority », or something else ?
- Many Tables and Figures are not referred in the text (Figures; Tables)
- Many datasets are detailed in submitted papers, which may end up being a problem if these papers are not accepted.
- I wonder about the use of atmospheric composition, since there is no dataset about aerosols, ozone, or N2O. I guess that vegetation models use CO2, but some may use ozone as well. And which models need CH4 as input? A few words about the models' need by sector could help here.
- Riverine inputs : section 3.4 explains that the time-varying riverine input dataset was produced by the LM3-TAN land model, and I guess they depend on fertilizers. In such a case, they result from both CRF and DHF. This might be acknowldeged. At L844, we read that no counterfactual riverine input dataset is provided : but isn't the 1955 dataset counterfactual ?
- Section 4 mentions group I, II and III simulations which are never defined (L939, L1051)
- In section 4.3, land use is limited to agriculture : why not account for urban areas, especially given that Table 13, p160, includes conversion to urban ? what about planted trees?
- Harmonization would be welcome for LUH2 vs LUH v2.
- The first sentence of subsection 4.9 (L1095), which relates the dataset to a sector of ISIMP, is extremely informative, and could serve as an example for all subsections.
- The last section should be a Conclusion and not a Discussion

**New references**

Alkama, R., Kageyama, M., and Ramstein, G. (2010), Relative contributions of climate change, stomatal closure, and leaf area index changes to 20th and 21st century runoff change: A modelling approach using the Organizing Carbon and Hydrology in Dynamic Ecosystems (ORCHIDEE) land surface model, *J. Geophys. Res.*, 115, D17112, doi:10.1029/2009JD013408

CRAMER, Wolfgang, YOHE, G., HOUSE, Joanna Isobel, *et al.* Detection and attribution of observed impacts: Climate Change 2014: impacts adaptation and vulnerability, contribution of working group ii to the fifth assessment report of the intergovernmental panel on climate change. In : *Climate Change 2014: Impacts Adaptation and Vulnerability, Contribution of Working Group II to the Fifth Assessment Report of the Intergovernmental Panel on Climate Change*. Cambridge University Press, 2014.

Gerrit Hansen, Daithi Stone, Maximilian Auffhammer, Christian Huggel, Wolfgang Cramer (2016). Linking local impacts to changes in climate: a guide to attribution. Regional Environmental Change, 16, 527–541. https://doi.org/10.1007/s10113-015-0760-y

Sterling, S., Ducharne, A. & Polcher, J. The impact of global land-cover change on the terrestrial water cycle. *Nature Clim Change* **3**, 385–390 (2013). https://doi.org/10.1038/nclimate1690

Tramblay, Y., Villarini, G., El Khalki, E. M., Gründemann, G., & Hughes, D. (2021). Evaluation of the drivers responsible for flooding in Africa. *WaterResources Research*, *57*, e2021WR029595. https://doi.org/10.1029/2021WR029595

Teuling, A. J., de Badts, E. A. G., Jansen, F. A., Fuchs, R., Buitink, J., Hoek van Dijke, A. J., and Sterling, S. M.: Climate change, reforestation/afforestation, and urbanization impacts on evapotranspiration and streamflow in Europe, Hydrol. Earth Syst. Sci., 23, 3631–3652, https://doi.org/10.5194/hess-23-3631-2019, 2019.

Vicente-Serrano, S. M., Peña-Gallardo, M., Hannaford, J., Murphy, C., Lorenzo-Lacruz, J., Dominguez-Castro, F., et al. (2019). Climate, irrigation, and land cover change explain streamflow trends in countries bordering the Northeast Atlantic. *Geophysical Research Letters*, 46, 10821–10833. https://doi.org/10.1029/2019GL084084

**Scenario set-up and forcing data for impact model evaluation and impact attribution within the third round of the Inter-Sectoral Model Intercomparison Project (ISIMIP3a)**

5

Katja Frieler1, Jan Volkholz1, Stefan Lange1, Jacob Schewe1, Matthias Mengel1, María del Rocío Rivas López1, Christian Otto1, Christopher P.O. Reyer1, Dirk Nikolaus Karger2, Johanna T. Malle2, Simon Treu1, Christoph Menz1, Julia L. Blanchard3, Cheryl S. Harrison4, Colleen M. Petrik5, Tyler D. Eddy6, Kelly Ortega-Cisneros7, Camilla

- 10 Novaglio3, Yannick Rousseau3, Reg A. Watson3, Charles Stock8, Xiao Liu9, Ryan Heneghan10, Derek Tittensor11, Olivier Maury12, Matthias Büchner1, Thomas Vogt1, Tingting Wang13, Fubao Sun13, Inga J. Sauer1,14, Johannes Koch1, Inne Vanderkelen15, 16, 17, Jonas Jägermeyr1,18, 19, Christoph Müller1, Jochen Klar1, Iliusi D. Vega del Valle1, Gitta Lasslop20, Sarah Chadburn21, Eleanor Burke22, Angela Gallego-
- 15 Sala23, Noah Smith21, Jinfeng Chang24, Stijn Hantson25, Chantelle Burton22, Anne Gädeke1, Fang Li26, Simon N. Gosling27, Hannes Müller Schmied20, 28, Fred Hattermann1, Jida Wang29, Fangfang Yao30, Thomas Hickler20, Rafael Marcé31,32, Don Pierson33, Wim Thiery15, Daniel Mercado-Bettín31, Matthew Forrest20, Michel Bechtold34
- 20

Affiliations:

[revised manuscript text omitted]

---

## Author Response (AR1)

**ISIMIP3a paper author replies**

**RC1**

*Technical corrections:*

- line 666: the word "one" should be deleted.

*Reply:*

Dear Jason Evans, thanks so much for your efforts and your positive review. We addressed your technical correction in the revised version of the manuscript.

**RC2**

We would like to thank the reviewer for the extremely careful review that has helped us to significantly improve the paper. The review must have been an enormous effort given the length of the paper. We can only express our gratitude for the patience!

**General comments:**

Introduction:
- With the exception of the final paragraph devoted to the outline of the paper, the Introduction only addresses the ISIMIP framework. This part could be made more informative to non ISIMIP specialists by clarifying the different rounds (now third) with references, and how the 'b' runs differ from the 'a' runs (L101). The different sectors which are addressed by ISIMIP models could also be reminded.

Thank you very much for the suggestion. We have modified the introduction by providing a more explicit explanation of ISIMIP's 'a' and 'b' parts.
*"Initialised in 2012, ISIMIP is organised in individual modelling rounds. The decision about their design and the development of the associated simulation protocols has been developed into an iterative process between stakeholders and users of ISIMIP data, the sectoral coordinators representing participating modelling teams, the Scientific Advisory Board, and the Cross-Sectoral and Coordination Team at PIK (ISIMIP Coordination Team, Sectoral Coordinators, Scientific Advisory Board, 2018). Since its second round the ISIMIP protocols comprise an 'a' part describing impact model simulations that cover the historical period forced by observational climate-related and direct human forcings (evaluation set-up), and a 'b' part dedicated to impact simulations based on simulated climate-related forcings including future projections. This paper describes the ISIMIP3a simulation framework only. Compared to ISIMIP2a the evaluation set-up based on observational forcing data has been extended to now include additional years up to 2021 and sensitivity experiments using high resolution historical climate forcing data to quantify associated improvements of impact simulations (see section 3.1). Besides, the set of historical observation-based*

*direct human forcings has been updated compared to previous ISIMIP simulation rounds (see **Table 1**). For the first time, and closely connected to the evaluation set-up, ISIMIP3a now also includes an 'impact attribution' scenario set-up designed to address the question "To what degree have observed changes in the climate-related systems contributed to observed changes in natural, human or managed systems compared to direct human influences?" Here, changes in climate-related systems mean climate change itself, changes in atmospheric $CO_2$ and $CH_4$ concentration, and sea level changes. The attribution question can both refer to the impacts of individual events (e.g. to what extent has long-term climate change contributed to the observed extent of a specific river flood?) and to long-term changes (e.g. to what extent have long-term climate change and increasing $CO_2$ fertilisation contributed to an observed change in crop yields?)."*

As the papers only provides the cross-sectoral framework of ISIMIP3a we believe that it is enough to mention the different sectors only in the beginning where the texts says:
*"The Inter-Sectoral Impact Model Intercomparison Project ISIMIP (www.isimip.org) provides a common scenario framework for cross-sectorally consistent climate impact simulations currently covering the following sectors: agriculture (global; in cooperation with AgMIP's Global Gridded Crop Model Intercomparison Project (GGCMI)), water (global and regional), lakes (global and regional), biomes (global), forest (regional), fisheries and marine ecosystems (global and regional), terrestrial biodiversity (global), fire (global), permafrost (global), peat (global), coastal systems (global), energy (global), health (temperature-related mortality; water-borne diseases; vector-borne diseases; and food security and nutrition) (global and local), and labour productivity (global and local)."*

- The innovation of ISMIP3a compared to ISMIP2a is to offer a well-thought framework for « climate impact attribution », but this expression is not defined before section 2.2, p22 ; I suggest to bring this part into the Introduction, and to develop a bit this discussion and the references. There is only one, O'Neill et al. (2022), thus omitting the seminal papers of Cramer et al. (2014) and Hansen et al. (2016), and a few newer ones probably.

That is a very valid point! We have moved the explanation from the beginning of section 2.2 to the introduction and have added the references to chapter 18 of the WG2 contribution to the IPCC AR5 (Cramer et al., 2014) and to the Cross-Working-Group Box on Attribution in Chapter 1 of the WG2 contribution to the IPCC AR6:

*"The IPCC AR5 (Cramer et al., 2014) and AR6 (O'Neill et al., 2022 and Hope et al., 2022) have established a framework for impact attribution according to which an 'observed impact of climate change or change in any other climate-related system' is defined as the difference between the observed state of the human, natural or managed system and a counterfactual baseline that characterises the system's behaviour in the absence of changes in the climate-related systems. This counterfactual baseline may be stationary or vary in response to direct human influences such as changes in land use patterns, agricultural or water management or population distribution and economic development affecting exposure and vulnerability to weather-related hazards. While the definition is quite straightforward, the number of studies addressing impact attribution based on this basic definition is still relatively small compared to the number of studies addressing climate attribution, i.e. the question to what degree anthropogenic emissions of climate forcers, in particular*

*greenhouse gases, have induced changes in the climate-related systems. While climate attribution is mainly confronted by the challenge of separating the anthropogenically forced changes from the internal variability of the climate-related systems, the focus of climate impact attribution is on separating the impacts of observed changes in these climate-related systems from the effects of other direct (human) drivers of changes in the considered natural, human or managed systems. 'Observed changes in the climate-related systems' does not necessarily imply 'changes induced by anthropogenic climate forcing', but only means 'any long-term trend' in line with the IPCC definition of climate change (see Glossary of the AR5 and AR6).*

*Impact attribution studies usually face the problem that the counterfactual baseline assuming no long-term changes in the climate-related systems cannot be observed (see Hansen et al. 2016 for examples). However, impact models such as the ones participating in ISIMIP are well suited to simulate this baseline. As the impact models usually account not only for the changes in climate or the climate-related systems but also for direct human forcings such as land use and irrigation changes, changes in water and agricultural management, population distributions etc. (see **Table 1** for a comprehensive list of direct human forcings provided within ISIMIP3a) they are ideal tools to address the attribution question: In line with the IPCC definition it requires the comparison of a factual simulation based on the observed variations in the climate-related and direct human drivers to a counterfactual simulations where only the climate-related forcings are replaced by counterfactual versions where long-term trends have been removed. While the factual simulations correspond to the evaluation runs within ISIMIP3a (see section **2.1**), the protocol now also includes the counterfactual simulations based on the newly generated counterfactual data sets derived from observational data of climate and coastal water levels (see sections **2.2** for the associated concept and scenario design and **Table 3** for a comprehensive list of the counterfactual climate and sea level forcing data that are described in more detail in section **3.1** and **3.3**, respectively). To allow for an attribution of 'observed changes in natural, human, and managed systems' in contrast to an attribution of simulated changes it has to be demonstrated that the processes represented in the impact model can explain the observed changes in the affected system, i.e. it has to be shown that the model forced by observed changes in the climate-related systems and accounting for the historical development of direct (human) forcings is able to reproduce the observed changes in the affected system. In this way the attribution exercise is closely linked to the ISIMIP3a evaluation exercise. Thereby, models can either explicitly represent known changes in non-climate drivers such as known adjustments of fertiliser input or growing seasons (explicit accounting for non-climate drivers) or implicitly account for their potential contributions by e.g., allowing for non-climate related temporal trends in empirical models as often done in empirical approaches (implicit accounting for non-climate drivers)."*

- In doing so, I think the authors could better distinguish two drivers of the changes in human-impacted sectors since 1901: past climate change vs other anthropogenic pressures. If I understood correctly, ISIMIP3a intends to quantify how much of past trends can be attributed to past climate change ; this is the climate impact attribution, using factual and counterfactual climate related forcing (CRF). But ISIMIP3a also offers counterfactual direct human forcing (DHF), which are planned to be used in sensitivity experiments, but could also support attribution : contrasting factual and counterfactual DHFs should allow to quantify how much of past trends can be attributed to past DHF. This strategy has long been explored in the sector of hydrology and water resources at various scales, for various DHFs, with more

or less quantification (e.g. Alkama et al., 2010 ; Sterling et al., 2013 ; Tramblay et al., 2021 ; Teuling et al., 2019 ; Vicente-Serrano et al., 2019) and ISIMIP3a would be a great opportunity to consolidate a comprehensive framework to attribute past trends to CRFs, DHFs, and their interactions. If too late to adapt the simulation set-up, this generalization could at least be discussed in Conclusion as a perspective. In this case, I suggest that the title is changed to « climate impact attribution » instead of « impact attribution ».

Yes, it is absolutely right that part of the sensitivity experiments also allow for an attribution to DHF, while the focus of ISIMIP3a is on the attribution to observed changes in climate-related systems. We now mention this aspect in the introduction. In addition, we make that aspect more explicit in the description of the individual experiments in section 2.1:

Introduction:

[revised manuscript text omitted]

- If space allows, I would also recommend to quickly compare ISIMIP to other important MIPs, especially the ones which address climate change impacts and climate impact attribution (IPCC since Cramer et al., 2014 ; LUMIP, sharing many DHF datasets with ISIMIP ; maybe TRENDY, etc.).

*Thank you very much for the very helpful suggestion. We now explicitly mention TRENDY and compare its attribution approach to the ISIMIP3a set-up (see modified text in response to the previous comment). In addition, we refer to LUMIP and DAMIP that address the attribution of historical changes in climate to observed changes in land use and anthropogenic climate forcing including the emission of GHG gases and aerosols:*

*"[...] However, in this paper the term 'impact attribution' is used as a short form of 'attribution of observed changes in natural, human and managed systems to observed changes in the climate-related systems' which is the focus of the ISIMIP3a experiments. In other cases the driver to which the changes are attributed is explicitly named. In addition to ISIMIP3a, there are other model intercomparison projects that address different kinds of 'climate attribution' questions such as the Land Use Model Intercomparison Project (LUMIP, Lawrence et al., 2016) and Detection and*

*Attribution Model Intercomparison Project (DAMIP, Gillett et al., 2016) embedded into the sixth phase of the Coupled Model Intercomparison Project (CMIP6). While the phase 2 LUMIP experiments include historical climate model simulations to quantify the contribution of historical land use changes to observed climate change, the DAMIP protocol include a counterfactual 'no anthropogenic climate forcing' baseline to attribute observed changes in climate to anthropogenic climate forcings."*

Paper structure and use of tables
- The reading is difficult because of the many long tables. We can distinguish three kinds of tables :
    - Tables 1 and 3 describe the datasets, factual and counterfactual respectively;
    - Tables 2 and 4 describe the experiments: evaluation and sensitivity experiments in Table 2, and climate impact attribution experiments in Table 4;
    - Tables 5 to 18 detail the variables which are included in the datasets.

  The latter ones are usually not cited in the text (true for 6, 7, 9, 11, 12, 13, 14, 15, 18, while Table 17 is only cited in the caption of Figure 10) and could conveniently be placed in annex or supplementary material without making the understanding more difficult, since the most important aspects of these tables are summarized by some text. The related figures, not cited either in the text for most of them, can either be kept in the main text or put in annex or supplementary, but should be cited.

*Thank you very much for the hints! We would like to keep the Tables in the main text as they are closely connected to the text providing some more detail on the included variables and units etc.. We would like to keep the paper self-contained that readers (in particular the modellers thinking about participating in ISIMIP3a) can use it as a kind of catalogue where they get all the information about the components of ISIMIP3a that they need to understand the design and the data that are provided to do the experiments. We fully agree that the paper is very long. However, we expect readers to jump to the parts that are most relevant for them but not to run through the entire manuscript. We now explicitly refer to the Tables and Figures in the main text.*

- The structure of section 2, which describes the experiments, could probably be made more efficient by better articulating the sub-sections, Figure 1, and Tables 1 to 4. Please consider the following suggestions :
    - Put the first part of section 2 under a numbered subsection 2.1 devoted to the ISIMIP3a rationale, based on Figure 1.
    - Split present subsection 2.1 « Model evaluation and sensitivity experiments » in two different subsections : one for the evaluation experiments, linked to Table 1 (datasets for factual experiments, and to the first part of Table 2), and one for the sensitivity experiments, as described in Table 2 starting at CO2 sensitivity under 'histsoc'. This part of Table 2 should refer to the subsections in which the used DHF datasets are described (within section 4).
    - It wouldn't hurt to do the same for the CRF datasets. To this end, I also suggest to split section 3.1 « Observational atmposheric climate forcing data (factual and counterfactual, starting at bottom of p 24) » in three different subsections : one for the factual (default) dataset, one for counterfactual dataset, and one for the high

resolution factual datasets. The paragraphs already exist, so it's easy, and it would be useful to refer to the description of these datasets inside the tables.

- If the authors accept the above suggestions, section 2.2 on the climate impact attribution experiments would become 2.4, linked to Table 3 (counterfactual CRF) and Table 4 (attribution experiments).

- The last paragraph of the Introduction could also be reshaped to introduce the different kinds of experiments, and I wonder if Tables 1 to 4 wouldn't be worth being mentioned here. I'm not sure, however, that the list of all the subsections of section 3 (CRF) and 4 (DHF) is particularly useful here.

As we would like to stress the connection between all experiments described in the original section 2.1 that makes this grouping different from the experiments based on the counterfactual CRF we have introduced a second level of subsections to split up section 2.1. To account for the suggestion related to section 3.1 and at the same time still stress the connection between the three data sets described in the original section 3.1. we have also introduced a second level of subsections to section 3.1.. We have adjusted the last paragraph of the introduction.

Section 2:
Keeping the sensitivity experiments in the same subsection 2.1 as the obsclim-based default experiments highlights their connection in the sense that they are all based on the 'obsclim' climate-related forcings. The structure reflects the file naming convention according to which all simulations described in section 2.1 share the common 'obsclim' specifier in the file name in contrast to the attribution experiments described in section 2.2 that all share the common 'counterclim' specifier for the CRF. We now mention this connection explicitly in the introduction of section 2 and section 2.1 to explicitly explain the structure. To account for subgrouping into the default experiments described in Figure 1 and the sensitivity experiments we have now introduced the subsections 2.1.1 and 2.1.2:

"**2.1 Model evaluation and sensitivity experiments based on observed CRFs ('obsclim')**

*The experiments described in this section are all based on observational (factual) climate data, coastal water levels, and atmospheric $CO_2$ as well as $CH_4$ concentrations including observed trends. The only exception are the sensitivity experiments where $CO_2$ concentrations are fixed at 1901 levels ('1901co2'). However, as these experiments only deviate in this one aspect from the factual CRF they are also described by the 'obsclim' CRF specifier but the '1901co2' sensitivity specifier to indicate the deviation. So all experiments described in this section share the common 'obsclim' CRF specifier in the file names. In contrast, all experiments described in section 2.2 can be identified by the 'counterclim' specifier in the names of the output files containing the impact model simulations.*

*2.1.1 Default evaluation experiments based on observed CRFs ('obsclim')*

*In the first part of section 2.1 we describe the default ISIMIP3a experiments (sensitivity specifier in the file names set to 'default') that are based on the standard observed climate-related forcings ('obsclim', see CRF part of Table 1) in combination with different assumptions regarding direct human forcings ('histsoc', '2015soc', '1901soc', and 'nat') illustrated in Figure 1. [...]*

*2.1.2 Sensitivity experiments based on observed CRFs ('obsclim')*

*This second part of section **2.1** is dedicated to the different sensitivity experiments described as deviations from the default cases described in section **2.1.1**. Instead of the 'default' specifier, all experiments described here are labelled by sensitivity specifiers indicating their deviation from the default cases. The experiments listed here are not explicitly depicted in **Figure 1**. [...]"*

Section 3:
We also decided in favour of splitting section 3.1 into three different subsections but keeping section 3.1 as an overarching structure as we would like to stress the connection between the three atmospheric climate data sets that all include the same variables listed in Table 5 while the other subsections of section 3 all refer to other forcing variables. In addition, the counterfactual and the high resolution data are directly derived from the default observational atmospheric climate data described at the beginning of the section (now subsection 3.1.1). We have also changed the order of the subsections on the high resolution data and the counterfactual data to better reflect the order of the description in section 2.1:

*"**3.1 Observational atmospheric climate forcing data (factual + counterfactual)***

*The data sets described in this sections all contain the variables listed in **Table 5** at the resolution indicated there. While section **3.1.1** described the standard atmospheric climate forcing as one component of the default 'obsclim' CRF used within the evaluation experiments (see section **2.1.1**), section **3.1.2** describes the derivation of the high resolution data used within the 'obsclim'-based sensitivity experiments (see section **2.1.2**), and section **3.1.3** provides a description of the basic approach and the references for the derivation of the counterfactual atmospheric climate forcings used for the counterclim experiments described in section **2.2**. [...]*

**3.1.1 Default factual data**
As one component of the desfault 'obsclim' CRFs, we provide four observational datasets specifically generated for the evaluation experiments of ISIMIP3a: GSWP3-W5E5, 20CRv3-W5E5, 20CRv3-ERA5, and 20CRv3. All four datasets have daily temporal and 0.5° spatial resolution and cover the variables listed in **Table 5**. [...]

**3.1.2 High resolution atmospheric factual data (CHELSA-W5E5)**
This dataset is provided to facilitate the high resolution sensitivity experiment described in section **2.1.2**. It covers the global land area at 30'' (~1 km) horizontal and daily temporal resolution from 1979 to 2016 for the variables precipitation (pr), surface downwelling shortwave radiation (rsds), and daily mean, minimum and maximum near-surface air temperature (tas, tasmin, tasmax).[...]

**3.1.3 Default counterfactual data**
To simulate the baseline 'no climate change' state of a human or natural system that is required for impact attribution, we provide a detrended version of the observational factual forcing data using the ATTRICI approach (ATTRIbuting Climate Impacts, Mengel et al., 2021). [...]"

Final paragraph of the introduction:
We have modified the last section of the introduction to now include the references to the four tables. Having in mind that the paper may be used as a catalogue to get the information about only

individual data sets that are relevant for a specific sector within ISIMIP, we decided to keep the references to all subsections of section 3 so that readers could easily jump there.

*"In the following section **2** of this paper, we provide the comprehensive list of all ISIMIP3a model evaluation and sensitivity experiments (see **Table 2** within section **2.1**) and the counterfactual 'no climate change' experiments (see **Table 4** within section **2.2**) describe the rationale behind the scenario set-ups. Detailed description of the climate-related forcing data sets (see CRF section of **Table 1** in section **2.1** and **Table 3** in section **2.2**) are provided in the third section: atmospheric climate data (see section **3.1**); tropical cyclone data (see section **3.2**); coastal water levels (see section **3.3**), and the ocean data (see section **3.4**). Section **4** presents the ISIMIP3a direct human forcing data sets (see DHF section of **Table 1**), comprising population data (see section **4.1**), gross domestic product (see section **4.2**), land use and irrigation patterns (see section **4.3**), fertiliser inputs (see section **4.4**), land transformations (see section **4.5**), nitrogen deposition (see section **4.6**), crop calendar (see section **4.7**), dams and reservoirs (see section **4.8**), fishing intensities (see section **4.9**), regional forest management (see section **4.10**), and desalination (see section **4.11**)."*

Other comments
- The link to get the ISIMIP3a protocol should be given in the paper

The reference is included in the introduction where we say *"The paper is accompanied by a simulation protocol (ISIMIP3a simulation protocol, 2023) providing all technical details such as file and variable naming conventions and sector-specific lists of output variables to be reported by the participating modelling teams."* The link is then provided in the reference list:

*ISIMIP3a simulation protocol: https://protocol.isimip.org/#ISIMIP3a, last access: 14 January 2023.*

- What is the difference between « default » and « mandatory » and « 1st priority »?

The term 'default' distinguishes between the set of core experiments (sensitivity specifier = 'default' in the file name convention for the impact simulations data submitted toISIMIP3a) and the sensitivity experiments that are described as deviations from the default experiments specified by the CRF and DHF specifier of the default runs and labelled by an associated alternative term replacing the 'default' label. The priority label attached to the different experiments only indicates that assuming limited capacities modellers should start from the 1st priority experiments because they are considered more important across sectors and only submit the 2nd priority one if possible. This is only meant as a hint where in the end it is also possible to only submit 2nd priority experiments to the ISIMIP3a repository.
In contrast the 'mandatory' versus 'optional' label refers to the forcing data sets. We hoped to describe the difference in this paragraph at the beginning of section 2. We have now slightly modified the text (red parts) to more clearly address your points:

*"ISIMIP3a includes a core ('default') set of experiments that are specified by a specific set of underlying climate-related forcings and direct human forcings that have to be indicated in the file*

*names when submitting simulation data to the ISIMIP repository. In the following we first introduce these default experiments by defining the combination of both types of forcing data sets. In the subheadings naming the experiments the associated CRF and DHF specifiers to be used in the file names are indicated in brackets where the third sensitivity specifier set to 'default' (CRF specifier + DHF specifier, default). The different combinations of the default sets of ISIMIP3a CRFs ('obsclim', 'counterclim') and DHFs ('histsoc', '2015soc', '1901soc', '1850soc', 'nat') are sketched in* **Figure 1** *and defined in more detail below (see* **Table 1** *for the default 'obsclim' CRF and the default DHFs and* **Table 3** *for the 'counterclim' CRF). Some of the forcing data sets are mandatory: i.e. if impact models account for the forcing, the specified dataset must be used; if an alternative input data set is used instead, the run cannot be considered an ISIMIP simulation. We also provide 'optional' forcing data that could be used but are not 'mandatory' in the above sense (see second column of* **Table 1** *and* **Table 3**). *In addition, the protocol includes a set of sensitivity experiments that are described as deviations from the default runs and labelled by the baseline CRF and DHF settings and the third specifier then indicating the deviation from this default setting instead of being set to 'default'. The ISIMIP3a sensitivity runs include experiments with high-resolution climate forcing ('30arcsec', '90arcsec', '300arcsec', or '1800arcsec'), fixed levels of atmospheric $CO_2$ concentrations ('1901co2'), a scenario assuming no water management ('nowatermgt'), simulations excluding the occurrence of wildfires ('nofire'), keeping irrigation patterns at 1901 levels ('1901irr'), and assuming fixed 1955 riverine inputs of freshwater and nutrients into the ocean ('1955-riverine-input') (see* **Table 2**). ***Table 2** and* **Table 4** *providing the comprehensive list of all 'obsclim' and 'counterclim'-based experiments, respectively, also indicate the priority of the experiments where '1st priority' means that modellers should focus on this set of experiments if their capacities were limited and they wanted to limit the set of experiments. However, this is just an indication trying to ensure the generation of a small set of experiments that is covered by as many impact models as possible. If an impact modeller can only do part of the first priority set-up or has to start from second priority simulations these fragmented data sets can also be submitted to the ISIMIP3a repository.* "

- Another term in the tables is « optional » : does it mean it is not mandatory, is it « 2nd priority », or something else ?

see explanation above

- Many Tables and Figures are not referred in the text (Figures; Tables)

We hope that all Figures and Tables are now referred to in the text!

- Many datasets are detailed in submitted papers, which may end up being a problem if these papers are not accepted.

The problem affects the following references:
- Treu et al., submitted 2023
- Frieler et al., submitted 2023, and
- Rousseau et al., submitted 2023.

As the Frieler et al. manuscript will not be accepted in time and is not critical for the ISIMIP3a protocol, we have deleted the reference. The Treu et al. paper has been re-submitted after minor revisions, so we hope that it will be accepted soon. Besides, the preprint is available online (https://doi.org/10.5194/essd-2023-112). The manuscript by Rousseau et al. is currently under revision. It will be resubmitted soon and the revised version will become available online as a preprint within the coming three weeks (including a doi). So we will be able to refer to the preprint in the very final version of the paper.

- I wonder about the use of atmospheric composition, since there is no dataset about aerosols, ozone, or N2O. I guess that vegetation models use CO2, but some may use ozone as well. And which models need CH4 as input? A few words about the models' need by sector could help here.

The provision of the input data within ISIMIP is very demand driven. So we provide the CH4 data as it is needed by some biomes/permafrost/crop models. So far none of the models needs e.g. ozone or other information about atmospheric composition. If we were approached by a (new) modelling team that needs the data we would try to find ways to provide this information and add it to the ISIMIP3a/b forcing data. In this sense the paper is a snapshot of the current set of input data that may be extended in the future. We provide a short documentation of the participating models and the set-up they have used within the different modelling rounds on our website (https://www.isimip.org/outputdata/). The list of output data can e.g. be filtered by modelling round and sector to find the available outputs. The individual documentations you find, when clicking on the associated model name e.g. contains the information about the input data that have been used (e.g. whether the model used CH4 or not). We have included this information in the paper now:

*"The provided forcing data sets (e.g. the climate variables or components of atmospheric composition or types of land use) is very much demand driven. The data we describe here represent a core set that is sufficient for the range of models participating so far (see ISIMIP output data table that also provides information about the input data used by the individual models) but may be extended if there were further demands."*

- Riverine inputs : section 3.4 explains that the time-varying riverine input dataset was produced by the LM3-TAN land model, and I guess they depend on fertilizers. In such a case, they result from both CRF and DHF. This might be acknowldeged. At L844, we read that no counterfactual riverine input dataset is provided : but isn't the 1955 dataset counterfactual ?

Yes, the inputs depend on climate and land use changes including changes in fertilizer inputs, i.e. they result from both CRF and DHF. The text includes a full description of the scenario that was missing before. It particularly highlights this aspect:

*"Fixed 1955 riverine input into the ocean sensitivity experiment (obsclim + histsoc; obsclim + nat; 1955-riverine-input).* In this '1955-riverine-input' experiment, riverine input into the ocean (amount of freshwater and nutrients) is held constant at 1955 levels. In comparison to the default 'obsclim + histsoc' simulation, the experiment allows for the quantification of the impacts of historical climate-induced variations in freshwater influx in combination with the climate and directly human*

*induced changes in nutrient inputs (attribution of observed changes in marine ecosystems and fisheries to long term changes in riverine freshwater and nutrient inputs). The riverine inputs in the 'obsclim + nat; 1955-riverine-input' experiment are identical to the ones in the 'obsclim + histsoc; 1955-riverine-input', i.e. the riverine inputs also account for the human contribution to the nutrient influx due to land use changes and fertiliser inputs and are not 'naturalised'. Instead the 'nat' specifier in the marine ecosystems and fisheries sector only means 'no fishing efforts'. Thus, the comparison to the naturalised default experiment (obsclim + nat; default) not accounting for any fishing efforts to the 'obsclim + nat; 1955-riverine-input' experiment allows for a quantification of the contribution of climate-induced changes in freshwater-influx to the overall impacts of climate change in combination with the contribution of the effect of the human contribution to nutrient inputs at 1955 levels. The sensitivity experiment has been introduced into the marine ecosystems and fisheries protocol. A potentially required spin-up should be done similar to the spin-up for the associated default experiments but assuming riverine inputs fixed at 1955 levels."*

Similar to the 'obsclim + histoc; 1901co2' experiment, the 'obsclim + histsoc, 1955-riverine-input' experiment can be considered a counterfactual as now described in the text. However, it is not a 'full' counterfactual as the associated simulations of changes in the marine ecosystems and fisheries will account for changes in ocean temperature etc.. The counterfactual only refers to the riverine inputs and is therefore rather considered a sensitivity experiment to the obsclim-based simulation than part of the 'counterclim' attribution experiments.

- Section 4 mentions group I, II and III simulations which are never defined (L939, L1051)

Thank you so much for the hint! We have deleted the reference to the different groups of experiments within ISIMIP3b and now only mention in the introduction that the ISIMIP3a DHF are also used for the historical simulations within ISIMIP3b:

*"This paper describes the ISIMIP3a simulation framework only where the DHF described here are also used for the historical simulations within ISIMIP3b."*

- In section 4.3, land use is limited to agriculture : why not account for urban areas, especially given that Table 13, p160, includes conversion to urban ? what about planted trees?

Yes, urban areas are explicitly included in the LUH2 data we provide. This is now explicitly mentioned in the main text and the Table. We do not specify other areas as the listed ones as that would constrain the dynamical vegetation models. We now explicitly mention this at the ends of the section:

*"The areas outside of the specified agricultural and urban land is considered 'natural vegetation' and not prescribed further to not constrain the dynamical vegetation models."*

- Harmonisation would be welcome for LUH2 vs LUH v2.

We have harmonised the label of this dataset throughout the document. It now says "LUH v2" almost everywhere. The only exception is the title of a cited manuscript

D. P., and Zhang, X.: Harmonization of global land use change and management for the period 850–2100 (LUH2) for CMIP6, Geosci. Model Dev., 13, 5425–5464, https://doi.org/10.5194/gmd-13-5425-2020, 2020)

which we cannot change.

- The first sentence of subsection 4.9 (L1095), which relates the dataset to a sector of ISIMP, is extremely informative, and could serve as an example for all subsections.

The fishing efforts data is one of the rather rare data sets that is only used within one sector. Most of the other data sets are used across multiple sectors and we do not want to name sectors explicitly to not limit the application. In contrast a data set that is considered mandatory has been used across all sectors (including potential new ones) that need this type of information as input. All the data we provide within ISIMIP3a are some kind of 'historical reconstructions' as the simulations are observation driven trying to represent the 'real' historical situation as closely as possible and the information whether the data set belongs to the CRF or the DHF can be found in Table 1 and is also given by the section number (section 3 on the CRF and section 4 on the DHF).

- The last section should be a Conclusion and not a Discussion

We have renamed the last section.

**New references**

- Alkama, R., Kageyama, M., and Ramstein, G. (2010), Relative contributions of climate change, stomatal closure, and leaf area index changes to 20th and 21st century runoff change: A modelling approach using the Organizing Carbon and Hydrology in Dynamic Ecosystems (ORCHIDEE) land surface model, J. Geophys. Res., 115, D17112, doi:10.1029/2009JD013408
- CRAMER, Wolfgang, YOHE, G., HOUSE, Joanna Isobel, et al. Detection and attribution of observed impacts: Climate Change 2014: impacts adaptation and vulnerability, contribution of working group ii to the fifth assessment report of the intergovernmental panel on climate change. In : Climate Change
- 2014: Impacts Adaptation and Vulnerability, Contribution of Working Group II to the Fifth Assessment Report of the Intergovernmental Panel on Climate Change. Cambridge University Press, 2014.
- Gerrit Hansen, Daithi Stone, Maximilian Auffhammer, Christian Huggel, Wolfgang Cramer (2016). Linking local impacts to changes in climate: a guide to attribution. Regional Environmental Change, 16, 527–541. https://doi.org/10.1007/s10113-015-0760-y
- Sterling, S., Ducharne, A. & Polcher, J. The impact of global land-cover change on the terrestrial water cycle. Nature Clim Change 3, 385–390 (2013). https://doi.org/10.1038/nclimate1690
- Tramblay, Y., Villarini, G., El Khalki, E. M., Gründemann, G., & Hughes, D. (2021). Evaluation of the drivers responsible for flooding in Africa. WaterResources Research, 57, e2021WR029595. https://doi.org/10.1029/2021WR029595

- Teuling, A. J., de Badts, E. A. G., Jansen, F. A., Fuchs, R., Buitink, J., Hoek van Dijke, A. J., and Sterling, S. M.: Climate change, reforestation/afforestation, and urbanization impacts on evapotranspiration and streamflow in Europe, Hydrol. Earth Syst. Sci., 23, 3631–3652, https://doi.org/10.5194/hess-23-3631-2019, 2019.
- Vicente-Serrano, S. M., Peña-Gallardo, M., Hannaford, J., Murphy, C., Lorenzo-Lacruz, J., Dominguez-Castro, F., et al. (2019). Climate, irrigation, and land cover change explain streamflow trends in countries bordering the Northeast Atlantic. Geophysical Research Letters, 46, 10821– 10833. https://doi.org/10.1029/2019GL084084

**Annotations to the manuscript:**

- Line 93 ff:

"The consistent design of the simulations does allow for the comparison of climate impact simulations within each sector. However, it also enables the bottom-up integration of impacts across sectors. Thus, it provides a unique basis for the estimation of the effects of climate change on, e.g., the economy, …"

*comment:* remove the sentence "However, …"

We would like to keep the sentence as this is a central component of ISIMIP that makes it different from other intercomparison projects. For individual sectors we would even include simulations from only one impact model to generate a more comprehensive picture of the impacts of climate change going beyond the classical idea of comparing model simulations.

- Line 126 f:

"The paper is accompanied by a simulation protocol (ISIMIP3a simulation protocol, 2023) providing all technical details"

*comment:* Not available on GMD page; why not give the "official" link? https://protocol.isimip.org/#12-simulation-round

We refer to "ISIMIP3a simulation protocol: https://protocol.isimip.org/#ISIMIP3a, last access: 14 January 2023." this leads to the exact same page you suggest to refer to. By including the simulation round [ISIMIP3a] in the link we hope to avoid potential confusion, since otherwise one could easily switch to the 3b simulation round by accident.

- Line 129:

"The ISIMIP3 simulation round was officially started…"

*comment:* ISIMIP3 or ISIMIP3a?

Both rounds have been started together. This is explained in the following paragraph of the main text:

*"In the second round of ISIMIP the observation-based model evaluation part (ISIMIP2a) was temporally separate from the climate model-based second part (ISIMIP2b, Frieler et al., 2017). This has led to inconsistencies in the models and model versions contributing to ISIMIP2a and ISIMIP2b. Also, not all models providing future projections within ISIMIP2b also provided model evaluation runs for ISIMIP2a. To avoid this problem and ensure that each model's set of future projections is accompanied by associated historical simulations allowing for model evaluation, in the third simulation round (ISIMIP3), the ISMIP3a and ISIMIP3b protocols were released together and participating in ISIMIP3 means contributing to ISIMIP3a and ISIMIP3b using the same impact model versions."*

- Line 133 ff:

"Impact modellers interested in contributing to ISIMIP should therefore refer to ISIMIP3a simulation protocol, 2023 for the most up to date reference for planned impact model simulations. It includes a unique version identifier on its front page for traceability."

*comment: where defined?*
We provide the commit hash at protocol.isimip.org that leads to the underlying code version on github where the protocol is being developed. To make this more clear we adjusted the sentence as follows:

*"Impact modellers interested in contributing to ISIMIP should therefore refer to ISIMIP3a simulation protocol, 2023 for the most up to date version for planned impact model simulations. The protocol landing page (protocol.isimip.org) includes a unique version identifier (the commit hash) that links to the latest protocol version on github for traceability."*

- Line 142 ff:

"To avoid this problem and ensure that each model's set of future projections is accompanied by associated historical simulations allowing for model evaluation, in the third simulation round, the…"

*comment:* make it explicit that third round is ISIMIP3
Done! We have adjusted the sentence as follows:
"To avoid this problem and ensure that each model's set of future projections is accompanied by associated historical simulations allowing for model evaluation, in the third simulation round *(ISIMIP3)*, [...]"

- Line 146:

"In the following, we describe the rationale behind the individual scenario set-ups (section 2)."

*comment:* remove "individual"
Done!

- Line 198 f:

"Standard evaluation experiment using observed variations of direct human forcings (obsclim + histsoc; default)."

*comment:* remove "using observed variations…"? As it is told in the following explanation.
Done!

- Line 203:

"To this end, we provide the climate-related ('obsclim'), direct human ('histsoc'), and static geographical forcings listed in Table 1 and described in more detail in sections 3 and 4."

*comment:* suggestion: cut sentence after *Table 1.* They are described…
Done!

- Line 225 (Table 1 header):

"...| Status | …"

*comment:* I don't understand what explains if a forcing is mandatory or optional: could you explain it in the text?

See related comment above. The text has been adjusted accordingly.

- Line 225 (Table 1: Local atmospheric climate forcing for lake locations):

*comment:* Section?

There is no dedicated section for this data set as it is simply derived from the standard atmospheric forcing data mentioned in the row before. The important information about the locations is included in the cited paper from Golub et al., 2022.

- Line 225 (Table 1: Lightning)

*comment:* Section?

The reason above applies here as well: There is no dedicated section for this data set as we provide the original data described in the given reference.

- Line 225 (Table 1:standard observation-based oceanic forcing):

*comment:* which resolution?

The standard resolution of the oceanic forcings is 0.25° (see section **3.4**). There the text reads:

*"To create the default 'obsclim' climate-related forcings for the fisheries and marine ecosystem models these ocean model simulation data have been interpolated to a regular 0.25° grid while vertical resolution is preserved. In contrast to the atmospheric data, oceanic CRF are provided at monthly temporal resolution."*

We have also adjusted the caption of Table 1 that now reads:

***"Table 1: Climate-related, direct human, and static geographic forcing data provided for the model evaluation and sensitivity experiments within ISIMIP3a.*** *The CRFs are grouped according to the definition of the default 'obsclim' CRF (30 arcmin for the atmospheric data and 15 arcmin for the oceanic data), the higher resolution '30arcsec', '90arcsec', '300arcsec', '1800arcsec' atmospheric CRF, the lower resolution '60arcmin' oceanic CRF, and the '1955-riverine-input' oceanic CRF for the sensitivity experiments. The listed set of DHFs defines the 'histsoc' set-up."*

- Line 225 (Table 1: regional oceanic climate forcing for regional marine ecosystems and fisheries sector):

"Extraction from data set above for 21 regional marine ecosystems."

*comment:* are they given somewhere? it would help to give here the related subsection number.

There is no dedicated subsection for this data set as it is simply derived from the original data described above.

- Line 225 (Table 1: atmospheric CH4 concentration):

*comment:* in which section?

There is no dedicated section for this data as it is fully described in the cited papers.

- Line 225 (Table 1: low resolution observation-based oceanic forcing)

*comment 1:* why insist on the low resolution here and not in the following table entry?

We wanted to stress the deviation from the default oceanic forcing in the same way as we do it for the atmospheric forcing in the row above. In contrast to the atmospheric data, the standard oceanic forcing is at higher resolution than the forcing provided for the sensitivity experiment.

*comment 2:* It would help to mention here that this dataset has time-varying riverine input (for comparison with the following table entry). An overview of the kind of variables comprising this oceanic forcing data would be useful at this stage.

We have added an explanation to the description of the standard oceanic forcing:

*"GFDL MOM6/COBALTv2 simulations driven by reanalysis-based atmospheric forcing (Liu et al., 2021). In contrast to the '1955-riverine-input' sensitivity experiment described below, the standard oceanic forcing accounts for climate induced changes in riverine freshwater input and climate and directly human induced changes in riverine nutrient input, see section **3.4**"*

- Line 225 (Table 1: water abstraction)

*comment:* It would be more consistent to give the description of this dataset in a subsection of Section 4. Why is RCP6.0 selected for the future?

There is no dedicated section for this data as the table entry and the reference are sufficient to fully describe the data. The extension of the underlying climate data by RCP6 based simulations instead of using observational climate data has been done for technical reasons. We simply do not have a similar data set that provides the water abstraction assuming observational climate forcing. The choice RCP6.0 compared to other RCPs is not critical as we do not expect a significant difference between the associated climate simulations for the period from 2006 to 2021. Overall, climate variability is only a minor driver of water abstraction which is dominated by long-term changes in socio-economic development. So it is actually more important that the expansion of the data after 2005 is based on SSP2. This has been modified in the Table.

"For modelling groups that do not have their own representation, we provide files containing the multi-model mean of domestic and industrial water withdrawal and consumption generated by the WaterGAP, PCR-GLOBWB, and H08 models (1850-2021). This data is based on ISIMIP2a 'varsoc' simulations for 1901-2005 and extended by SSP2-based simulations from the Water Futures and Solutions project up to 2021 (Wada et al., 2016b) <https://paperpile.com/c/dymCb2/KeFR2>. Years before 1901 have been filled with the value for year 1901."/

- Line 225 (Table 1: Lake and reservoir surface data):

*comment:* Is it detailed somewhere in section 4?

As we consider the short information in the Table and the references sufficient to describe the data that has not been modified by us, we have not included a dedicated subsection in section 4.

- Line 225 (Table 1: GDP data):

*comment:* why not list the datasets in this Table in the same order as in section4?

This is a very good point. We have adjusted the Table accordingly!

- Line 225 (Table 1: Lake volume at different depths):

*comment:* The first 8 lines of this part of the table are about lakes: why not lump them into one entry, with a dedicated subsection in section4 ?

We would like to keep all data in one Table as it then provides a full overview of the data available for the standard evaluation experiments. The lake data belong to the 'static geographic' data that do not have a dedicated section at all. Section 3 provides more information  about the 'Climate-related forcings' and section 4 is on the 'direct human influences', but the lake data do not belong to the latter category. That is why they are not included in section 4.

- Line 225 (Table 1: Big lakes mask):

*comment:* Why not lump the two entries about big lakes?

We intended to list all available data sets in Table 1. As there is no dedicated section on the 'static geographic data' we have been a little bit more explicit on these data in the table. It would have been an option to include a fifth section on the 'static geographic data' in the paper, but we decided against it, because most of the data is from other sources and can be described briefly by the references and this third category of data is less central for the scenario design of ISIMIP3a that is defined by the CRF and the DHF.

- Line 225 (Table 1: Soil data)

*comment:* Does this dataset only describe mineral soil soils, or does it include decsriptors of the organic fraction?

Yes, the data set only describes mineral soils and includes all variables covered by HWSD, including organic soil carbon. Organic soils (e.g. peatland) are not covered at all.

- Line 225 (Table 1: Fractional country mask)

*comment:* why not lump?

As there is no dedicated section on the 'static geographic data' we have been a little bit more explicit on these data in the table and would like to make them all visible as separate data sets.

Line 225 (Table 1: Regional Marine Ecosystem masks)

*comment:* details should be given in a subsection

The category 'static geographic data' does not have a separate section in the paper (see response to comment above).

Line 242 f:

"Impact of historical changes in direct human forcings - Fixed 1901 direct human forcing baseline (obsclim + 1901soc; default)"

*comment:* the highlighted part could be removed for consistency with the other experiment titles; the rationale can be explained in the related paragraph
Done!

- Line 256:

"Impact of direct human forcings - No direct human forcing baseline (obsclim + nat; default)."

*comment:* the highlighted part could be removed for consistency with the other experiment titles; the rationale can be explained in the related paragraph
Done!

- Line 264 ff:

"The 'nat' experiment can also be used to quantify the natural carbon sequestration potential of natural vegetation without any management or land-use as an important counterfactual to assess the additionality of carbon sequestration measures."

*comment:* missing noun
Thanks, we complemented the sentence as follows:
*"The 'nat' experiment can also be used to quantify the natural carbon sequestration potential of natural vegetation without any management or land-use as an important counterfactual baseline to assess the additionality of carbon sequestration measures."*

- Line 208 f:

"The 30" atmospheric data (1979-2016) is derived from by a topographic downscaling of the observational W5E5 data …"

*comment:* either from or by
Thanks for pointing to this, the corrected version now reads:
*"The 30" atmospheric data (1979-2016) is derived from a topographic downscaling of the observational W5E5 data …"*

- Line 311:

"Low resolution sensitivity experiment (obsclim + nat; 60arcmin}."

*comment:* This paragraph should be merged with the above one.

We decided to keep this paragraph separate as it only refers to the marine ecosystems and fisheries sector and does exist for the other resolutions (30arcsec, 90arcsec, 300arcsec, 1800arcsec). As such a merged version would not fit into the system of the naming of the experiments in the brackets used for the other paragraphs.

- Line 398 ff:

"While the definition is quite straightforward, the number of studies addressing impact attribution based on this basic definition is still relatively small compared to the number of studies addressing climate attribution, i.e. the question Io what degree anthropogenic emissions of climate forcers, in particular greenhouse gases, have induced changes in the climate-related systems."

*comment:* not really

We have have modified the text that now says:
*"While the definition is established for about a decade at least, the number of studies addressing impact attribution based on this basic definition is still relatively small compared to the number of studies addressing climate attribution, i.e. the question Io what degree anthropogenic emissions of climate forcers, in particular greenhouse gases, have induced changes in the climate-related systems."*

- Lines 408 ff:
"In the case of impact attribution, that means simulations of the considered natural, human or managed system in the absence of climate change, sea level rise, and changes in C02 concentrations."

*comment:* rather "climate impact attribution"

The problem is that 'climate impact attribution' may be too narrow as it does not include all changes in the 'climate-related systems' including sea level rise or changes in atmospheric CO2 and CH4 concentrations. When adjusting the introduction in response to the comment on introducing the definition in the beginning we have added the following sentence to be as explicit as possible:

*"However, in this paper the term 'impact attribution' is used as a short form of 'attribution of observed changes in natural, human and managed systems to observed changes in the climate-related systems' which is the focus of the ISIMIP3a experiments."*

- Lines 417 ff:
"Thereby, models can either explicitly represent known changes in non-climate drivers such as known adjustments of fertiliser input or growing seasons (explicit accounting for non-climate drivers) or implicitly account for their potential contributions by e.g., allowing for non-climate related temporal trends in empirical models as often done in empirical approaches (implicit accounting for non-climate drivers).

*comment:* I don't get it
In empirical models it is quite typical to allow for fixed effects such as a common temporal trends in GDP when building a model to quantify the effect of 'number of people affected by flooding' on national economic development or a model explaining the effects of low soil moisture on observed crop yields. These temporal trends then represent general socio-economic development or technological progress in agriculture without explaining it in terms of the DHF we provide (e.g.

fertiliser inputs). This is what is meant by the sentence that is used in a similar way in chapter 16 of the IPCC AR6.

- Line 474:

"... we focussed on including …"

*comment:* single "s"
Corrected accordingly!

- Line 482 ff:

"W5E5 v2.0 combines WFDE5 v2.0 (WATCH Forcing Data methodology applied to ERA5 reanalysis data over land; Cucchi et al., 2020) with data from the latest version of the European Reanalysis (ERA5; Hersbach et al., 2020) over the ocean.

*comment:* "can you remind if WFDE5 includes bias correction? on which variables?"

Thanks, yes it includes bias correction. We added the following sentence to the manuscript to explain this: *"WFDE5 v2.0 is generated with the WATCH Forcing Data methodology that includes bias adjustment of all variables (Cucchi et al., 2020)."*

- Line 519:

"We then remove these observed trends since 1901 from the observational data …"

*comment:* prefer passive form

We adjusted the sentence as follows: "These observed trends since 1901 are then removed from the observational data …"

- Line 645 (Table 6: "1-minute sustained wind speed", "Maximum 1-minute sustained wind speed during the whole storm duration")

*comment:* the two lines may be lumped in a unique entry (gridded maps of max wind speed?)

We would like to keep variables in separate rows if they have different dimensions. The first variable has an explicit time dimension while the second has not. We revised the variable names and descriptions to make this more explicit.

- Line 645 (Table 6: "National territory exposed to wind speeds of at least 34, 48, 64, 96 knots")

*comment:* This entry could be lumped with the following two, under "wind exposure' for instance

All the Tables in section 3 and 4 should be structured in a 'one variable per row' manner. We would like to keep this structure to mimic the input data tables provided in the simulation protocol. We only lump variables that basically have the same names only distinguished by e.g. the different wind speeds in Table 6 or the '_rainfed' or '_irrigated' specifier in Table 11. We have adjusted the structure of Table 11, 12, and 13 accordingly.

- Line 645 (Table 6: "rainfall - mm")

*comment:* it's not a rainfall rate, the time step is missing

We provide the rainfall totals. We improved the wording in the text and the naming of the variable to make this clear.

- Line 645 (Table 6: "Maximum 24-hourly rainfall during the whole storm duration - mm"

*comment:* These two entries could be merged

See comment above

*comment 2:* mm/d?

We provide rainfall totals (in this case, maximum of daily totals). We improved the wording in the text and the naming of the variable to make this clear.

- Lines 648 f:

"As additional CRF, we provide historical TC tracks (information about the observed location of minimal pressure), associated gridded wind and rain fields.

*comment:* with associated

Indeed, thanks for correcting! The text has been adjusted accordingly.

- Line 671:

"The data set we provide uses best track data from 1950 to 2021"

*comment:* best according to who?

As noted in the paragraph above, the term "best track" is commonly used by meteorological agencies to differentiate between different stages of the meteorological data describing a TC: *The data is marked as provisional before it is replaced by so-called best track data one up to two years after the events.* The "best track" data is the final stage as opposed to forecast and real time (provisional) data. In the sentence you marked, we use the term to emphasise that we do not provide any provisional data in ISIMIP3a, but only best track data.

- Lines 698 ff:

"We derive two different gridded wind field products from an extrapolation of the observed TC track information to gridded estimates of surface wind speeds (1 -minute sustained winds at 10 metres above ground), at a spatial resolution of 300 arc-seconds (approximately 10 km).

*comment:* why two products? which one should be preferred?

We give more details on this in the same and the following paragraphs and hope this is sufficient information for an appropriate use of the data sets. We do not recommend using one or the other. Ideally both versions could be used to test the sensitivity of the impact simulations in terms of the applied wind field model. The approach is similar to providing four different factual and counterfactual atmospheric forcing data sets. We now refer to it in the conclusion (see comment below).

- **Line 745: Figure 3**

*comment:* cited p6 l220 but not in this section

Indeed, thanks! The Figure is now also cited in this section.

- **Line 750: Table 7**

*comment:* Do we need a table for one variable?

The subsections in section 3 and 4 are intended to all have the same structure starting with a Table with the variable names and units.

- **Lines 762 f:**

"The counterfactual is derived from the factual dataset by removing the trend in relative sea level since 1900."

*comment:* missing noun

Thanks for correcting! It now reads "The counterfactual coastal water levels are derived from…"

- **Lines 765 ff:**

"Hence, the data can be used for an event-based attribution of, e.g., observed flooding to observed relative sea-level rise with tuples of impact simulations driven with the factual and counterfactual dataset.

*comment 1:* tuple is not a very common term: prefer pair or ensemble maybe?

We now use 'pair'. Thanks for the hint.

*comment 2:* datasets

Indeed, thanks! We have added the 's'.

- **Lines 811 ff:**

"For the fisheries and marine ecosystem models, we provide a number of physical and biogeochemical variables for the period 1961 to 2010 at different depth levels in the ocean (see Table 10).

*comment:* Table 8?

Indeed, thanks for spotting! We have adjusted the number!

- **Lines 841 f:**

"The data is interpolated to a regular 0.25 degree grid in the same way as the default 'obsclim' CRFs."

*comment:* send back to Table2?

Thanks. We would like to avoid overloading the table and would therefore keep it as is. Readers can easily jump to the more detailed description of the experiment through the '1955-riverine-input' label.

- **Line 855: Figure 4**

*comment:* not cited

Indeed, thanks. We now cite the figure in this section

- Lines 863 f:

"For the year 2021, we use the "medium-variant" of the probabilistic forecast also provided by the WPP.

*comment:* couldn't we have real values of 2021 by now?

Indeed, the 2022 WPP report came out recently. However, we have already started to provide the GDP data to the modelling teams. Therefore, we decided to keep the current version of the dataset to assure consistency across all model runs using this data.

- Line 889: Figure 5

*comment:* not cited

Indeed, thanks. We now cite the figure in this section

- Line 892 f:

"Time series of per-capita GDP for the time period 1960- 2021 are taken from the World Bank's World Development Indicator database (Anon, 2008) and converted"

*comment:* add (WDI) at then end

Ok, thanks! Done!

- Line 926: Figure 6

*comment:* not cited

Indeed, thanks. We now cite the figure in this section

- Line 931: Figure 7

*comment:* Not cited, and panels are too small [solved]

We now cite the figure in this section and show panel A above panel B to increase the size.

- Lines 939 f:

"Historical land use and irrigation patterns for ISIMIP3a and ISIMIP3b, group I and group II simulations are taken from LUH2 (Hurtt et al., 2020)."

*comment:* what is it?

This is indeed not explained. Since this manuscript is only about ISIMIP3a we simply deleted the "... ISIMIP3b, group I and group II …" part of the sentence. See related comment earlier.

- Line 969: Figure 8

*comment:* not cited

Indeed, thanks. We now cite the figure in this section

- Lines 978 ff:

"For the pure crop runs within ISIMIP, where the considered crops are assumed to be grown everywhere without a land use specification, the LUH2 national fertiliser inputs are assumed to be applied everywhere within the country.

*comment:* I don't understad what a pure crop run is, thus when LUH2 is used or not…

The LUH2 land use and irrigation patterns are only applied in post-processing, i.e. the yields from the pure crop runs are multiplied with the land use patterns to calculate the production. This is now explained in the text.

*"For the pure crop runs within ISIMIP, where the considered crops are assumed to to be grown everywhere without a land use specification, the LUH v2 national fertiliser inputs are assumed to be applied everywhere within the country. To calculate crop production, the LUH2 v2 land use patterns are applied in post-processing, i.e. by multiplying the crop yields from the pure crop run with the land use patterns (fraction of the grid cell where the crop has been grown)."*

- Lines 987 ff:

"These datasets are based on the LUH v2h Harmonization Data Set covering 850 to 2015 (Hurtt et al., 2020).

*comment:* LUH v2h or v2 (or LUH2?)

We have unified this to "LUH v2" throughout the document.

- Line 1001: Table 14
*comment:* why not combined with fertilizer input?

Yes, that would have been an option. We decided against combining both tables as the data sources are very different.

- Line 1009: Section 4.7 Crop calendar
*comment:* why not place this section just after land use and irrigation?

There also is a strong connection between the information on land use + irrigation and the fertiliser inputs and land transformation. They can all be considered part of the LUH2-based "land use package". The growing season information stems from another source and is therefore documented afterwards.

- Lines 1011: Table 15:
*comment:* mention the two variables are given for 18 crop types

We have added the information to the caption that now reads:
*"Crop calendar provided as optional representation of agricultural management. The information is given for 18 crop types."*

- Lines 1021 f:

"Within the crop models different crop varieties are represented by the heat units required to reach physiological maturity."

*comment:* replace by "different"

Done!

- Lines 1040 ff:

"In order to offer a consistent and common source of information about reservoirs and associated dams for climate impact modellers {see Table 16), we joined the Global Reservoir and Dam Database of the Global Water System Project {GRanD v1.3; Lehner et al., 2011a, b) with a subset of the Georeferenced global Dams And Reservoirs {GeoDAR v1 .2) database {Wang et al., 2022), developed at Kansas State University {KSU), and provided by Jida Wang ahead of publication."

_comment:_ this seems contradictory with the citation to Wang et al., 2022 at previous line

In 2023 this reads a bit irritating indeed. However, we launched the ISIMIP3 simulation round including the dam data set referred to here, in 2020 already, and at that time, the GeoDAR database (Wang et al., 2022) was not yet published. To clarify this we adjusted the sentence as follows: "… _developed at Kansas State University (KSU), and provided by Jida Wang ahead of publication, so that it could be included in launching ISIMIP3 in 2020_ ."

- Lines 1046 f:

"In total, the combined database now includes 7331 dams whose construction was finished by 2025."

_comment:_ will be

Indeed, thanks for correcting! The text has been adjusted!

- Lines 1047 f:

"It includes dams that were constructed before, but still in existence during, the simulation period …"

_comment:_ before when?

Thanks for pointing to this, it is indeed not fully clear. We refer to the simulation period, i.e. dams that have been built before the simulation period, but are still operational during the simulation period. We make this now more clear by rephrasing the sentence as follows: "_It includes dams that were constructed before the simulation period, but still exist._"

- Lines 1048:

"In total the reported dams have a global cumulative storage capacity of approximately 6932 km3 (Figure 2)."

_comment:_ This sentence would be better placed at the end of the paragraph. And Figure is 9, not 2.

We have corrected the Figure number and moved the sentence to the end of the paragraph.

- Lines 1072 ff:

"Due to the grid cell resolution, reservoirs might get assigned to a grid cell that include the main stream or already a confluence of one or more tributaries even though the dam is located in a particular tributary according to the database."

*comment:* this part is unclear

The coarse resolution can lead at times to artefacts. An example is a dam that is located at a tributary, but is misinterpreted to be located at the main stream, either before or after the confluence of both rivers. We have adjusted the sentence that now reads:

*"Due to the low resolution of the model grid, reservoirs might get wrongly assigned to e.g. the main stream (either before or after the confluence of two rivers), even though the dam is located in a particular tributary according to the database."*

- Lines 1095 f:

"The data set of reconstructed historical fishing efforts (Rousseau et al., 2022) serves as the DHF for the marine ecosystems and fisheries sector."

*comment:* This introductive sentence is extremely informative. I wish it existed for all subsections.

See response to earlier comment.

- Line 1115: 4.10 Forest management for regional forest sector

*comment:* what link with the global wood harvest data?

Thanks for checking! The regional forest data is site-specific and therefore not harmonised to the global gridded wood harvest data that does not resolve this detail. We now clarify that in the manuscript:

*"The regional forest management data has not been harmonised to the global gridded wood harvest data provided for the biomes sector, because the data is very site-specific and the variation not resolved in the global data set."*

- Lines 1122 ff:

"For the historical period, observed stem numbers and forest thinning types are provided in the same ways as in ISIMIP2b from the PROFOUND Database (Reyer et al., 2020b) so that modellers can mimic the exact management that has happened at the site and perform the histsoc runs as close to reality as possible."

*comment:* clarify how ISMIP2b compares or not to ISIMIP3a
For ISIMIP2b the management assumptions also refer to the future period, while ISIMIP3a only refers to the historical period. As this may be misleading here, we simply deleted the sentence.

- Lines 1133 ff:

"This paper aims to give an overview over the ISIMIP3a experiments and the provided climate-related and direct human forcing data sets. It is intended to work as a catalogue where modellers can find all relevant information about the data sets they need as reference for the impact model simulations within ISIMIP3a."

*comment 1:* of

Indeed, thanks! The text has been adjusted accordingly.

*comment 2:* remove

Done!

- Lines 1138 ff:

"So this paper can also be read as a call for contributing additional data that could i) either be provided optionally within the current round as the optional data sets do not have to be harmonised across all model simulations or ii) as mandatory forcing for an upcoming simulation round."

*comment:* have not been

The optional data (see explanation in response to an earlier comment) are not harmonised within or across sectors. So a newly available data set could be used by modelling teams joining after the release without becoming inconsistent with the protocol. To make this clearer the sentence now reads: *"So this paper can also be read as a call for contributing additional data that could i) be provided within the current round (ISIMIP3) as optional data set that is not harmonised within or across sectors (see explanation in the introduction) …"*

- Lines 1150 ff:

"As impact models encode our process knowledge on how several drivers (climate-related ones as well as direct human influences) come together to generate observed changes, they are ideal tools for this task."

*comment:* which task?

Thanks, this phrase is a bit misleading, indeed. With "this task" we referred to the process of "understanding observed changes in natural and human systems and their respective drivers" (see sentence before). We now phrase it like this: "... they are ideal tools to be applied within ISIMIP3a to advance this understanding."

- Lines 1152 ff:

"The ISIMIP3a evaluation experiments will help to clarify how well the current generation of impact models can explain observed changes in impacted systems based on provided information about the different forcings.

*comment:* the effect of uncertainties in the forcing datasets should be discussed, at least a bit

We have simply extended the sentence to include that aspect:

*"The ISIMIP3a evaluation experiments will help to clarify how well the current generation of impact models can explain observed changes in impacted systems based on provided information about the different forcings that certainly als is subject to uncertainties or even missing (see list above). We at least try to capture part of this uncertainty by providing four different observational atmospheric climate forcing data and associated counterfactual forcings (see section 2.1) and TC windfields derived from two different modelling approaches (see section 3.2)."*

- Lines 1154 ff:

"High explanatory power then allows for impact attribution through the ISIMIP3a attribution experiments following the IPCC-WGII definition of AR6, disentangling changes in climate-related forcings from other drivers of change."

*comment:* more precise references are needed

We have included the full reference to chapter 16 of the IPCC-WGII contribution to the AR6.

- Lines 1159 ff:

"The here presented work can thus lay the ground for urgently necessary works to inform climate litigation, the loss and damage debate, and last but not least also decisions about short term adaptation measures"

*comment:* can you add a few references about scientific work on these issues?

Indeed it is good to add references here and we did so. We now write

"The presented work can thus lay the ground for urgently necessary works to inform climate litigation (Burger et al., 2020; Burger and Tigre, 2023), the loss and damage debate (Mechler et al., 2018; Wyns, 2023), and last but not least also decisions about short term adaptation measures."

**In addition to the above changes, we made the following changes:**

- Three additional authors joined (Ana I. Ayala (isabel.ayala.zamora@ebc.uu.se), Robert Ladwig (ladwigjena@gmail.com), Sam Rabin (sam.rabin@rutgers.edu)). They have contributed as new coordinators of the lake sector and agriculture sector.
- We added a Figure of the precipitation fields associated with Hurrican Harvey that made landfall in Texas (USA) in August 2017, while before we only showed wind fields (Figure 2b), and we changed the caption accordingly. Related variable names in Table 6 have been adjusted to finally agree with the information in the files we are providing.
- Update the Figure on coastal water levels based on the revised underlying manuscript which is now also accepted.